# SoCo: Progressive Spectrum Optimization for Large Language Model Compression

## Abstract

1Large language models (LLMs) have demonstrated remarkable capabilities, yet prohibitive parameter complexity often hinders their deployment. Existing singular value decomposition (SVD) based compression methods equate singular values with component importance, an assumption that often fails to correlate with downstream task performance. In this work, we introduce SoCo (**S**ingular spectrum **o**ptimization for large language model **Co**mpression), a novel framework that learns to rescale SVD components. Concretely, we employ a learnable diagonal matrix to assign importance scores and introduce Progressive Spectrum Optimization, a principled strategy that operates in a single, continuous training run. Inspired by heuristic optimization, this process guides the learnable scores through distinct functional phases—from an initial exploration of the solution space, through an oscillation refinement, to a final, decisive sparsification—thereby navigating the complex optimization landscape to balance compression and performance. Thanks to this adaptive process, SoCo prunes components based on their learned importance, rather than a fixed order. More importantly, amplified scores on preserved components allow them to compensate for the information loss from pruning. Experimental evaluations across multiple LLMs and benchmarks demonstrate that SoCo surpasses state-of-the-art methods in large language model compression.

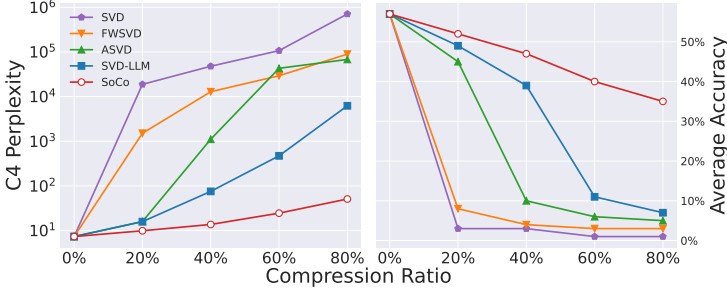

Figure 1: SoCo outperforms existing methods across a range of compression ratios, yielding lower perplexity on C4 (left) and higher average classification accuracy on LM-Evaluation-Harness (right).

## 1 Introduction

Recent advances in large language models (LLMs) have led to remarkable breakthroughs in natural language processing (NLP) tasks (Waswani et al., 2017; Devlin, 2018; Brown et al., 2020; Kaplan et al., 2020; Touvron et al., 2023a; Bie et al., 2024). However, the vast number of parameters in these deep transformer architectures results in prohibitive storage and computational requirements and poses significant challenges for deployment and inference efficiency (Wan et al., 2023; Zhu et al., 2023). Consequently, compression of LLMs has emerged as a vital problem (Dettmers et al., 2022; Ma et al., 2023; Zhong et al., 2024b; Hsu et al., 2022).

A classical approach to model compression performs singular value decomposition (SVD) (Stewart, 1993; Wall et al., 2003) for weight matrices and only preserves the principal components with the largest several singular values, which consist of low-rank matrices and largely reduce the number

of parameters. These SVD-based methods assume that the decomposed components with smaller singular values are less important for model performance. However, the order of singular values only reflects their contribution to low-rank approximation but does not necessarily correlate with the performance of a downstream task. To address this, recent works, such as ASVD (Yuan et al., 2023) and SVD-LLM (Wang et al., 2024), introduce activation awareness with calibration data into the decomposition process to mitigate such mismatch. Nonetheless, these methods still perform truncation by the order of singular values without re-evaluating the intrinsic importance of their corresponding components. As a result, the model performance will be rapidly unacceptable when the compression ratio increases. For example, as shown in Fig. 1, when compression ratio is larger than 40%, all the compared methods including original SVD (Wall et al., 2003), FWSVD (Hsu et al., 2022), ASVD (Yuan et al., 2023), and SVD-LLM (Wang et al., 2024) lead to catastrophic perplexity on C4 (Raffel et al., 2019). Besides, most existing methods cannot compensate for the loss caused by the pruned components (Hsu et al., 2022; Yuan et al., 2023).

In this paper, we propose **SoCo** (**S**ingular spectrum **o**ptimization for large language model **Co**mpression), a novel framework that learns to rescale the singular spectrum in a data-driven manner. Concretely, SoCo introduces a learnable diagonal matrix where each element acts as an importance score, enabling an end-to-end re-evaluation of each SVD component.

To optimize these scores, we address the core challenge of compression: the task is a highly complex combinatorial optimization problem, where direct optimization easily gets trapped in suboptimal local minima. To navigate this complex landscape, we introduce Progressive Spectrum Optimization, a principled strategy inspired by heuristic algorithms (Holland, 1992; Kirkpatrick et al., 1983; Dorigo, 1991; Kennedy & Eberhart, 1995; Glover, 1986) like Simulated Annealing (Kirkpatrick et al., 1983). Operating within a single, continuous training run, this process guides the importance scores through three distinct functional phases. It begins with a compression-driven exploration to rapidly identify a feasible solution space. This is followed by an oscillatory refinement phase to stabilize the scores of borderline components. Finally, a decisive sparsification phase polarizes the scores, creating a clear separation between critical and redundant components.

This adaptive, data-aware strategy allows SoCo to prune components based on their learned, task-aware importance, rather than simply truncating by the fixed order of singular values. This overcomes the task-agnostic limitations of prior studies and enables a more nuanced selection of components. More importantly, the scores of preserved components can be amplified to compensate for information loss from compression. The entire optimization process is highly efficient, as we only train a small set of importance scores while keeping all original model parameters frozen.

The main contributions of this work are:
- We propose SoCo, a learning-based compression framework that optimizes the singular value spectrum through trainable importance scores to overcome the mismatch between the order of singular values from SVD and the true contribution to downstream task performance.
- We design Progressive Spectrum Optimization, a principled, single-run strategy inspired by heuristic optimization. It guides importance scores through functional phases of exploration, refinement, and sparsification to find a superior compression-performance trade-off, where preserved components compensate for pruning loss via amplified scores.
- Extensive evaluations with popular LLMs (from 7B to 30B parameters) across various benchmarks in Tab. 2 demonstrate that SoCo achieves better compression performance than the state-of-the-art methods. Notably, compared with SVD-LLM (Wang et al., 2024), SoCo gains improvements of up to 37.6%, 81.8%, 94.8%, and 99.2% in performance at 20%, 40%, 60%, 80% compression rates on the C4 dataset (Raffel et al., 2019).

## 2 RELATED WORK

**Large Language Model Compression.** LLM compression methods aim to make LLMs more efficient for deployment (Wan et al., 2023; Zhu et al., 2023). Common techniques include quantization, pruning, knowledge distillation, and low-rank decomposition. Quantization reduces the precision of model weights and activations, thereby decreasing memory usage and computational load. (Dettmers et al., 2022; Kim et al., 2024; Shen et al., 2024; Lin et al., 2024). Pruning involves removing less significant weights or neurons from the model to create a sparser architecture. However, pruning often necessitates retraining to recover potential performance degradation (Ma et al., 2023; Ashkboos

et al., 2024; Zhong et al., 2024a; Hsieh et al., 2023). Knowledge distillation transfers knowledge from a large "teacher" model to a smaller "student" model by training the latter to replicate the former's outputs. This approach enables the student model to achieve performance comparable to the teacher model while being more efficient (Zhong et al., 2024b; Muralidharan et al., 2024; Hsieh et al., 2023). Low-rank decomposition compresses weight matrices by factorizing them into smaller, computationally efficient components (Hsu et al., 2022; Yuan et al., 2023; Wang et al., 2024).

**Low-rank Decomposition Compression.** A significant portion of existing low-rank compression techniques (Li et al., 2023; Noach & Goldberg, 2020; Chen et al., 2023; Yang et al., 2020; Hsu et al., 2022; Yuan et al., 2023; Wang et al., 2024; Yang et al., 2024) relies on Singular Value Decomposition (SVD) (Stewart, 1993; Wall et al., 2003) to factorize and compress model parameters. Although original SVD (Stewart, 1993; Wall et al., 2003) decomposes weight matrices into lower-rank components for reconstruction, its objective may lead to information loss if not tailored for downstream tasks. Improved SVD-based methods address these limitations. For example, FWSVD (Hsu et al., 2022) leverages Fisher information to preserve critical components, ASVD (Yuan et al., 2023) scales singular values based on activation sensitivity, and SVD-LLM (Wang et al., 2024) incorporates data whitening and layer-wise updates to compensate for accuracy loss at high compression ratios. However, these approaches depend on a fixed descending order of singular values from SVD, allowing only truncation adjustments without re-evaluating the true significance of each decomposed component. In contrast, our method introduces a learnable mechanism that re-evaluates component importance end-to-end, enabling a more adaptive and effective compression strategy.

## 3 METHOD

Section 3.1 reviews LLM architectures and SVD-based compression. Section 3.2 details the SoCo framework, while Section 3.3 introduces our Progressive Spectrum Optimization. Finally, Section 3.4 examines the learning dynamics, demonstrating how SoCo preserves essential information while effectively reducing parameters.

### 3.1 PRELIMINARIES

Large language models are built on transformer layers that rely on Multi-Head Self-Attention (MHSA) mechanisms. In each layer, the input sequence is transformed into queries, keys, and values via learned linear transformations to compute attention, followed by processing through a feed-forward network (FFN). The weight matrices $W$ (used for query, key, value, and output projections in attention, as well as up, down, and gate projections in the FFN) comprise the majority of the model's parameters. These high-dimensional matrices often exhibit redundancy, with key information concentrated in a few decomposed components (Haink, 2023; Han et al., 2015), making them the primary targets for compression. This phenomenon can be formally captured using SVD, which factorizes a weight matrix $W \in \mathbb{R}^{m \times n}$ as follows:

$$W = U\Sigma V^\top, \tag{1}$$

where $\mathbb{R}$ denotes real numbers, $U \in \mathbb{R}^{m \times min(m,n)}$ and $V \in \mathbb{R}^{min(m,n) \times n}$ are orthogonal matrices and $\Sigma \in \mathbb{R}^{min(m,n) \times min(m,n)}$ is a diagonal matrix containing the singular values in descending order. The magnitude of each singular value quantifies the amount of energy in its corresponding direction, effectively identifying the principal decomposed components of the matrix (Stewart, 1993; Wall et al., 2003). By preserving only the top $k$ largest singular values and their associated vectors, SVD provides a low-rank approximation:

$$W \approx U_k \Sigma_k V_k^\top, \tag{2}$$

which preserves the essential features of the parameter matrices. This property is useful for LLM compression, as it reduces LLM's storage and computational requirements when $k \ll \min(m, n)$.

### 3.2 OVERALL FRAMEWORK

Although recent approaches such as ASVD (Yuan et al., 2023) and SVD-LLM (Wang et al., 2024) introduce compression-specific strategies, they still rely on truncating by the descending order of singular values from SVD. Since the original singular values only reflect their contributions to low-rank approximation, this rigid truncation is not necessarily in accordance with the true importance of

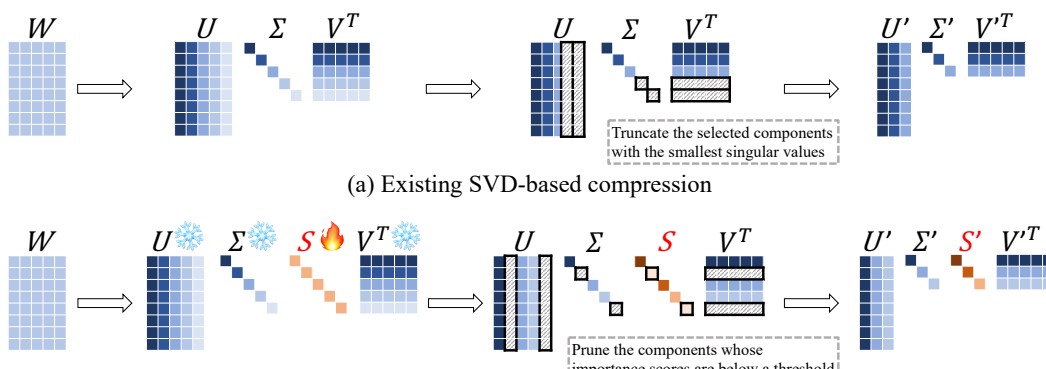

(a) Existing SVD-based compression

(b) **S**ingular spectrum **o**ptimization for **Co**mpression (SoCo)

Figure 2: Illustration of the overall framework of SoCo. The pre-trained weight matrix $W$ is decomposed into $U\Sigma V^\top$, where $\Sigma$ is a diagonal matrix with diagonal elements arranged in descending order. a) Existing SVD-based compression methods truncate the smaller singular values and their corresponding vectors in $U$ and rows in $V^\top$. b) The proposed SoCo assigns a diagonal matrix $S$ as importance scores to singular values in $\Sigma$. After training, components are pruned if their score is below a threshold (e.g., 0.5). The remaining scores rescale the preserved singular values to compensate for pruning loss.

each decomposed component for downstream tasks and may lead to suboptimal performance at higher compression ratios. To address these limitations, we propose SoCo (**S**ingular spectrum **o**ptimization for large language model **Co**mpression), a learning-based framework that adaptively re-evaluates and adjusts the singular spectrum. In SoCo, a trainable diagonal matrix $S \in \mathbb{R}^{min(m,n) \times min(m,n)}$ is introduced to assign importance scores to the singular values. Instead of selecting singular values based on their magnitude, the preserved singular values are determined according to the assigned scores in $S$, as illustrated in Fig. 2. The modified weight matrix $W'$ is defined as

$$W' = U(\Sigma \odot S)V^\top, \tag{3}$$

where $\odot$ denotes element-wise multiplication. After training, $S$ dynamically adjusts the importance distribution of the decomposed components based on their true influences on model performance, thereby improving the trade-off between model compression and performance preservation.

We define the importance scores $S$ as:

$$S = \frac{\lambda_m}{1 + e^{-\lambda_s z + \ln(2\lambda_m - 1)}}, \tag{4}$$

where $z$ is a learnable diagonal matrix (with the same shape as $\Sigma$), $\lambda_m$ controls the overall numerical scale of $S$ (thereby determining the magnitude of re-evaluation), and $\lambda_s$ modulates the steepness of this sigmoid-like function (larger $\lambda_s$ results in a more rapid response to changes in $z$). The term $\ln(2\lambda_m - 1)$ is employed to ensure that the function yields a consistent value at $z = 0$. (More details about $S$ are in A.1 of Appendix.)

To compensate for any deviation introduced by modifying weight matrix $W'$, we further add a trainable deviation term $d$ after the linear transformation of $W'$. The original model parameters, including the singular values $\Sigma$ and singular vectors $U$ and $V$, are frozen during our training.

### 3.3 PROGRESSIVE SPECTRUM OPTIMIZATION

The central challenge guiding our design is balancing compression and performance, two inherently conflicting optimization objectives. Fundamentally, the task we address—selecting an optimal subset of singular components to preserve—is a highly complex combinatorial optimization problem. The solution space for such problems is vast and filled with numerous suboptimal local optima. A direct optimization approach, which must simultaneously reconcile these conflicting goals, can easily become trapped in one of these local minima.

This theoretical pitfall is confirmed by our empirical observations. As demonstrated by the outcome of a direct optimization (visualized as the Phase 1 distribution in Fig. 3), the process converges to a state where importance scores exhibit a continuous and ambiguous distribution. In this state, pruning

Table 1: Conditions and loss functions for each training phase. $step$ denotes the procedure of the whole training process, where $step \in [0, 1]$ (with 0 representing the start and 1 representing the end).

| Phase | Conditions | Loss Functions |
|-------|-----------|----------------|
| 1 | $step < 0.5, r \geq R,$ 
 until $r < R$ | $L_{inc} + L_{dec}$ |
| 2 | $step < 0.5, r < R$ | $L_{inc}$ |
|   | $step < 0.5, r \geq R$ | $L_{inc} + L_{dec}$ |
| 3 | $step \geq 0.5$ | $L_{inc} + L_{spa}$ |

with a fixed threshold is ineffective, as it risks removing numerous components near the threshold whose cumulative influence is significant, leading to substantial performance degradation. This observation underscores the need for a more effective optimization strategy—one that can escape this suboptimal state and create the polarized, sparsified scores necessary for clean pruning.

This specific class of problem—escaping local optima in a complex landscape—is exactly what heuristic algorithms (Holland, 1992; Dorigo, 1991; Kennedy & Eberhart, 1995; Glover, 1986) were designed to address. Among these, Simulated Annealing (Kirkpatrick et al., 1983) provides a particularly apt inspiration. Its core strategy—transitioning from a broad, high-energy exploration of the solution space to a focused, low-energy convergence on a high-quality solution. This principle directly informs our phased approach, motivating a design that can intelligently navigate the optimization landscape rather than prematurely converging to a suboptimal state.

In light of these considerations, we introduce our Progressive Spectrum Optimization strategy. Operating within a single, continuous training run, this unified process instantiates the heuristic principle of a phased, exploration-to-convergence process. It first rapidly achieves the compression ratio target, then refines the importance scores through controlled oscillation, and finally sparsifies them into a polarized distribution. This process is described through three distinct functional phases, each governed by a composite loss function, as summarized in Tab. 1.

### 3.3.1 PHASE 1: COMPRESSION-DRIVEN EXPLORATION

The process begins in a phase of rapid exploration to find configurations that satisfy the target compression ratio R, analogous to the "high-temperature search" in Simulated Annealing (Kirkpatrick et al., 1983). At the start of training, the current compression ratio $r = 1.0$ exceeds the target compression ratio $R$. In this phase, both the compression loss ($L_{dec}$) and the performance-preservation loss ($L_{inc}$) are active. The compression ratio $r$ itself serves as $L_{dec}$, which is defined as:

$$\mathcal{L}_{dec} = r = \frac{\sum_{l=1}^{L} (in_l + out_l) \times c_l}{\sum_{l=1}^{L} (in_l \times out_l)}, \tag{5}$$

where $in_l$ and $out_l$ denote the input and output dimensions of layer $l$, $L$ denotes the number of transformer layers in certain LLM, and $c_l$ is the number of selected decomposed components to preserve in the layer. When working as $L_{dec}$, to enable gradient flow through the hard thresholding operation, we use the Straight-Through Estimator (Bengio et al., 2013; Courbariaux et al., 2015):

$$c_l = \sum_{d=1}^{D} \mathbf{1} \left( S_l^d \geq 0.5 \right)$$

$$= \sum_{d=1}^{D} \mathbf{1} \left( S_l^d \geq 0.5 \right) - sg[S_l^d] + S_l^d, \tag{6}$$

where $S_l^d$ is the importance score of the $d$-th singular value of layer $l$ and $sg[\cdot]$ halts gradient propagation. $\mathbf{1}(\cdot)$ represents the indicator function, which returns 1 if the condition is true and 0 otherwise. The loss $L_{dec}$ thus applies a positive gradient to reduce $S$, favoring compression.

Simultaneously, $L_{inc}$ counteracts excessive compression to maintain performance. We define

$$\mathcal{L}_{inc} = \frac{1}{T} \sum_{t=1}^{T} \left( \sum_{c=1}^{C} y_t^c \log \frac{y_t^c}{p_t^c} \right), \tag{7}$$

where $T$ is the number of tokens at the current training step, $C$ is the vocabulary size, $y_t^c$ and $p_t^c$ are the predicted probability distributions before and after model compression. This KL-divergence loss (Cobbe et al., 2021a) forces the model to preserve performance during compressing.

In practice, the gradient from $L_{dec}$ is numerically dominant over the gradient from $L_{inc}$ (the KL-divergence loss) during this phase. Consequently, the optimization rapidly compresses the model until $r < R$. While this compression-driven phase successfully achieves the target ratio, its aggressive nature produces a predictable yet problematic outcome. As visualized by the blue line (Phase 1) in Fig. 3, the importance scores at the conclusion of this phase remain in the continuous and ambiguous state that we identified as a key challenge. This distribution lacks a clear cutoff for pruning, rendering it an unsuitable final state. This outcome directly necessitates the subsequent refinement phase, which is specifically designed to resolve this ambiguity.

### 3.3.2 PHASE 2: OSCILLATORY REFINEMENT AND STABILIZATION

Once the target ratio is met, the system enters a crucial refinement phase, whose function is analogous to the "gradual cooling and annealing" process in Simulated Annealing (Kirkpatrick et al., 1983). After phase 1, the distribution of importance scores (blue line) in Fig. 3 shows no clear cutoff for selecting critical components, rendering fixed-threshold pruning inefficient. To address this, we employ an alternating optimization strategy: when $r < R$, $L_{dec}$ is deactivated, allowing $L_{inc}$ enables the recovery of critical components that are pruned too aggressively; when $r \geq R$, $L_{dec}$ is reactivated to enforce the compression constraint. This alternation induces controlled oscillations around the target ratio $R$. With continued training and increased data exposure, the importance assessments gradually stabilize. Such re-evaluation allows for a more stable and robust assessment of each component's true importance, leading to a better singular spectrum for compression (see Tab. 5).

### 3.3.3 PHASE 3: DECISIVE SPARSIFICATION AND CONVERGENCE

The final phase introduces a powerful sparsity loss ($L_{spa}$), fundamentally altering the optimization landscape in a process that can be seen as a rapid "cooling and crystallization". After the previous exploration and refinement, enforcing sparsity in the importance scores becomes essential for clearly distinguishing essential components from redundant ones. The sparsity loss $L_{spa}$ is introduced to drive $S_l^d \leq 0.5$ toward 0 (to prune) and $0.5 < S_l^d \leq 1.0$ toward 1 (to preserve):

$$\mathcal{L}_{spa} = \frac{1}{LD} \sum_{l=1}^{L} \sum_{d=1}^{D} spa_l^d, \qquad (8)$$

where the quadratic penalty $spa_l^d$ is defined as

$$spa_l^d = \begin{cases} (S_l^d)^2, & \text{if } S_l^d \leq 0.5, \\ (S_l^d - 1)^2, & \text{if } 0.5 < S_l^d \leq 1.0, \\ 0, & \text{if } 1.0 < S_l^d. \end{cases} \qquad (9)$$

Combined $spa_l^d$ with the continued optimization of $L_{inc}$, phase 3 reduces the contributions of pruned components while reinforcing essential components. As shown by the red curve in Fig. 3, this process "crystallizes" the score distribution into a decisive bimodal shape, creating a wide "flat basin" between the two peaks. This separation allows a threshold (e.g., 0.5) to be set with minimal ambiguity, ensuring a clean pruning decision with negligible performance degradation.

### 3.4 ANALYSIS

As a pioneering work that integrates decomposed component re-evaluation into SVD-based compression, SoCo provides interesting insights. We summarize our findings as follows:

**Validation of Progressive Spectrum Optimization:** As shown in Tab. 5, each phase contributes to progressive performance gains, highlighting the effectiveness of phased optimization. This phenomenon validates our design propels the model beyond local optima to a more optimal state.

**Importance Score Distribution Evolution** Fig. 3 illustrates the evolution of importance score distributions across three phases. In Phase 1, the scores are broadly and continuously distributed. In Phase 2, the distribution polarizes toward two opposing numerical extremes. By Phase 3, $S$ is driven

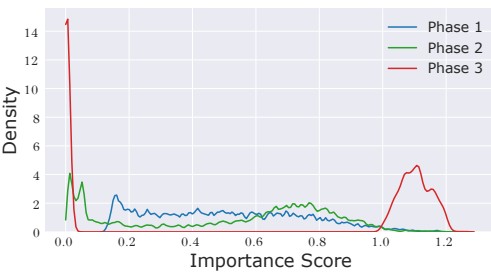 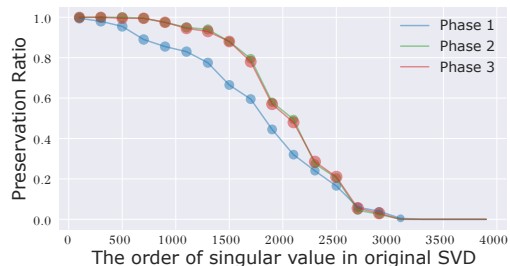

Figure 3: Distribution of importance scores after each of the three optimization phases, illustrating the dynamic re-evaluation process.

Figure 4: Preservation ratio of components in the original SVD-based importance order across the three optimization phases.

further toward a bimodal distribution, with scores converging near 0.0 (discardable) and 1.0 (critical). This separation minimizes ambiguity and limits loss accumulation in threshold-based pruning, as fewer important components fall near the cutoff.

**Adaptation for Component Preservation:** Notably, the emergence of importance scores exceeding 1.0 (the right peak of red curve in Fig. 3) unveils a novel mechanism in our framework: the combination of sparsity and output-alignment loss in Stage 3 amplify preserved singular values to counterbalance the loss from pruned components. Unlike existing SVD-based compression methods, which truncate decomposed components, our SoCo can rescale the singular spectrum broadening the effectiveness and potential of SVD.

As shown in Fig. 4, the evolution of the top singular value preservation ratios across the three phases reveals the intrinsic adjustments in component importance. This dynamic re-assessment ensures the real importance distribution of decomposed components for model compression.

## 4 EXPERIMENTS

### 4.1 EXPERIMENT SETTINGS

**Datasets:** We use WikiText-2 (Merity et al., 2016) training set as the primary training corpus for our pipeline. For evaluation, we employ the WikiText-2 (Merity et al., 2016) and C4 (Raffel et al., 2019), and LM-Evaluation-Harness (Gao et al., 2024) testing sets to ensure a comprehensive assessment.

**Baselines:** Our experiments assess seven LLMs spanning five architectural families and three parameter scales: LLaMA-7B/13B/30B (Touvron et al., 2023a), LLaMA2-7B (Touvron et al., 2023b), OPT-6.7B (Zhang et al., 2022), Vicuna-7B (Chiang et al., 2023), and Mistral-7B (Jiang et al., 2023).

**Evaluation Metrics:** We employ a dual-metric approach for a comprehensive evaluation. We use Perplexity (PPL) as it is a standard indicator of language modeling quality. However, as LIMA (Zhou et al., 2023) highlighted low perplexity does not guarantee strong downstream performance. Therefore, to resolve the uncertainty regarding practical effectiveness, we supplement PPL with direct evaluations of Task-Specific Performance. This provides a balanced view of both fluency and practical capability.

- **Perplexity (PPL):** on WikiText-2 (Merity et al., 2016) and C4 (Raffel et al., 2019).
- **Task-Specific Accuracy:** Evaluated using LM-Evaluation-Harness (Gao et al., 2024) on:
  - 6 classification tasks: OpenbookQA (Mihaylov et al., 2018), WinoGrande (Sakaguchi et al., 2019), HellaSwag (Zellers et al., 2019), ARC-e (Clark et al., 2018), PIQA (Bisk et al., 2019), and MathQA (Amini et al., 2019). "Average↑" in the tables below denotes the mean accuracy across the six classification tasks.
  - 1 generative task: TruthfulQA (factual consistency) (Lin et al., 2021).

### 4.2 MAIN RESULTS

For a comprehensive comparison, we evaluate SoCo against two categories of model compression techniques: SVD-based methods and pruning methods. For fair evaluations, all experiments are conducted without fine-tuning after compression. The main experiments maintain identical settings to the baselines (Hsu et al., 2022; Yuan et al., 2023; Wang et al., 2024).

Table 2: Comparison of SoCo with other SVD-based compression methods on the LLaMA-7B model across various compression ratios. The training dataset is the WikiText-2 training set, and evaluation is conducted on the WikiText-2, C4, and LM-Evaluation-Harness testing sets.

| Ratio | Method | WikiText-2↓ | C4↓ | Openb.↑ | ARC_e↑ | WinoG.↑ | HellaS.↑ | PIQA↑ | MathQA↑ | Average↑ | TruQA.↑ |
|---|---|---|---|---|---|---|---|---|---|---|---|
| 0% | Original | 5.68 | 7.34 | 34% | 75% | 70% | 57% | 79% | 27% | 57% | 30% |
| 20% | SVD | 20061 | 18800 | 5% | 4% | 1% | 3% | 2% | 3% | 3% | 0% |
| | FWSVD | 1727 | 1511 | 9% | 11% | 5% | 8% | 10% | 5% | 8% | 0% |
| | ASVD | 11.14 | 15.93 | 29% | 53% | 64% | 41% | 68% | 17% | 45% | 21% |
| | SVD-LLM | 7.94 | 15.84 | **31%** | 62% | 61% | 45% | 71% | 21% | 49% | 26% |
| | **SoCo** | **6.67** | **9.89** | 28% | **70%** | **65%** | **51%** | **75%** | **25%** | **52%** | **33%** |
| 40% | SVD | 52489 | 47774 | 4% | 4% | 5% | 1% | 3% | 2% | 3% | 0% |
| | FWSVD | 18156 | 12847 | 6% | 5% | 2% | 0% | 5% | 3% | 4% | 0% |
| | ASVD | 1407 | 1109 | 8% | 11% | 9% | 8% | 13% | 8% | 10% | 1% |
| | SVD-LLM | 13.73 | 75.42 | **25%** | 33% | **61%** | 40% | 63% | 12% | 39% | 17% |
| | **SoCo** | **8.60** | **13.71** | 23% | **63%** | 60% | **45%** | **70%** | **23%** | **47%** | **36%** |
| 60% | SVD | 105474 | 106976 | 1% | 3% | 1% | 0% | 1% | 2% | 1% | 0% |
| | FWSVD | 32194 | 29292 | 6% | 2% | 1% | 1% | 2% | 3% | 3% | 0% |
| | ASVD | 57057 | 43036 | 5% | 4% | 6% | 9% | 8% | 5% | 6% | 0% |
| | SVD-LLM | 66.62 | 471.83 | 10% | 5% | 17% | 10% | 21% | 4% | 11% | 1% |
| | **SoCo** | **12.20** | **24.53** | **19%** | **48%** | **55%** | **35%** | **63%** | **21%** | **40%** | **39%** |
| 80% | SVD | 687291 | 708243 | 0% | 4% | 2% | 1% | 1% | 0% | 1% | 0% |
| | FWSVD | 96872 | 89243 | 1% | 2% | 0% | 6% | 9% | 0% | 3% | 0% |
| | ASVD | 80425 | 67927 | 4% | 3% | 3% | 7% | 10% | 1% | 5% | 0% |
| | SVD-LLM | 1349 | 6224 | 7% | 1% | 12% | 10% | 7% | 6% | 7% | 0% |
| | **SoCo** | **21.03** | **50.86** | **14%** | **36%** | **53%** | **28%** | **57%** | **20%** | **35%** | **0%** |

Table 3: Comparison of SoCo with other SVD-based compression methods on various model families at 20% compression ratio. The training dataset is WikiText-2, while evaluation is conducted on the WikiText-2 and LM-Evaluation-Harness's 6 classification task testing sets.

| | OPT-6.7B | | LLaMA2-7B | | Mistral-7B | | Vicuna-7B | | LLaMA-13B | | LLaMA-30B | |
|---|---|---|---|---|---|---|---|---|---|---|---|---|
| Method | WikiText-2↓ | Average↑ | WikiText-2↓ | Average↑ | WikiText-2↓ | Average↑ | WikiText-2↓ | Average↑ | WikiText-2↓ | Average↑ | WikiText-2↓ | Average↑ |
| Original | 10.86 | 52% | 5.47 | 57% | 5.25 | 61% | 6.78 | 56% | 5.09 | 59% | 4.10 | 61% |
| SVD | 66275 | 3% | 18192 | 9% | 159627 | 3% | 18644 | 5% | 946.31 | 21% | 54.11 | 33% |
| FWSVD | 14559 | 6% | 2360 | 12% | 6357 | 8% | 2758 | 9% | 15.98 | 43% | 20.54 | 42% |
| ASVD | 82.00 | 32% | 10.10 | 36% | 13.72 | 32% | 16.23 | 33% | 6.74 | 54% | 22.71 | 44% |
| SVD-LLM | 16.04 | 41% | 8.50 | **53%** | 10.21 | 42% | 8.41 | 51% | 6.61 | 54% | 5.63 | 57% |
| **SoCo** | **10.44** | **50%** | **6.69** | 52% | **6.69** | **54%** | **7.34** | **53%** | **5.92** | **56%** | **5.17** | **58%** |

### 4.2.1 COMPARISON WITH SVD-BASED METHODS

In Tab. 2, SoCo consistently outperforms the competing SVD-based approaches (Hsu et al., 2022; Yuan et al., 2023; Wang et al., 2024) across compression ratios of 20%, 40%, 60%, and 80%. Notably, compared with SVD-LLM (Wang et al., 2024), SoCo achieves relative performance improvements of up to 37.6%, 81.8%, 94.8%, and 99.2% on the C4 dataset (Raffel et al., 2019), corresponding to absolute performance gains ranging from 15.84 to 9.89, 75.42 to 13.71, 471.83 to 24.53, and 6224 to 50.86, respectively. Especially at compression ratios of 60% and 80%, where previous SVD-based methods become impractical, SoCo is the only approach capable of delivering robust performance. Next, Tab. 3 compares our SoCo with other SVD-based compression approaches (Hsu et al., 2022; Yuan et al., 2023; Wang et al., 2024) across different model families and sizes at 20% compression ratio. SoCo consistently achieves superior performance, demonstrating its general applicability across diverse model architectures and sizes.

### 4.2.2 COMPARISON WITH PRUNING METHODS

We compare SoCo with pruning methods, including LLM-Pruner (Ma et al., 2023), SliceGPT (Ashkboos et al., 2024), and BlockPruner (Zhong et al., 2024a), on the LLaMA-7B (Touvron et al., 2023a) at compression ratios of 26%, 33%, 40%, and 47%. For fair comparison, we follow the baseline methods using the Alpaca (Taori et al., 2023) dataset in training, and evaluate on the WikiText-2 testing set. As shown in the Tab. 4, SoCo achieves lower PPLs than all compared pruning methods across the compression ratios, demonstrating its effectiveness beyond just SVD-based approaches.

### 4.3 ABLATION STUDY

**Effect of Optimization Phases.** Tab. 5 reports the performance on the WikiText-2 (Merity et al., 2016) and C4 (Raffel et al., 2019), and LM-Evaluation-Harness (Gao et al., 2024) testing sets. The results demonstrate that our Progressive Spectrum Optimization refines the importance scores,

Table 4: Comparison with pruning methods on the LLaMA-7B model at compression ratios of 26%, 33%, 40%, and 47%. Following the compared methods, we use the Alpaca dataset for training, and the PPL metric is tested on the WikiText-2 testing set.

| METHOD | 10GB(26%) | 9GB(33%) | 8GB(40%) | 7GB(47%) |
|---|---|---|---|---|
| LLM-Pruner | 9.88 | 12.21 | 18.94 | 21.68 |
| SliceGPT | 8.78 | 12.73 | 16.39 | 27.41 |
| BlockPruner | 9.40 | 12.76 | 19.78 | 43.05 |
| **SoCo**$_{alpaca}$ | **8.67** | **10.13** | **12.36** | **16.50** |

Table 5: Ablation study of the three phases of Progressive Spectrum Optimization on the LLaMA-7B (Touvron et al., 2023a) model at compression ratio of 20%. Performance is measured using PPL on WikiText-2 and C4, and the average accuracy on LM-Evaluation-Harness's 6 classification tasks.

| RATIO | PHASE | WikiText-2↓ | C4↓ | Average↑ |
|---|---|---|---|---|
| 20% | 1 | 16.21 | 24.72 | 41.41% |
| | 1+2 | 7.13 | 10.76 | 50.88% |
| | **1+2+3(SoCo)** | **6.67** | **9.89** | **52.48%** |
| 40% | 1 | 109.80 | 178.16 | 33.54% |
| | 1+2 | 27.05 | 57.14 | 41.21% |
| | **1+2+3(SoCo)** | **8.60** | **13.71** | **47.16%** |
| 60% | 1 | 77.51 | 146.26 | 32.32% |
| | 1+2 | 31.26 | 66.29 | 33.06% |
| | **1+2+3(SoCo)** | **12.20** | **24.53** | **40.22%** |
| 80% | 1 | 69.86 | 165.45 | 32.68% |
| | 1+2 | 44.37 | 104.46 | 33.41% |
| | **1+2+3(SoCo)** | **21.03** | **50.86** | **34.63%** |

consistently improving performance phase by phase. The SoCo achieves the best trade-off between compression and performance across all compression ratios.

**Comparison with Fine-Tuned Baselines** A valid concern is that our optimization of importance scores can be viewed as a form of fine-tuning, potentially creating an unfair comparison with baselines that lack a similar process. To address this and ensure an equitable comparison, we evaluate SoCo against baselines augmented with LoRA. We adopt the results and implementation from the SVD-LLM, where both ASVD and SVD-LLM are enhanced with LoRA fine-tuning. Tab. 6 shows that SoCo consistently outperforms both fine-tuned baselines across all compression ratios.

Table 6: Perplexity on C4 for SoCo vs. baselines with LoRA fine-tuning on LLaMA-7B.

| METHOD | 20% | 40% | 60% | 80% |
|---|---|---|---|---|
| ASVD + LoRA | 8.37 | 14.86 | 44.81 | 271.00 |
| SVD-LLM + LoRA | 7.78 | 10.65 | 17.93 | 33.19 |
| **SoCo** | **6.67** | **8.60** | **12.20** | **21.03** |

We attribute this sustained advantage to SoCo's fundamentally different optimization process. The decompositions in ASVD and SVD-LLM are not learnable end-to-end; fine-tuning is applied only after a fixed decomposition, resulting in two separate, sequential optimizations. In contrast, SoCo jointly learns the decomposition (via the importance scores) and fine-tunes the component scaling within a single, end-to-end process. This allows SoCo to achieve a better global optimization of the final compressed weights.

# 5 CONCLUSION

In this paper, we introduce SoCo (**S**ingular spectrum **o**ptimization for large language model **Co**mpression), a novel compression framework that leverages a learnable diagonal matrix to adaptively rescale the singular spectrum of weight matrices. Unlike traditional SVD-based truncation methods, SoCo re-evaluates the intrinsic importance of decomposed components and prunes them accordingly. Our Progressive Spectrum Optimization refines the importance scores of the decomposed SVD components, enabling a more fine-grained and effective compression strategy. Extensive experiments on multiple large language models and benchmarks demonstrate that SoCo consistently outperforms existing SVD-based and pruning methods, delivering superior performance even at high compression ratios. These results highlight the potential of our adaptive, data-driven approach for model compression, paving the way for more efficient deployment of large language models in resource-constrained environments.

## 6 ETHICS STATEMENT

We confirm that this work adheres to the ICLR Code of Ethics. All datasets utilized in our experiments are publicly available and do not contain personally identifiable information. Our research does not involve human subjects, and we have considered the potential societal impacts of our work, finding no immediate ethical concerns or risks of misuse.

## 7 REPRODUCIBILITY STATEMENT

A detailed description of our experimental setup, model architectures, and implementation specifics can be found in Sec. 4. All hyperparameters, training details, and the computational environment are comprehensively documented in Appendix A. Details on the datasets and evaluation metrics are provided in Sec. 4 to facilitate the replication of our results.

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

# A APPENDIX

## A.1 FUNCTIONS AND EQUATIONS

**Function graph of importance scores** $S$**.** As shown in Fig. 5 (with $\lambda_m = 2$ and $\lambda_s = 10$ in SoCo), the function $S$ facilitates performance compensation and rapid convergence by tuning $\lambda_m$ and $\lambda_s$. Specifically, $\lambda_m = 2$ confines the overall numerical range of $S$ to $[0, 2]$, allowing $S$ to compensate for the information loss incurred by pruned components with values exceeding 1.

Meanwhile, $\lambda_s$ modulates the steepness of the sigmoid-like function $S$. As shown in Eq. 10, the derivative $\frac{dS}{dz}$ is directly proportional to $\lambda_s$, indicating that a larger $\lambda_s$ yields a faster response to changes in $z$, thereby accelerating training convergence.

$$\frac{dS}{dz} = \frac{\lambda_m \lambda_s (2\lambda_m - 1) e^{-\lambda_s z}}{\left[1 + (2\lambda_m - 1) e^{-\lambda_s z}\right]^2}. \tag{10}$$

The term $\ln(2\lambda_m - 1)$ in Eq. 4 is included to ensure that the function yields the same value as the sigmoid function at $z = 0$. This consistency makes it easy to set a threshold for $S$.

**Function graph of** $spa$ **in** $L_{spa}$**.** The function $spa$ in $L_{spa}$ drives each importance score $s$ in $S$ toward 0 when $S \leq 0.5$ and toward 1 when $0.5 < S \leq 1$. Moreover, as illustrated by the orange

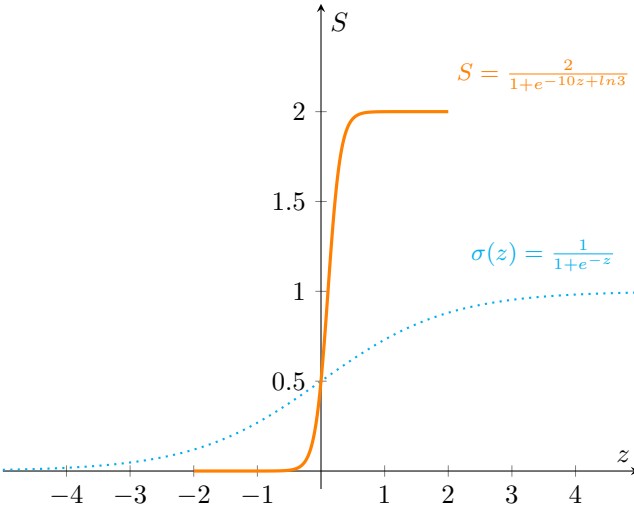

Figure 5: Function graph of the importance scores $S$. The orange function denotes the $S$ used in SoCo, while the blue function represents the standard sigmoid function.

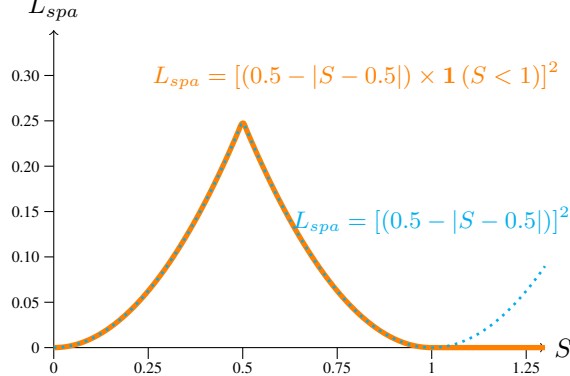

Figure 6: Illustration of the function of $spa$ in $\mathcal{L}_{spa}$ that drive importance scores toward 0 or 1. The orange function used in SoCo converges toward 0 or 1 for scores within the range [0,1] and has no effect on scores above 1. And the blue function ensures that scores greater than 1 are gradually pulled back toward 1.

curve in Fig. 6, $spa_l^d$ imposes a zero gradient for $S > 1$ in Eq. 11, thereby allowing $S$ to exceed 1. This mechanism enables compensation for the information loss induced by pruned components.

Therefore, the gradient of $L_{spa}$ with respect to $S_l^d$ is defined piecewise to capture different behaviors depending on the value of $S_l^d$:

$$\frac{\partial L_{spa}}{\partial S_l^d} = \begin{cases} \frac{2S_l^d}{LD}, & \text{if } S_l^d \leq 0.5, \\ \frac{2(S_l^d-1)}{LD}, & \text{if } 0.5 < S_l^d \leq 1.0, \\ 0, & \text{if } 1.0 < S_l^d. \end{cases} \tag{11}$$

This formulation encourages sparsity by penalizing values closer to the midpoint of the interval $[0, 1]$, while eliminating gradients for the values larger then 1, thereby promoting the compensation mentioned in the paper.

## A.2 ALGORITHM IMPLEMENTATION

**Algorithm 1** Removal of singular values whose importance scores $S < 0.5$.

```
1 #pytorch code
2 idx = (S>=0.5).nonzero().flatten()
3 U_k = U.index_select(1, idx)    # U_k stands for Uk in equation (2)
4 V_k = V.index_select(0, idx)    # V_k stands for Vk in equation (2)
5 # Sigma_k stands for Sigmak in equation (2)
6 Sigma_k = Sigma.index_select(0, idx)
7 # S_k stands for S', importance score after selection
8 S_k = S.index_select(0, idx)
```

### A.3 IMPLEMENTATION DETAILS

**Training Settings.** The WikiText-2 dataset is concatenated and segmented into samples of 1024 tokens. The hyperparameters $\lambda_m$=2.0 and $\lambda_s$=10.0 are used as the default values in $S$. We optimize the models using AdamW with default parameters (weight decay = 0.0, $\beta_1 = 0.9$, $\beta_2 = 0.999$, and $\epsilon = 1 \times 10^{-8}$). Different learning rates for compressing various models are detailed in Tab. 7. A cosine scheduler with a warm-up ratio of 0.05 is applied, and the maximum gradient norm is set to 1.0. Training is performed with a batch size of 1 using BFloat16 precision and gradient checkpointing to optimize GPU memory usage. Most experiments run on a single A100 GPU; however, for LLaMA-30B, we employ FSDP (FULL_SHARD mode) across 4 A100 GPUs.

**Learning Rate for Different Settings.** Our experiments show that using 0.0002 generally achieves state-of-the-art performance across various models and compression targets. The specific values are provided in Tab. 7.

Table 7: Learning Rate across Different Models and Compression Ratio.

| MODEL | TARGET COMPRESSION RATIO | LEARNING RATE |
|---|---|---|
| LLaMA-7B | 20% | 0.002 |
| LLaMA-7B | 40% | 0.002 |
| LLaMA-7B | 60% | 0.002 |
| LLaMA-7B | 80% | 0.002 |
| LLaMA-13B | 20% | 0.002 |
| LLaMA-30B | 20% | 0.002 |
| LLaMA2-7B | 20% | 0.002 |
| OPT-6.7B | 20% | 0.002 |
| Mistral-7B | 20% | 0.002 |
| Vicuna-7B | 20% | 0.002 |

**Threshold Selection for Pruning** The distribution of importance scores in Fig. 3 clearly shows that after Phrase 3, a large portion of values cluster near 0. This creates a wide "flat basin" where any value in that region can be used as an effective threshold. We observe that, regardless of the model, training consistently yields this property, thereby mitigating concerns about generalizability. Moreover, such a basin provides a large tolerance margin for choosing the threshold. For instance, our experiments in Tab. 8 confirm that selecting any threshold within a wide range, such as 0.3 to 0.7, results in negligible performance change. This safe pruning zone stands in sharp contrast to the continuous distribution after Phase 1 (the blue line in Fig. 3), where any fixed threshold would inevitably cut through a dense region of influential components, risking substantial performance degradation.

This flexibility is more user-friendly for broad applications compared to methods requiring a precise threshold where slight deviations can lead to performance degradation.

**GPU Memory and Speed for Inference** Real-world deployment criteria are important. To address this, we have conducted additional experiments in Tab. 9 below. The compare GPU memory usage and inference speed (FLOPs) on LLaMA-7b. This table demonstrates that our method effectively reduces memory consumption during inference and improves inference speed.

**Learnable parameter and Runtime for Training**

Table 8: Impact of pruning threshold selection on LLaMA-7B at 20% compression. Performance remains stable across a wide range of thresholds (0.3 to 0.7), confirming the robustness of the final bimodal score distribution.

| PRUNING THRESHOLD | 0.2 | 0.3 | 0.4 | 0.5 | 0.6 | 0.7 | 0.8 |
|---|---|---|---|---|---|---|---|
| Perplexity↓ | 7.35 | 6.71 | 6.67 | 6.67 | 6.67 | 6.98 | 8.42 |

Table 9: These results demonstrate that the compressed models significantly reduce resource consumption, which is beneficial for real-world applications such as edge deployment or real-time inference.

| COMPRESSION RATIO | PARAMS (B) | GPU MEMORY (GB) | FLOPs (TFLOPS) |
|---|---|---|---|
| 0% | 6.74 | 12.55 | 13.53 |
| 20% | 5.43 | 10.15 | 10.86 |
| 40% | 4.14 | 7.77 | 8.21 |
| 60% | 2.85 | 5.36 | 5.56 |
| 80% | 1.56 | 2.94 | 2.91 |

Tab. 10 summarizes the trainable parameter counts and GPU requirements across various model sizes. Our results indicate that SoCo is highly resource-efficient, requiring only a small number of trainable parameters and modest GPU resources—on the same order of magnitude as common LoRA fine-tuning—to enable rapid singular spectrum optimization.

Table 10: Trainable parameters, GPU resources, and training time required for different model sizes.

| RESOURCE | LLAMA-7B | LLAMA-13B | LLAMA-30B |
|---|---|---|---|
| Trainable parameters | 1.84M (2.83‰) | 2.87M (2.26‰) | 5.59M (1.74‰) |
| GPU number | 1 | 1 | 4 |
| Memory per GPU | 23G | 44G | 36G |

To provide a clear comparison of practical efficiency, we benchmarked the runtime of SoCo against leading methods, ASVD and SVD-LLM, under a unified setting using LLaMA-7B and 256 calibration samples ( 0.5M tokens). The results, summarized in Tab. 11, highlight SoCo's exceptional balance of speed and performance.

SoCo demonstrates a significant efficiency advantage over ASVD, completing its optimization in just 25 minutes compared to ASVD's lengthy 5.5-hour search process. This makes SoCo a far more practical solution for rapid compression cycles. While SVD-LLM's closed-form update is faster at 15 minutes, this speed comes at a considerable performance cost, as shown in our main results (Table 2). SoCo's 25-minute, single-pass optimization achieves a markedly superior compression-performance trade-off. Furthermore, it is worth noting that enhancing SVD-LLM with subsequent fine-tuning (e.g., LoRA) to improve its performance brings its total runtime into a similar range as SoCo's, yet SoCo still maintains a clear performance advantage. Therefore, SoCo represents a highly compelling and efficient approach for achieving high-quality model compression.

### A.4 SUPPLEMENTARY ABLATION STUDY

**Deviation Term $d$.** To compensate for any deviation introduced by modifying the weight matrix $W'$, we add a trainable deviation term $d$ after the linear transformation of $W'$ in SoCo. This deviation term quantifies the average deviation and helps mitigate performance loss. Its effectiveness is demonstrated in Tab. 12.

**Hyper Parameters $\lambda_m$ and $\lambda_s$ in $S$.** The results vary only slightly in Tab. 13, demonstrating the robustness of SoCo with respect to these hyperparameters. We adopt the setting $(\lambda_m, \lambda_s) = (2.0, 10.0)$ and used it in all experiments in the paper, this configuration consistently yielded robust performance across different model sizes, data domains, and compression ratios. Thus, extensive hyperparameter tuning is not required in practice.

**Sensitivity of the Phase 2: Oscillatory refinement and stabilization**

Table 11: Runtime comparison for compressing LLaMA-7B with 256 calibration samples. SoCo achieves a superior performance-to-time trade-off.

| METHOD | KEY STEPS | RUNTIME |
|---|---|---|
| ASVD | Normalize + Search | 5.5 h (5 min + 5.5 h) |
| SVD-LLM | Truncation-Aware Data Whitening + Layer-Wise Update | 15 min (10 min + 5 min) |
| SoCo | Progressive Spectrum Optimization | 25 min |

Table 12: Ablation experiments of SoCo on the use of the deviation term $d$, conducted on the LLaMA-7B model at the compression ratios of 20%, 40%, 60%, and 80%.

| RATIO | METHOD | WikiText-2↓ | C4↓ | Average↑ |
|---|---|---|---|---|
| 20% | w/o Deviation | 6.77 | 9.92 | 52.36% |
| | **w Deviation** | **6.67** | **9.89** | **52.48%** |
| 40% | w/o Deviation | 9.69 | 15.56 | 45.52% |
| | **w Deviation** | **8.60** | **13.71** | **47.16%** |
| 60% | w/o Deviation | 16.27 | 31.80 | 38.01% |
| | **w Deviation** | **12.20** | **24.53** | **40.22%** |
| 80% | w/o Deviation | 33.72 | 81.57 | 33.22% |
| | **w Deviation** | **21.03** | **50.86** | **34.63%** |

Our Phase 2 employs an alternating optimization strategy over the entire model rather than on a per-layer basis. This unified optimization ensures that the overall loss is minimized jointly, reducing the sensitivity to differences in activation distributions across layers. Empirical evidence from our experiments (as shown in the convergence curves in Fig. 7 below) demonstrates stable and consistent convergence across different models and compression targets.

Figure 7: We illustrate the proportion of singular values with importance scores greater than 0.5 during the Progressive Spectrum Optimization to further demonstrate the stability of the optimization process. The training process is illustrated as follows: the Y-axis shows the proportion of singular values with importance scores above 0.5, and the X-axis represents the training steps. Stage 1 (0–500 steps) exhibits rapid convergence; in the early part of Stage 2 (500–3000 steps), the proportion quickly approaches the target value and oscillates within a narrow range; Stage 3 (3000–6000 steps) then maintains a stable proportion. Moreover, across different model sizes of LLaMA, Vicuna, Mistral, and OPT, the convergence in all three phases remains stable and smooth.

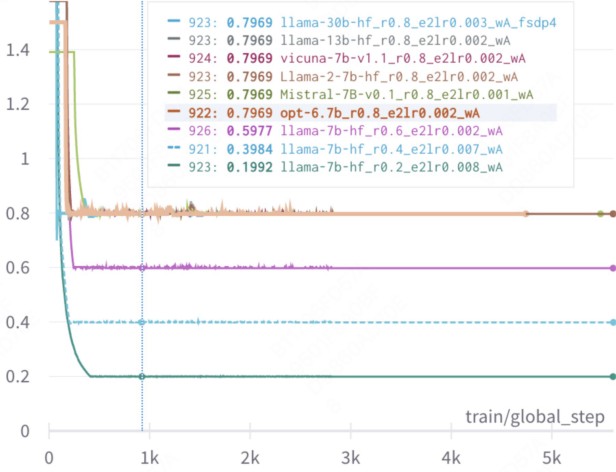

**Analysis of the loss $L_{inc}$ dynamics**

To further elucidate the training dynamics of Progressive Spectrum Optimization, we visualize the trajectory of the performance preservation loss ($L_{inc}$) throughout the optimization process in Fig. 8. It illustrates the evolution of the KL-divergence loss($L_{dec}$) during the compression of LLaMA-7B.

Table 13: Ablation study on the hyperparameters $\lambda_m$ and $\lambda_s$ in $S$. Performance is evaluated via PPL on WikiText-2 and C4, and the averaged accuracy of 6 classification tasks in LM-Evaluation-Harness.

| $\lambda_m$ | $\lambda_s$ | WikiText-2↓ | C4↓ | Average↑ |
|---|---|---|---|---|
| 2.0 | 5.0 | 6.81 | 10.02 | 52.12% |
| **2.0** | **10.0** | **6.67** | **9.89** | **52.48%** |
| 2.0 | 15.0 | 6.73 | 9.97 | 52.34% |
| 2.0 | 20.0 | 6.77 | 9.99 | 52.37% |
| 1.0 | 1.0 | 6.86 | 10.05 | 51.58% |
| 1.0 | 10.0 | 6.80 | 9.99 | 52.42% |
| **2.0** | **10.0** | **6.67** | **9.89** | **52.48%** |
| 4.0 | 10.0 | 6.75 | 9.98 | 51.97% |
| 8.0 | 10.0 | 6.77 | 10.00 | 51.73% |

Figure 8: The trajectory of the performance preservation loss ($L_{inc}$) during training. The curve reveals two distinct phases that align with the functional goals of SoCo: an initial rapid ascent driven by aggressive compression, followed by a gradual recovery as the model refines its importance scores.

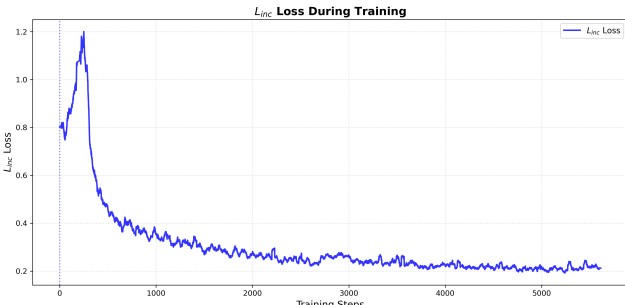

The curve exhibits two distinct behaviors that directly correspond to the functional phases of our framework:

(1) Initial Rapid Ascent (Aligns with Phase 1: Exploration): In the initial stage (approximately steps 0–500), the loss spikes sharply, peaking around step 250. This behavior corresponds to Phase 1, where the optimization is primarily "Compression-Driven". During this phase, the compression loss ($L_{dec}$) exerts a dominant gradient to forcibly drive the model toward the target compression ratio R. As importance scores are rapidly reduced to satisfy this hard constraint, the model parameters undergo significant perturbation, resulting in a temporary spike in $L_{inc}$ as the model's fidelity initially drops.

(2) Subsequent Gradual Recovery (Aligns with Phases 2 & 3: Refinement and Sparsification): Following the peak, the curve enters a long, steady decline, dropping from ∼1.2 to ∼0.2. This recovery spans Phase 2 (Refinement) and Phase 3 (Sparsification). Once the target compression ratio is achieved, the optimization focus shifts from aggressive reduction to performance preservation. In these stages, $L_{inc}$ takes the leading role, effectively guiding the model to recover its capabilities within the constrained low-rank subspace. The convergence to a stable, low-loss state confirms that the model successfully identifies and amplifies the critical components necessary to maintain performance.

## A.5 GENERALIZATION TO ADVANCED MODELS AND BENCHMARKS

**Evaluation on More Domain-Specific or Real-World Tasks** In addition to standard generative and classification benchmarks in the paper (WikiText-2, OpenbookQA, TruthfulQA, etc.), we have conducted further experiments on domain-specific tasks including long-text understanding and reasoning, as well as medical domains in Tab. 14. These additional experiments confirm that SoCo maintains its effectiveness and robustness across a wide range of tasks, thereby validating its generalizability.

Table 14: Evaluation on long-text understanding and reasoning, as well as medical domains.

| COMPRESSION RATIO | LongBench ↑ | LongBench_e ↑ | MedMCQA ↑ | MedQA ↑ | WikiText-2 ↓ | C4 ↓ | PTB ↓ |
|---|---|---|---|---|---|---|---|
| 0% | 64.51% | 70.73% | 26.13% | 28.44% | 5.68 | 7.34 | 13.12 |
| 20% | 62.90% | 68.83% | 27.32% | 26.32% | 6.67 | 9.89 | 13.21 |
| 40% | 57.34% | 64.44% | 30.53% | 27.65% | 8.60 | 13.71 | 18.51 |
| 60% | 49.04% | 54.71% | 31.32% | 28.28% | 12.20 | 24.53 | 33.19 |
| 80% | 44.20% | 53.04% | 31.05% | 27.57% | 21.03 | 50.86 | 69.28 |

**Evaluation on the latest model architectures** To address concerns about performance on the latest generation of models and to demonstrate SoCo's broad applicability, we conducted extensive evaluations on the state-of-the-art LLaMA-3.1-8B (AI@Meta, 2024) and Qwen-2.5-7B (Xu et al., 2025) architectures.

First, to establish a direct performance comparison, we benchmarked SoCo against the SVD-based baselines at representative low (20%) and high (80%) compression ratios. The results, presented in Table 15, confirm that SoCo maintains its significant advantage, consistently outperforming all competing methods on these advanced models.

Table 15: Perplexity comparison of different compression methods on LLaMA-3.1-8B and Qwen-2.5-7B.

| MODEL | COMPRESSION RATIO | METHOD | WikiText-2↓ | C4↓ | PTB↓ |
|---|---|---|---|---|---|
| | 0% | Original | 6.24 | 9.58 | 13.12 |
| | 20% | SVD | 46721 | 23835 | 18793 |
| | 20% | FWSVD | 2034 | 1836 | 1542 |
| | 20% | ASVD | 17.64 | 27.03 | 43.38 |
| | 20% | SVD-LLM | 12.25 | 26.75 | 35.47 |
| LLaMA-3.1-8B | 20% | **SoCo** | **9.24** | **17.70** | **19.80** |
| | 80% | SVD | 603372 | 797613 | 830981 |
| | 80% | FWSVD | 158993 | 127834 | 359826 |
| | 80% | ASVD | 138762 | 74242 | 209251 |
| | 80% | SVD-LLM | 5923 | 11273 | 19872 |
| | 80% | **SoCo** | **46.89** | **234.14** | **359.91** |
| | 0% | Original | 6.85 | 11.86 | 11.37 |
| | 20% | SVD | 17639 | 19769 | 22980 |
| | 20% | FWSVD | 1578 | 2197 | 4305 |
| | 20% | ASVD | 16.37 | 28.92 | 41.66 |
| | 20% | SVD-LLM | 11.76 | 25.35 | 26.72 |
| Qwen-2.5-7B | 20% | **SoCo** | **8.84** | **16.42** | **18.61** |
| | 80% | SVD | 623912 | 717319 | 761919 |
| | 80% | FWSVD | 104283 | 298627 | 349198 |
| | 80% | ASVD | 37392 | 44293 | 98103 |
| | 80% | SVD-LLM | 4730 | 7891 | 10285 |
| | 80% | **SoCo** | **32.41** | **118.77** | **147.73** |

Next, to provide a more granular analysis of SoCo's own robustness, Table 16 details its performance curve across a full spectrum of compression ratios. The results confirm that our method is also effective across the compression ratios on the latest models, demonstrating its broad applicability.

Table 16: Evaluation on the latest large language models across different compression ratios.

| MODEL | COMPRESSION RATIO | WikiText-2↓ | C4↓ | PTB↓ |
|---|---|---|---|---|
| Llama-3.1-8B | 0% | 6.24 | 9.58 | 13.12 |
| Llama-3.1-8B | 20% | 9.24 | 17.70 | 19.80 |
| Llama-3.1-8B | 40% | 13.06 | 35.58 | 38.01 |
| Llama-3.1-8B | 60% | 22.97 | 90.86 | 124.40 |
| Llama-3.1-8B | 80% | 46.89 | 234.14 | 359.91 |
| Qwen2.5-7B | 0% | 6.85 | 11.86 | 11.37 |
| Qwen2.5-7B | 20% | 8.84 | 16.42 | 18.61 |
| Qwen2.5-7B | 40% | 11.33 | 23.93 | 30.26 |
| Qwen2.5-7B | 60% | 16.88 | 45.10 | 64.41 |
| Qwen2.5-7B | 80% | 32.41 | 118.77 | 147.73 |

**Evaluation on Domain-Specific Fine-Tuned Models**

We evaluated its performance on a domain-specific, fine-tuned model. A critical question is whether a compression method preserves the specialized knowledge acquired during fine-tuning, a more chal-

lenging task than compressing general-purpose foundation models. For this, we selected WizardMath-Llama-3-8B (Luo et al., 2023), a model specifically fine-tuned for mathematical reasoning, and evaluated its performance on the GSM8k (Cobbe et al., 2021b) and MATH (Hendrycks et al., 2021) benchmarks.

The results, presented in Tab. 17, demonstrate that SoCo maintains its significant performance advantage even on this highly specialized model.

Table 17: Evaluation on the domain-specific model WizardMath-Llama-3-8B. We report accuracy (%) on GSM8k and MATH datasets at different compression ratios.

| COMPRESSION RATIO | METHOD | GSM8k ↑ | MATH ↑ |
|---|---|---|---|
| 0% | Original | 90.3 | 58.8 |
| 20% | SVD | 6.2 | 4.3 |
| 20% | FWSVD | 12.7 | 9.6 |
| 20% | ASVD | 64.9 | 32.1 |
| 20% | SVD-LLM | 80.4 | 47.0 |
| 20% | **SoCo** | **85.3** | **53.4** |
| 80% | SVD | 3.2 | 1.5 |
| 80% | FWSVD | 4.4 | 2.1 |
| 80% | ASVD | 9.0 | 7.2 |
| 80% | SVD-LLM | 25.2 | 17.7 |
| 80% | **SoCo** | **54.1** | **35.9** |

Across both low and high compression ratios, SoCo substantially outperforms all baselines, indicating that its principled approach to re-evaluating and rescaling the singular spectrum is highly effective at preserving not just general knowledge, but also the critical, fine-grained information embedded during domain-specific fine-tuning. This confirms SoCo's broad utility for compressing a wide range of deployed LLMs.

### A.6 SUPPLEMENTARY COMPARISON WITH BASELINES

**Comparison with Fine-Tuned Baselines** In addition to what discussed in main paper of Tab. 6, while SVD-LLM's initial compression is faster (15 min vs. SoCo's 25 min) in Tab. 11, adding the necessary LoRA fine-tuning brings its total runtime into a comparable range, yet SoCo still delivers superior performance. This highlights SoCo's excellent efficiency-to-performance trade-off.

**SoCo vs. Post-Truncation Singular Value Fine-Tuning**

An insightful question is whether the benefits of SoCo stem from its ability to re-rank and prune components, or simply from the act of rescaling singular values. A straightforward baseline to investigate this is to perform end-to-end fine-tuning directly on the singular values after a standard SVD truncation. This approach keeps the original SVD-based selection of components but allows their magnitudes to be adjusted. To isolate the unique contributions of our method, we conducted a direct comparison against this baseline.

The experiment was performed on LLaMA-2 using the WikiText-2 dataset, and the results are presented in Tab. 18. The data clearly shows that SoCo significantly outperforms the direct fine-tuning baseline across all compression ratios, demonstrating that merely rescaling a fixed, pre-truncated set of components is a suboptimal strategy.

Table 18: Perplexity on WikiText-2 for SoCo vs. end-to-end fine-tuning of singular values after standard SVD truncation. Lower is better.

| METHOD | 20% | 40% | 60% | 80% |
|---|---|---|---|---|
| Post-Truncation SV Fine-tuning | 12.34 | 18.19 | 61.59 | 363.56 |
| **SoCo** | **6.67** | **8.60** | **12.20** | **21.03** |

We attribute SoCo's superior performance to three key design advantages : 1. Global Re-ranking vs. Local Adjustment: The baseline is fundamentally limited as it only performs local adjustments on a subset of components pre-selected by the potentially misleading SVD magnitude order. In contrast, SoCo globally re-evaluates the entire spectrum, allowing it to identify and preserve components that are truly important for the downstream task, leading to a more globally optimal solution. 2. Stable Optimization via Learnable Masks: SoCo's importance scores act as learnable masks initialized to 1, creating a stable and well-conditioned starting point for optimization. Directly tuning the raw singular

values, which can vary by orders of magnitude, is numerically less stable and more challenging for the optimizer. 3. Effective Regularization: The mask-based formulation of our importance scores facilitates meaningful and effective regularization (e.g., the sparsity loss $L_{spa}$), which is difficult to apply in a principled manner when directly fine-tuning the singular values themselves.

### A.7 COMPRESSION RESULT OBSERVATIONS

**Decomposed Components Preservation Ratio Across All Layers in LLaMA at Different Compression Ratios.** The four subplots in Fig. 9 show the proportion of preserved components across the layers of LLaMA-7B at compression ratios of 20%, 40%, 60%, and 80%. In each subplot, each point on the horizontal axis represents a transformer layer, and the plot shows seven colored lines—each corresponding to a different linear module's weight matrix within that layer. The vertical values of these lines indicate the preservation proportion of decomposed components from the original SVD for each module.

Overall, the preservation ratio of decomposed components for the FFN linear modules (down_pro, up_proj, gate_proj) is higher than that for the Multi-Head Attention linear modules (q_proj, k_proj, v_proj, o_proj). Moreover, as the compression ratio increases, the decrease in the preservation ratio is less pronounced for the Multi-Head Attention modules than for the FFN modules.

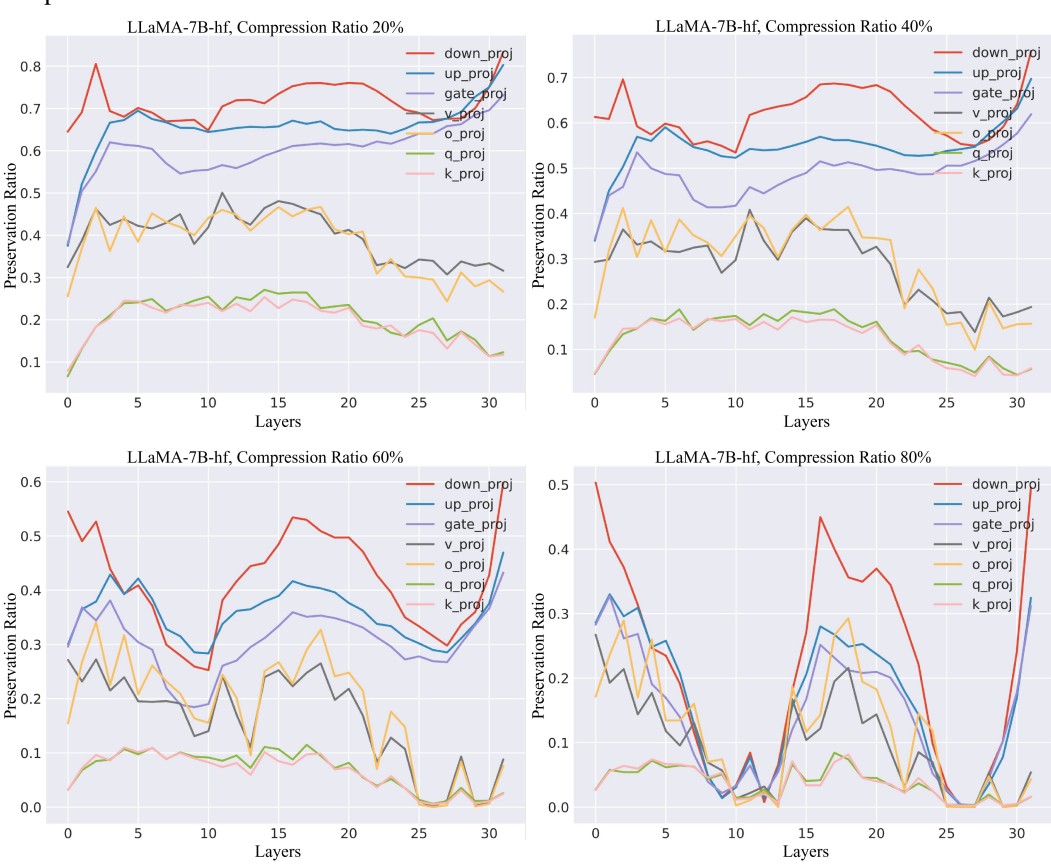

Figure 9: The four subplots depict the preservation ratio of decomposed components in SoCo for LLaMA-7B at the compression ratios of 20%, 40%, 60%, and 80%.

**Decomposed Components Preservation Ratio Across All Layers in Different LLMs at 20% Compression Ratio.**

Fig. 10 presents the preserved component proportions from SoCo across the transformer layers of various models—LLaMA-7B, LLaMA-30B, Mistral-7B, Vicuna-7B, Llama-2-7b, and OPT-6.7B—all evaluated at a 20% compression ratio. In general, the preservation ratio of decomposed components for the FFN linear modules is higher than that for the Multi-Head Attention linear modules (q_proj, k_proj, v_proj, o_proj). Furthermore, no clear pattern emerges across the various models.

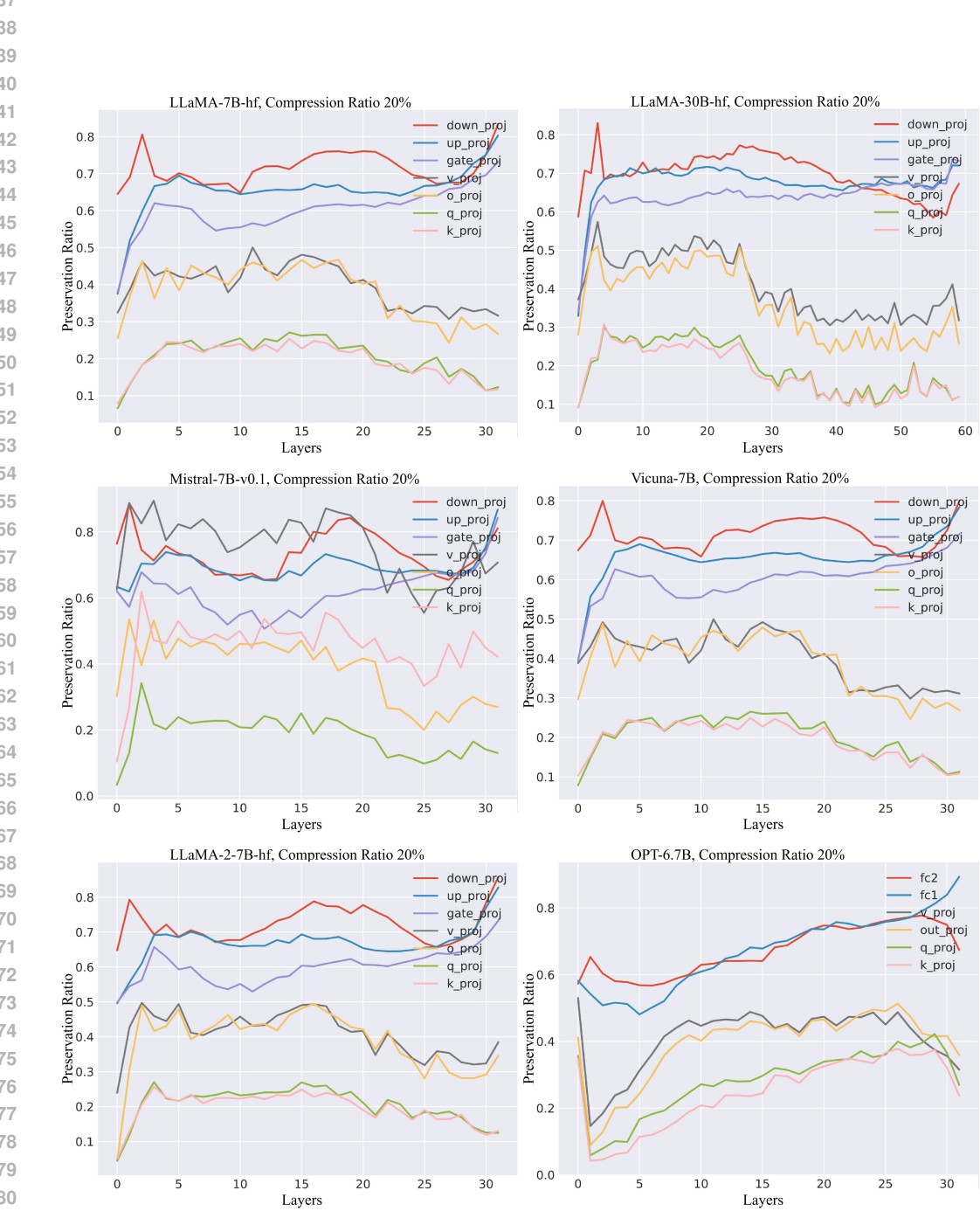

Figure 10: The six subplots depict the preservation ratio of decomposed components in SoCo across all transformer layers and linear modules in various LLMs at 20% compression ratio.

