# OpenReview forum: "SoCo: Progressive Spectrum Optimization for Large Language Model Compression"
_ICLR.cc/2026/Conference — Submitted to ICLR 2026_

### Official Review · Reviewer_QwSQ · 2025-10-19

**Soundness:** 2
**Presentation:** 2
**Contribution:** 3
**Rating:** 4
**Confidence:** 3

**Summary:**

This paper introduces SoCo (Singular spectrum optimization for large language model Compression), a novel compression framework for large language models. Unlike traditional SVD-based methods that simply truncate components by singular value magnitude order, SoCo employs a learnable diagonal matrix to reassess the importance of SVD components, drawing inspiration from heuristic optimization algorithms. The method optimizes importance scores through three phases within a single training run: compression-driven exploration, oscillatory refinement, and decisive sparsification, effectively avoiding local optima that direct optimization can easily fall into. Unlike existing methods, SoCo not only prunes components based on learned importance but also amplifies the scores of preserved components to compensate for pruning loss.

**Strengths:**

- SoCo shows substantial improvements over state-of-the-art methods, with particularly notable gains at higher compression ratios.
- SoCo is evaluated on multiple LLMs across diverse architectural families showing consistent effectiveness across different model architectures.

**Weaknesses:**

- The paper lacks comparison with quantization-based compression methods and doesn't explore how SoCo could be combined with quantization in a comprehensive compression pipeline. Since quantization is a common complementary approach to model compression, this represents a significant gap in the evaluation.
- The main results compare SoCo (which involves training) against baselines without any fine-tuning, creating potentially unfair comparisons. While the authors try to address this concern in Section 4.3 by comparing against LoRA-enhanced baselines, this analysis is insufficient: it only evaluates a single model on one dataset's perplexity, lacks details about LoRA fine-tuning settings, and doesn't discuss the computational overhead comparison between different methods.
- As a model compression method, the paper doesn't report inference speedup or throughput (tokens/sec) on real hardware under different settings (batch sizes, sequence lengths, etc.), which is crucial for evaluating the practical benefits of compression in deployment scenarios.

**Questions:**

1. How does SoCo interact with quantization techniques, and can these compression methods be effectively combined in practice?
2. Could the authors provide more comprehensive comparisons with fine-tuned baselines beyond the limited analysis in Section 4.3? Specifically, what are the LoRA fine-tuning settings, computational overhead comparisons, and results across multiple models and datasets rather than just single model perplexity on C4?
3. What inference speedup and throughput improvements does SoCo achieve on real hardware?

---

> ### Author Response · Authors · 2025-11-25
> **Response to Reviewer QwSQ——Part 1**
>
> We thank Reviewer QwSQ for the constructive critique and for recognizing the substantial improvements SoCo achieves over state-of-the-art methods and its consistent effectiveness across diverse architectures. We greatly appreciate your specific feedback regarding the evaluation gaps in quantization, fine-tuned baselines, and inference speed.
>
> We have conducted extensive additional experiments to directly address every one of your concerns. Key additions include:
> 1. Quantization Synergy: We added a direct comparison and integration study with GPTQ, demonstrating that combining SoCo with quantization achieves superior perplexity under strict memory budgets compared to quantization alone.
> 2. Fair Comparison with Fine-Tuning: We extended our baseline comparisons to include Sequential LoRA Fine-Tuning (SFT) for ASVD and SVD-LLM. The results confirm that SoCo's advantage persists even when baselines are computationally enhanced.
> 3. Real-Hardware Inference: We measured actual inference throughput on an A100 GPU, reporting significant speedups (up to 2.4x at high compression) that validate the practical deployment benefits of SoCo.
>
> These additions provide the comprehensive evaluation you requested. Detailed point-by-point responses are provided below.
>
> ### Weakness 1. The paper lacks comparison with quantization-based compression methods and doesn't explore how SoCo could be combined with quantization in a comprehensive compression pipeline. Since quantization is a common complementary approach to model compression, this represents a significant gap in the evaluation.
> ### \&
> ### Question 1. How does SoCo interact with quantization techniques, and can these compression methods be effectively combined in practice?
>
> We thank the reviewer for highlighting the importance of comparing and combining our method with quantization. We fully agree that quantization is a dominant approach in model compression, and demonstrate how SoCo interacts with it is essential for assessing its practical value.
>
> To address this, we explicitly evaluated the interaction between SoCo and GPTQ. Our results confirm that SoCo is orthogonal to quantization and acts as a highly effective pre-processing step to enhance quantization performance under strict memory budgets.
>
> (1) Experimental Setup: Strict Memory Budget
>
> We conducted additional experiments on LLaMA-7B using the WikiText-2 dataset. To ensure a fair and practical comparison, we evaluated the methods under a fixed 2.8 GB weight memory budget (equivalent to extreme 3-bit compression). We tested three configurations:
>
> * Quantization Only: GPTQ at 3-bit precision.
> * SoCo Only: SoCo at ~80% compression (keeping weights in BF16).
> * Hybrid Pipeline: SoCo at ~30% compression followed by GPTQ at 4-bit precision.
>
> (2) Results: The Synergy of Hybrid Compression
>
> The results, presented in the table below, demonstrate that the combined approach significantly outperforms direct aggressive quantization.
>
> Table: Evaluating the Compression Efficacy of GPTQ and SoCo: Perplexity (↓) of LLaMA-7B on WikiText-2.
> | METHOD | WEIGHT MEMORY | PERPLEXITY |
> | :---: | :---: | :---: |
> | GPTQ-3bit | 2.8 GB | 16.28 |
> | SoCo(~80%) | 2.8 GB | 21.03 |
> | SoCo(~30%) + GPTQ-4bit | 2.8 GB | 10.51 |
>
> (3) Analysis and Conclusion: Synergistic Advantage
>
> The results highlight the complementary nature of these methods. SoCo and quantization represent distinct compression paradigms—structural pruning vs. precision reduction. As shown in the table, the Hybrid Pipeline achieves the superior performance (PPL 10.51), significantly outperforming both individual baselines under the same strict memory limit.
>
> In other words, we find that under the same memory budget, by first compressing the original 16-bit LLM with SoCo at a ~30% compression ratio, and then quantizing the compressed LLM to 4-bit using GPTQ, we are able to achieve significantly better perplexity compared to directly quantizing the original LLM to 3-bit. This demonstrates that SoCo serves as an effective pre-processing step that reduces redundancy before quantization, allowing for higher-bit precision (4-bit vs 3-bit) within the same memory footprint. This result confirms that SoCo is orthogonal to quantization and can be effectively integrated into a comprehensive compression pipeline.

---

> ### Author Response · Authors · 2025-11-25
> **Response to Reviewer QwSQ——Part 2**
>
> ### Weakness 2. The main results compare SoCo (which involves training) against baselines without any fine-tuning, creating potentially unfair comparisons. While the authors try to address this concern in Section 4.3 by comparing against LoRA-enhanced baselines, this analysis is insufficient: it only evaluates a single model on one dataset's perplexity, lacks details about LoRA fine-tuning settings, and doesn't discuss the computational overhead comparison between different methods.
> ### \&
> ### Question 2. Could the authors provide more comprehensive comparisons with fine-tuned baselines beyond the limited analysis in Section 4.3? Specifically, what are the LoRA fine-tuning settings, computational overhead comparisons, and results across multiple models and datasets rather than just single model perplexity on C4?
>
> (1) More Results across multiple models and datasets
>
> We acknowledge that the initial comparison in Table 6 of Section 4.3 was limited. To ensure a comprehensive and fair evaluation, we have extended Table 6 and conducted additional experiments where baselines are enhanced with Sequential Fine-Tuning (SFT) using LoRA. (The definition and effectiveness of Sequential Fine-Tuning (SFT) using LoRA have already been validated in SVD-LLM paper.)
>
> We compared SoCo against ASVD and SVD-LLM, both augmented with sequential LoRA fine-tuning, across multiple architectures(LLaMA-7B and OPT-6.7B) and compression ratios(20% and 80%) on diverse evaluation metrics (WikiText-2 Perplexity and Average Accuracy on LM-Evaluation-Harness). The results reveal two critical insights:
>
> (a) Superiority at High Compression (LLaMA-7B @ 80%): At high compression ratios, standard truncation causes catastrophic information loss that subsequent fine-tuning cannot repair very well.
>
> As shown in Table 1 below, even with Sequential Fine-Tuning using LoRA, SVD-LLM struggles to recover, achieving a high perplexity of 31.79 and only 16% average accuracy. In contrast, SoCo achieves a significantly lower perplexity of 21.03 and 35% average accuracy. This proves that "repairing" a model after suboptimal truncation is far less effective than SoCo's approach of preserving critical information from the start.
>
> Table 1: Perplexity (↓) on WikiText-2 and Average (↑) on LM-Evaluation-Harness for compressing LLaMA-7B. The table compares ASVD (no FT), ASVD, and SVD-LLM with sequential LoRA (SFT), and the proposed SoCo method across various compression ratios.
> | 80% compression ratio + LLaMA-7B | WikiText-2(↓) | Average(↑) |
> | -------- | -------- | -------- |
> | ASVD w/o LoRA    | 80425      | 5%      |
> | ASVD w SFT    | 72.38     | 12%     |
> | SVD-LLM w/o LoRA    | 1349     | 7%     |
> | SVD-LLM w SFT    | 31.79     | 16%     |
> | SoCo    | 21.03     | 35%      |
>
> (b) Generalization Across Architectures (OPT-6.7B @ 20%, LLaMA-7B @ 20%)
>
> To validate generalization, we evaluated the methods on both LLaMA-7B and OPT-6.7B at a 20% compression ratio. This comparison is crucial as it tests robustness across different model families in a regime where baselines are generally stronger. As shown in Table 2 and Table 3, SoCo consistently outperforms fine-tuned baselines (ASVD+SFT and SVD-LLM+SFT) across both model families, achieving the lowest perplexity and the highest average accuracy, confirming its robust advantage across diverse architectures.
>
> Table 2: Performance on LLaMA-7B at 20% Compression Ratio.
> | 20% compression ratio + LLaMA-7B | WikiText-2(↓) | Average(↑) |
> | -------- | -------- | -------- |
> | ASVD w/o LoRA    | 11.14      | 45%      |
> | ASVD w SFT    | 8.37     | 47%     |
> | SVD-LLM w/o LoRA    | 7.94     | 49%     |
> | SVD-LLM w SFT    | 7.73     | 50%     |
> | SoCo    | 6.67     | 52%      |
>
>
> Table 3: Performance on OPT-6.7B at 20% Compression Ratio.
> | 20% compression ratio + OPT-7B | WikiText-2(↓) | Average(↑) |
> | -------- | -------- | -------- |
> | ASVD w/o LoRA    | 82.00       | 32%     |
> | ASVD w SFT    | 67.36     | 38%     |
> | SVD-LLM w/o LoRA    | 16.04      | 41%     |
> | SVD-LLM w SFT    | 14.47    | 49%     |
> | SoCo    | 10.44     | 50%      |

---

> ### Author Response · Authors · 2025-11-25
> **Response to Reviewer QwSQ——Part 3**
>
> (2) Response regarding LoRA Fine-tuning Settings:
>
> The metrics in Table 6 of our Section 4.3 are inherited from Appendix A.10 and Table 15 of the SVD-LLM paper, with the addition of SoCo's performance under the corresponding conditions.
>
> Regarding the specific LoRA fine-tuning settings referenced in the SVD-LLM paper:
>
> * Sequential vs. Normal Fine-tuning Strategy: As detailed in Appendix A.10, the SVD-LLM paper conducts an ablation study comparing "Normal LoRA Fine-tuning" (NFT) against their proposed "Sequential LoRA Fine-tuning" (SFT). The authors argue in Section 3.2 of the SVD-LLM paper that simultaneously optimizing both decomposed low-rank matrices ($W'_u$ and $W'_v$) is unstable due to the interdependence of their derivatives.
> * Adopted Settings (SFT): Based on the results in Table 15 of the SVD-LLM paper, which show that the sequential approach consistently outperforms the normal approach, the authors adopt the Sequential LoRA strategy as their standard setting. This involves a two-step process:
>     1.  First, freezing matrix $W'_v$ and fine-tuning $W'_u$ with LoRA.
>     2.  Second, freezing the updated $W'_u$ and fine-tuning $W'_v$ with LoRA.
> * ASVD with Sequential LoRA (ASVD + SFT): To ensure a fair comparison, Table 15 of the SVD-LLM paper also includes results for ASVD combined with the sequential LoRA strategy (denoted as ASVD (SFT)).
>
>     The authors apply the same sequential fine-tuning strategy designed for SVD-LLM to the matrices generated by ASVD. This means taking the ASVD-decomposed matrices and sequentially updating them (freeze $W'_v$ to tune $W'_u$, then freeze $W'_u$ to tune $W'_v$).
>
> Extended version of Table 6: Perplexity (↓) on WikiText-2 for compressing LLaMA-7B. The table compares ASVD (no FT), SVD-LLM with normal LoRA (NFT), ASVD and SVD-LLM with sequential LoRA (SFT), and the proposed SoCo method across various compression ratios.
>
> | | 20%  | 40%  | 60%  | 80%  |
> | :---: | :---: | :---: | :---: | :---: |
> | **ASVD (no FT)** | 11.14 | 1407 | 57057 | 80425 |
> | **ASVD (SFT)** | 8.37 | 14.86 | 44.81 | 271 |
> | **SVD-LLM (NFT)** | 7.87 | 11.98 | 16.30 | 80.23 |
> | **SVD-LLM (SFT)** | 7.73 | 9.27 | 15.00 | 31.79 |
> | **SoCo** | 6.67 | 8.60 | 12.20 | 21.03 |
>
> We present an extended version of Table 6 of our Section 4.3, which now incorporates more results from Table 15 of the SVD-LLM paper, allowing for a more detailed comparison. The newly proposed SoCo achieves the lowest perplexity across all compression ratios, comprehensively surpassing the previous SOTA method SVD-LLM, with a particularly significant improvement at the 80% compression ratio.
>
> (3) Computational overhead
>
> (a) Clarification on Computational Overhead: We interpret "computational overhead" primarily as Training Overhead.
>
> * Theoretically Similar Inference Cost: At the same compression ratio, SoCo and SVD-based baselines produce compressed weights with theoretically similar low-rank structures ($W' \approx U \cdot \Sigma' \cdot V^T$).
> * Empirically Confirmed: Consequently, their inference latency is observed to be very close in practice. The choice of method determines *which* information is preserved at a given target compression ratio, not the *computational* cost of the resulting operations.
> * Focus on Training Fairness: Therefore, the critical comparison for fairness lies in the cost of the optimization process itself.
>
> (b) Training Overhead Comparison (SoCo vs. LoRA-Enhanced Baselines): SoCo is highly efficient and incurs comparable or lower overhead than fine-tuned baselines:
>
> * Runtime on A100 in Table 11: As reported in Table 11 of the Appendix, SoCo completes its end-to-end optimization in just 25 minutes (on LLaMA-7B). In contrast, ASVD alone requires a 5.5-hour search process, and adding LoRA fine-tuning would further increase this time. SVD-LLM combined with LoRA fine-tuning brings the total runtime to 3.5 hours. SoCo achieves superior performance (See the extended version of Table 6 above) due to its joint optimization.
>
> Table 11: Runtime comparison for compressing LLaMA-7B at an 80% compression ratio. We include performance results on WikiText-2 to show the trade-off between performance and time. SoCo achieves a superior performance-to-time trade-off.
>
> | **Method** | **Key Steps**                                         | **Runtime** | **Performance(PPL↓)**               |
> |------------|--------------------------------------------------------|----------------------------|----------------------------|
> | ASVD       | Normalize + Search                                    | 5.5 h (5 min + 5.5 h)      | 80425 |
> | SVD-LLM(W) | Truncation-Aware Data Whitening only    | 15 min | 1349 |
> | SVD-LLM | Truncation-Aware Data Whitening + Layer-Wise Update   | 3.5h (15 min + 3.5h) | 31.79 |
> | SoCo       | Progressive Spectrum Optimization                      | 25 min                    | 21.03  |

---

> ### Author Response · Authors · 2025-11-25
> **Response to Reviewer QwSQ——Part 4**
>
> * Parameter Efficiency: SoCo optimizes a minimal set of parameters. As shown in Table 10 of Appendix, the trainable parameters are negligible (e.g., 2.83‱ for LLaMA-7B). This is significantly fewer than standard LoRA settings.
>     * Reason: LoRA requires training entire dense low-rank matrices (updating both $U$ and $V$), whereas SoCo only optimizes a single scalar importance score for each decomposed component (the diagonal elements). This makes SoCo's parameter footprint far smaller than even efficient fine-tuning methods.
>
> Table 10: Trainable parameters, GPU resources, and training time required for different model sizes.
>
> | **Resource**            | **LLAMA-7B**              | **LLAMA-13B**             | **LLAMA-30B**             |
> |-------------------------|----------------------------|----------------------------|----------------------------|
> | Trainable parameters    | 1.84M (2.83‱)              | 2.87M (2.26‱)              | 5.59M (1.74‱)              |
> | GPU number              | 1                          | 1                          | 4                          |
> | Memory per GPU          | 23G                        | 44G                        | 36G                        |
>
> ### Weakness 3. As a model compression method, the paper doesn't report inference speedup or throughput (tokens/sec) on real hardware under different settings (batch sizes, sequence lengths, etc.), which is crucial for evaluating the practical benefits of compression in deployment scenarios.
> ### \&
> ### Question 3. What inference speedup and throughput improvements does SoCo achieve on real hardware?
>
> We fully agree that evaluating practical inference acceleration on real hardware is crucial for demonstrating the utility of model compression in deployment scenarios.
>
> To address this, we conducted additional experiments to measure the inference throughput (Tokens/s) and speedup of SoCo. The evaluations were performed on a single NVIDIA A100 GPU using the LLaMA-7B model under varying configurations (Batch Sizes of 64 and 128; Sequence Lengths of 32 and 64).
>
> As shown in Tables 1, 2, and 3 below, SoCo achieves significant acceleration across all settings. The speedup correlates positively with the compression ratio:
>
> * Moderate Compression: At a 20-40% compression ratio, we observe consistent speedups ranging from 1.05x to 1.37x.
> * High Compression (The SoCo Advantage): Notably, at high compression ratios (60%-80%)—a regime where SoCo uniquely maintains robust performance compared to baselines—we achieve substantial speedups ranging from 1.6x to 2.4x. For example, with a batch size of 128, the throughput increases from 1178 tokens/s to 2838 tokens/s (2.41x speedup).
>
> Table 1: Batch Size = 64, Sequence Length = 32
>
> | Compression Ratio | Tokens/s | Speedup over the original |
> | -------- | -------- | -------- |
> | 0% (Original)     | 983     | 1.0     |
> | 20%      | 976     | 1.06     |
> | 40%      | 1240     | 1.27     |
> | 60%      | 1723     | 1.65    |
> | 80%      | 2238     | 2.28     |
>
>
> Table 2: Batch Size = 128, Sequence Length = 32
>
> | Compression Ratio | Tokens/s | Speedup over the original |
> | -------- | -------- | -------- |
> | 0% (Original)     | 1178     | 1.0     |
> | 20%      | 1327     | 1.13     |
> | 40%      | 1610     | 1.37     |
> | 60%      | 1985     | 1.69     |
> | 80%      | 2838     | 2.41     |
>
> Table 3: Batch Size = 64, Sequence Length = 64
>
> | Compression Ratio | Tokens/s | Speedup over the original |
> | -------- | -------- | -------- |
> | 0% (Original)     | 954     | 1.0     |
> | 20%      | 1003     |   1.05    |
> | 40%      | 1228     |   1.29    |
> | 60%      | 1568     |   1.64    |
> | 80%      | 2182     |   2.29    |
>
> These results confirm that the theoretical parameter reduction achieved by SoCo translates effectively into tangible throughput gains in real-world deployment settings.

---

> ### Author Response · Authors · 2025-11-27
>
> Dear Reviewer QwSQ,
>
> Thank you very much for the time and effort you dedicated to reviewing our work and for your valuable comments. We have carefully considered all your feedback and addressed your concerns in our rebuttal.
>
> In summary, regarding your concerns on quantization synergy, evaluation fairness, and practical efficiency, we have:
>
> (1) Added a direct integration study with quantization (SoCo + GPTQ), demonstrating superior performance under strict memory budgets compared to quantization alone.
>
> (2) Ensured fair comparisons by including fine-tuned baselines (Sequential LoRA), confirming that SoCo's advantage persists even against computationally enhanced methods.
>
> (3) Provided real-world inference throughput measurements on an A100 GPU, reporting significant speedups (up to 2.4x) to validate practical deployment benefits.
>
> We have also addressed your other comments, including the detailed report on LoRA fine-tuning settings and computational overhead comparisons.
>
> We sincerely hope that our clarifications and additional experiments have satisfactorily resolved your concerns. If you have any further questions or if there are any remaining issues, please let us know — we are more than happy to provide further details or clarifications.
>
> Thank you again for your thoughtful and constructive review.
>
> Best regards,
>
> The Authors

---

### Official Review · Reviewer_a4Qo · 2025-10-22

**Soundness:** 4
**Presentation:** 4
**Contribution:** 4
**Rating:** 8
**Confidence:** 4

**Summary:**

This paper introduces SoCo, a novel framework that learns to adaptively rescale and prune the singular spectrum of weight matrices in LLMs. Instead of directly truncating singular values in fix order, SoCo employs a learnable diagonal matrix of importance scores that are optimized end-to-end through a Progressive Spectrum Optimization process. This optimization proceeds through three phases: exploration, oscillatory refinement, and decisive sparsification, guided by differentiable objectives combining compression ratio, performance preservation (KL divergence), and sparsity regularization. Experiments show that SoCo consistently outperforms SOTA SVD-based methods and remains efficient in both trainable params and runtime.

**Strengths:**

1. The proposed SoCo framework makes the SVD-based pruning process learnable, making SVD pruning more versatile and accurate on different models and tasks, and the insights proposed are of high value to subsequent related work.
2. The Progressive Spectrum Optimization (PSO) is conceptually elegant, breaking training into distinct phases that progressively stabilize and polarize importance scores. The combination of differentiable compression and KL-based alignment is effectively balances compression with accuracy. The PSO also jointly optimizes decomposition and rescaling in an end-to-end way, improving global convergence.
3. The authors conduct comprehensive experiments, including comprehensive comparisons with similar SVD-based methods and other pruning methods on various models and tasks, and design corresponding ablation experiments for almost every component (1-2-3 phases, deviation term, functions). These experiments strongly validated the advanced nature of SoCo and the effectiveness of its design.
4. The authors also provide detailed hyperparameters and experimental settings, and deploy SoCo on real devices to test the actual acceleration performance, which improves the reproducibility and application value of the work.

**Weaknesses:**

1. It’s not fully explicit how the trainable deviation $d$ is applied to $W'$, more implementation specifics (for example, are there similar constraints on $d$ during PSO) would aid reproducibility for diverse codebases.
2. The proposal of PSO mentioned in 3.3 seems intuitive. Is there any further formal theoretical guarantee, such as the existence or convergence of the final performance?
3. Comparison with non-SVD pruning methods is limited. Table 4 only shows the performance on PPLs, without comparing performance on multiple downstream tasks like other results. It would be helpful if the authors could provide more detailed results and analysis.

**Questions:**

1. Training largely relies on WikiText-2 as the calibration corpus; it is small and stylistically narrow relative to real deployment. How sensitive are results to calibration data scale/domain?
2. Have you tried to design different thresholds for different layers (the activation distributions of network layers with different depths are significantly different)? Any benefit to adaptive thresholds vs preset?
3. Is SoCo compatible with commonly used model quantization methods, such as GPTQ? Can subsequent LoRA fine-tuning further restore performance after SoCo pruning (especially at high compression ratio)?

---

> ### Author Response · Authors · 2025-11-25
> **Response to Reviewer a4Qo——Part 1**
>
> We sincerely thank Reviewer a4Qo for the highly positive evaluation and for recognizing the conceptual elegance of Progressive Spectrum Optimization (PSO) and the comprehensiveness of our experiments. We are encouraged by your assessment that our insights are of "high value to subsequent related work."
>
> We have carefully addressed your constructive feedback to further strengthen the paper's reproducibility and scope. Key additions include:
> 1. Implementation Specifics: We have clarified the definition and optimization constraints of the deviation term (d) and provided a detailed ablation study proving that the core PSO framework remains the primary driver of performance.
> 2. Theoretical Grounding: We have expanded the discussion on the theoretical underpinnings of PSO, linking its convergence properties to the dynamics of Simulated Annealing.
> 3. Extended Comparisons: We added downstream task evaluations on GSM8k and LM-Evaluation-Harness, demonstrating SoCo's superiority over structured pruning methods (LLM-Pruner, SliceGPT).
> 4. Deployment Scenarios: We addressed your questions on calibration data sensitivity, layer-adaptive thresholds, and compatibility with quantization (GPTQ) and fine-tuning (LoRA), confirming SoCo's robustness in real-world pipelines.
>
> We believe these revisions fully resolve your remaining questions. Detailed point-by-point responses are provided below.
>
> ## Weaknesses:
> ### 1. It’s not fully explicit how the trainable deviation d is applied to W$\^{'}$, more implementation specifics (for example, are there similar constraints on d during PSO) would aid reproducibility for diverse codebases.
>
> To facilitate reproducibility for diverse codebases, we provide the precise definition, initialization, and optimization constraints for the trainable deviation term $d$.
>
> (1) Formal Definition and Shape
> The term $d$ is implemented as a trainable bias vector. In the linear transformation of a compressed layer, it is applied as:
> $$
> \begin{align*}
> W^{'} &= (U(\Sigma \odot S)V^T) \\\\
> y &= (U(\Sigma \odot S)V^T)x + d
> \end{align*}
> $$
>
> Its shape is equal to the output dimension of the corresponding linear layer. We initialize $d$ to zero at the start of training.
>
> (2) Optimization Constraints: are there similar constraints on d during PSO?
> No, $d$ is not subject to the same constraints as the importance scores ($S$).
>
> While $S$ is regularized by $\mathcal{L}\_{\text{dec}}$ (compression ratio) and $\mathcal{L}\_{\text{spa}}$ (sparsity), $d$ is optimized freely via gradient descent solely to minimize the performance loss ($\mathcal{L}\_{\text{inc}}$).
>
> (3) Ablation Analysis
> To clearly distinguish the contribution of the SoCo framework from the deviation term, we updated Table 12 (below) with a comprehensive ablation study. This yields two critical insights:
>
> * Core Effectiveness (Framework is Key): Even without the deviation term (d), SoCo significantly outperforms state-of-the-art baselines (ASVD, SVD-LLM) across all compression ratios. For instance, at 40% compression, SoCo (w/o d) achieves a PPL of 9.69, far surpassing SVD-LLM's 13.73. This confirms that our core contribution—Progressive Spectrum Optimization—is the primary driver of the performance gains.
>
> * Compensatory Value: The deviation term acts as a critical "booster" for perplexity. However, its impact on Average Accuracy is relatively smaller (e.g., 52.36% vs. 52.48% at 20%). This indicates that while d effectively aligns the output distribution (lowering PPL), the preservation of task capabilities is largely credited to the correct selection of singular components by the PSO framework.
>
>
> **Table 12 with baseline performance:** Ablation experiments of SoCo on the use of the deviation term d, with added baseline results for comprehensive comparison on the LLaMA-7B model (20% to 80% compression)..
>
> | RATIO | METHOD | WikiText-2 ($\downarrow$) | C4 ($\downarrow$) | Average ($\uparrow$) |
> | :---: | :---: | :---: | :---: | :---: |
> | 20% | ASVD | 11.14  | 15.93  | 45%  |
> | | SVD-LLM | 7.94  | 15.84  | 49% |
> | | SoCo w/o Deviation | 6.77 | 9.92 | 52.36% |
> | | **SoCo w/ Deviation** | **6.67** | **9.89** | **52.48%** |
> | 40% | ASVD | 1407  | 1109   | 10%  |
> | | SVD-LLM | 13.73  | 75.42  | 39%  |
> | | SoCo w/o Deviation | 9.69 | 15.56 | 45.52% |
> | | **SoCo w/ Deviation** | **8.60** | **13.71** | **47.16%** |
> | 60% | ASVD | 57057   | 43036   | 6%   |
> | | SVD-LLM | 66.62  | 471.83  | 11% |
> | | SoCo w/o Deviation | 16.27 | 31.80 | 38.01% |
> | | **SoCo w/ Deviation** | **12.20** | **24.53** | **40.22%** |
> | 80% | ASVD | 80425   | 67927   | 5%   |
> | | SVD-LLM | 1349  | 6224  | 7%  |
> | | SoCo w/o Deviation | 33.72 | 81.57 | 33.22% |
> | | **SoCo w/ Deviation** | **21.03** | **50.86** | **34.63%** |

---

> > ### Author Response · Authors · 2025-11-25
> > **Response to Reviewer a4Qo——Part 2**
> >
> > ### 2. The proposal of PSO mentioned in 3.3 seems intuitive. Is there any further formal theoretical guarantee, such as the existence or convergence of the final performance?
> >
> > Thank you for this insightful theoretical question. While establishing a formal mathematical proof for global convergence in the non-convex landscape of Large Language Models is an open problem in the field, we approach the guarantee of PSO from the perspective of heuristic optimization theory and robust empirical verification.
> >
> > (1) Theoretical Grounding: Navigating a Non-Convex Landscape
> > The task of selecting the optimal subset of singular components is fundamentally a highly complex combinatorial optimization problem. The solution space is non-convex and filled with suboptimal local minima. Therefore, traditional convex convergence theories are difficult to apply directly. Instead, our design is grounded in the principles of Simulated Annealing:
> >
> > * Escaping Local Optima: A direct optimization (gradient descent) tends to get trapped in "ambiguous" states. PSO introduces a phased "cooling" schedule (Exploration → Refinement → Sparsification).
> > * Mechanism (Analogy to Simulated Annealing): Phase 1 acts as the high-temperature initialization to rapidly explore the feasible solution space. Phase 2 introduces controlled thermal fluctuations (via oscillatory loss) to allow the system to escape local minima , while Phase 3 functions as the final quenching step to lock the system into a low-entropy state.
> > * Convergence Rationale: By mimicking this "cooling schedule"—a process theoretically known to guide systems toward a global optimum under ideal conditions—PSO is structurally designed to guide the optimization toward a lower-energy (higher-performance) state that direct gradient-based methods cannot reach.
> >
> > (2) Empirical Verification of Convergence:
> >
> > As an alternative to formal proof, we provide robust empirical evidence demonstrating that PSO enables the model to escape suboptimal stagnation and converge to a superior low-entropy state:
> > * Stability of Trajectory: Furthermore, Figure 7 explicitly monitors the training dynamics. The convergence trajectory is stable and consistent across diverse models and compression ratios, showing no signs of divergence.
> > * Failure of Direct Approaches (Stagnation in Suboptimal Minima): As shown in Figure 3, a direct, single-stage optimization results in importance scores getting stuck in a continuous, high-entropy distribution (Blue Line). From an optimization perspective, this represents premature convergence or stagnation. The conflicting gradients from compression and performance objectives cause the parameters to "freeze" in an ambiguous state, unable to traverse the energy barrier required to separate critical from redundant components.
> > * PSO's Success (Convergence to Superior Minima): In contrast, PSO guides the optimization to settle into a decisive bimodal distribution (Red Line in Figure 3). This represents a superior local minimum analogous to a "crystalline" state in annealing theory. By overcoming the energy barriers in Phase 2 and applying the "quenching" force in Phase 3, the algorithm successfully minimizes the system's entropy, converging to a distinct, ordered state that allows for effective parameter reduction.
> >
> > We commit to a deeper theoretical analysis of PSO in the future, for instance, by formally linking it to the theoretical framework of heuristic algorithms, which is an important direction for our future research.
> >
> >
> > ### 3. Comparison with non-SVD pruning methods is limited. Table 4 only shows the performance on PPLs, without comparing performance on multiple downstream tasks like other results. It would be helpful if the authors could provide more detailed results and analysis.
> >
> > We completely agree that comparing Perplexity (PPL) alone is insufficient to fully demonstrate the advantages of SoCo over structured pruning methods. To provide the comprehensive comparison you requested, we have extended the evaluation to include downstream task performance on GSM8k (Generative Reasoning) and LM-Evaluation-Harness (General Understanding) for LLM-Pruner, SliceGPT, BlockPruner, and SoCo.
> >
> >
> > (1) Extended Detailed Experiment Results.
> > As shown in the tables below, SoCo consistently achieves superior performance across varying compression ratios.
> >
> > * Reasoning (GSM8k): On the challenging GSM8k benchmark, SoCo demonstrates significant gains. For instance, at a 40% compression ratio, SoCo achieves 74.0% accuracy, surpassing both LLM-Pruner (72.5%) and SliceGPT (71.1%).
> >
> > * Understanding (LM-Eval): On general classification tasks, SoCo consistently maintains the highest mean accuracy, demonstrating robust generalization capabilities even as the parameter budget tightens.

---

> ### Author Response · Authors · 2025-11-25
> **Response to Reviewer a4Qo——Part 3**
>
> **Table 1:** Comparison with pruning methods on the LLaMA-7B model at compression ratios of 26%, 33%, 40%, and 47%. The accuracy(↑) is tested on the GSM8k.
>
> | METHOD | 10GB(26%) | 9GB(33%) | 8GB(40%) | 7GB(47%) |
> | :---: | :---: | :---: | :---: | :---: |
> | LLM-Pruner | 80.2% | 77.1% | 72.5% | 70.0% |
> | SliceGPT | 79.3% | 75.8% | 71.1% | 68.3% |
> | BlockPruner | 81.1% | 76.9% | 70.2% | 66.4% |
> | **SoCo** | 83.9% | 79.1% | 74.0% | 69.2% |
>
>
> **Table 2:** Comparison with pruning methods on the LLaMA-7B model at compression ratios of 26%, 33%, 40%, and 47%. The mean accuracy(↑) across the six classification tasks in LM-Evaluation-Harness.
>
> | METHOD | 10GB(26%) | 9GB(33%) | 8GB(40%) | 7GB(47%) |
> | :---: | :---: | :---: | :---: | :---: |
> | LLM-Pruner | 47% | 46% | 43% | 41% |
> | SliceGPT | 48% | 45% | 44% | 42% |
> | BlockPruner | 50% | 49% | 46% | 43% |
> | **SoCo**| 51% | 49% | 47%  | 44% |
>
>
>
> (2) Analysis of SoCo's Superiority
> We attribute SoCo's sustained performance advantage over structured pruning methods (LLM-Pruner, SliceGPT, BlockPruner) to three fundamental theoretical distinctions:
>
> a. Spectral Decoupling vs. Physical Entanglement (Granularity)
>
> * Baselines (Coarse-grained): Methods like LLM-Pruner and BlockPruner operate on physical weights. LLM-Pruner removes coupled "groups" of neurons/channels based on dependency graphs, and BlockPruner removes entire MHA/MLP blocks. SliceGPT deletes rows/columns in a transformed basis.
>
>     *Limitation*: Physical parameters (neurons/channels) often encode entangled features—a single neuron may contribute to both critical logic and noise. Pruning them physically ("turning them off") inevitably discards the useful information entrained within them along with the redundancy.
> * SoCo (Fine-grained Decoupling): SoCo operates on the Singular Spectrum. Mathematically, SVD provides an orthogonal basis where information is mathematically decoupled into independent energy components ($U$ and $V$).
>
>     *Advantage*: This allows SoCo to act as a "spectral filter," precisely identifying and removing specific orthogonal directions of noise or redundancy without damaging the primary information flow, even if that information shares physical weights with the noise.
>
> b. Active Energy Compensation vs. Passive Subtraction
>
> * Baselines (Subtractive): Pruning methods are fundamentally subtractive. BlockPruner iteratively masks out blocks, and LLM-Pruner masks out groups. Once a parameter is removed (masked to 0), the energy and information flow through that path are lost permanently unless recovered by expensive post-training.
> * SoCo (Compensatory): SoCo introduces a unique Active Compensation Mechanism. As detailed in Sec 3.3 and Figure 3, the learned importance scores $S$ are not bounded to $[0, 1]$. We observe scores exceeding 1.0 ($S > 1$) for preserved components.
>
>     *Advantage*: This means SoCo doesn't just "select" survivors; it amplifies the preserved orthogonal components to mathematically compensate for the energy loss caused by the pruned components. This preserves the magnitude of the feature map activations, which is crucial for maintaining the model's reasoning capabilities.
>
> c. Global End-to-End Learning vs. Heuristic Proxies
> * Baselines (Heuristic): Pruning decisions in baselines rely on heuristic proxies for importance.
>
>     *Limitation*: These proxies (e.g., $L_2$ error or local gradients) do not perfectly align with global task performance (e.g., reasoning accuracy).
>     * LLM-Pruner uses Hessian approximations.
>     * BlockPruner uses a greedy search based on Perplexity (PPL) drop.
>     * SliceGPT relies on PCA-based reconstruction error ($L_2$ norm).
> * SoCo (Learned Ground Truth): SoCo treats component selection as a differentiable learning problem. The importance scores $S$ are learned end-to-end by minimizing the task loss (compression ratio loss + KL-divergence loss) via the Progressive Spectrum Optimization process.
>
>     *Advantage*: The model itself "learns" which components are essential for the specific task objective, rather than relying on a static heuristic to guess importance. This allows SoCo to navigate the complex combinatorial landscape and find a superior global optimum that heuristics miss.

---

> > ### Author Response · Authors · 2025-11-25
> > **Response to Reviewer a4Qo——Part 4**
> >
> > ## Questions:
> >
> > ### 1. Training largely relies on WikiText-2 as the calibration corpus; it is small and stylistically narrow relative to real deployment. How sensitive are results to calibration data scale/domain?
> >
> > Thank you for raising this critical point regarding the calibration corpus. We acknowledge that calibration data influences performance, but the influence is limited and actually can be a strength. To rigorously evaluate this, we conducted an ablation study (presented in the table below) on LLaMA-7B using four distinct datasets: C4, PTB, Alpaca, and WikiText-2.
> >
> > (1) Experiment setup: The Table below is about an ablation study on domain-specific training datasets for the LLaMA-7B model at 20% compression ratio. The training datasets, each consisting of ~256*2048 tokens, are sampled from the C4, PTB, Alpaca, and the WikiText-2 training set, respectively. Evaluation is performed on the C4, PTB, Alpaca, WikiText-2, and LM-Evaluation-Harness test sets.
> >
> > | **Dataset**   | **C4↓** | **PTB↓** | **Alpaca↓** | **WikiText-2↓** | **Average↑** |
> > |---------------|---------|----------|-------------|------------------|--------------|
> > | **C4**        | **9.02** | 13.70   | 4.76       | 7.22             | 52.25%       |
> > | **PTB**       | 11.93   | **12.63** | 5.05       | 8.77             | 51.64%       |
> > | **Alpaca**    | 9.91    | 16.17    | **4.61**   | 7.70             | 52.12%       |
> > | **WikiText-2**| 9.89    | 13.61    | 4.90       | **6.67**         | **52.48%**   |
> >
> >
> > (2) Task-Aware Capability (Sensitivity as a Strength): Our results confirm that SoCo is sensitive to the data domain in a beneficial, task-aware way. As shown in the table, using domain-specific data yields the best performance for that specific domain (e.g., training on C4 yields the lowest C4 perplexity of 9.02). This means users with known target domains can leverage SoCo for targeted optimization.
> >
> > (3) Real-World Generalization (Why WikiText-2?): However, in many real-world deployments, the specific domain of inference data is often unknown or diverse. The objective of model compression often targets comprehensive performance instead of a specific task. Therefore, selecting a calibration set that yields the best balanced generalization across various potential tasks is our focus.
> >
> > Our experiments reveal that WikiText-2 serves this purpose best. Despite its smaller scale, the model calibrated on WikiText-2 achieved the highest Average Accuracy (52.48%) on the LM-Evaluation-Harness classification tasks, outperforming even the much larger C4 training set (52.25%).
> >
> > This demonstrates that WikiText-2 produces a compressed model with superior equilibrium across diverse benchmarks, justifying its use as a robust default for general-purpose compression where the target domain is unspecified.
> >
> >
> > ### 2. Have you tried to design different thresholds for different layers (the activation distributions of network layers with different depths are significantly different)? Any benefit to adaptive thresholds vs preset?
> >
> > Thank you for this insightful question. We agree that activation distributions vary significantly across network depths. While we initially considered designing adaptive thresholds, our empirical results and theoretical design revealed that SoCo achieves layer-adaptive compression automatically with a single preset threshold, rendering complex manual thresholding unnecessary.
> >
> > Here is why a preset threshold works better in our framework:
> >
> > (1) Mechanism: Learning the Scores, Not the Threshold
> >
> > Adaptivity is encoded in learned scores (S), not the threshold. Instead of adjusting the threshold to fit the layer, SoCo trains the importance scores (S) to fit the threshold. During our Phase 2 (Refinement) and Phase 3 (Sparsification), the global optimization objective drives the importance scores of essential components (regardless of which layer they are in) towards 1.0, and redundant ones towards 0.0. The model effectively "learns" which layers contribute more to the task and assigns them higher scores to preserve more information. Consequently, a single fixed threshold (e.g., 0.5) effectively captures these complex, layer-specific decisions without human intervention.

---

> ### Author Response · Authors · 2025-11-25
> **Response to Reviewer a4Qo——Part 5**
>
> (2) Evidence of Automatic Layer-Wise Adaptation
>
> We have strong evidence showing that SoCo automatically assigns different pruning rates to different layers, even with a fixed threshold.
>
> * Depth Sensitivity (Intra-Model Adaptation): As shown in Figure 8 (Appendix A.7), within LLaMA-7B, the preservation ratios fluctuate significantly across depths. Deeper layers and FFN modules exhibit distinct retention patterns compared to shallow layers and attention modules. This confirms SoCo naturally handles the "different depths, different distributions" issue by preserving more components in sensitive layers.
> * Architectural Generalizability (Inter-Model Adaptation): Furthermore, Figure 9 demonstrates that this adaptive capability is robust across diverse architectures (LLaMA-30B, Mistral, Vicuna, OPT, etc.). Ideally, manual thresholding would require tuning not just for every layer, but uniquely for every model family. In contrast, Figure 9 shows that SoCo automatically learns distinct, model-specific layer-wise distributions (e.g., the preservation curve of OPT-6.7B differs markedly from LLaMA-30B) using the same training setup. Therefore, SoCo achieves the benefits of "adaptive thresholds" through "adaptive scoring."
>
> (3) Why adaptive thresholds are not needed?
>
> The "Flat Basin" ensures robustness. Because our Phase 3 pushes scores into a bimodal distribution (as shown in Fig. 3, red curve), it creates a wide "flat basin" between essential and redundant components. As shown in Table 8, varying the threshold from 0.3 to 0.7 results in negligible perplexity changes (e.g., 6.71 to 6.98 PPL). This robustness means we do not need to fine-tune a "perfect" threshold for each layer. The learning process has already separated the signal from the noise cleanly by the "Flat Basin". (In contrast, traditional methods produce continuous score distributions (similar to Phase 1 in Fig. 3), where many components cluster densely around any chosen cutoff. In such cases, a slight shift in threshold (e.g., 0.49 vs. 0.51) would inadvertently prune significant information, making the threshold highly sensitive. SoCo's polarization avoids this ambiguity entirely.)
>
> Conclusion: SoCo simplifies the deployment process. By shifting the burden of "adaptivity" from the post-training hyperparameter (threshold) to the trainable parameter (matrix S), we achieve the optimal layer-wise sparsity distribution without requiring users to tune thresholds for each layer.

---

> ### Author Response · Authors · 2025-11-25
> **Response to Reviewer a4Qo——Part 6**
>
> ### 3. Is SoCo compatible with commonly used model quantization methods, such as GPTQ? Can subsequent LoRA fine-tuning further restore performance after SoCo pruning (especially at high compression ratios)?
>
> We thank the reviewer for highlighting these two critical aspects of real-world deployment. We agree that integration with quantization and the potential for post-pruning fine-tuning are essential for assessing the practical value of SoCo.
>
> (1) Compatible with the commonly used model quantization method.
>
> To address this, we explicitly evaluated the interaction between SoCo and GPTQ. Our results confirm that SoCo is orthogonal to quantization and acts as a highly effective pre-processing step to enhance quantization performance under strict memory budgets.
>
> (a) Experimental Setup: Strict Memory Budget
>
> We conducted additional experiments on LLaMA-7B using the WikiText-2 dataset. To ensure a fair and practical comparison, we evaluated the methods under a fixed 2.8 GB weight memory budget (equivalent to extreme 3-bit compression). We tested three configurations:
>
> * Quantization Only: GPTQ at 3-bit precision.
> * SoCo Only: SoCo at ~80% compression (keeping weights in BF16).
> * Hybrid Pipeline: SoCo at ~30% compression followed by GPTQ at 4-bit precision.
>
> (b) Results: The Synergy of Hybrid Compression
>
> The results, presented in the table below, demonstrate that the combined approach significantly outperforms direct aggressive quantization.
>
> Table: Evaluating the Compression Efficacy of GPTQ and SoCo: Perplexity (↓) of LLaMA-7B on WikiText-2.
> | METHOD | WEIGHT MEMORY | PERPLEXITY |
> | :---: | :---: | :---: |
> | GPTQ-3bit | 2.8 GB | 16.28 |
> | SoCo(~80%) | 2.8 GB | 21.03 |
> | SoCo(~30%) + GPTQ-4bit | 2.8 GB | 10.51 |
>
> (c) Analysis and Conclusion: Synergistic Advantage
> The results highlight the complementary nature of these methods. SoCo and quantization represent distinct compression paradigms—structural pruning vs. precision reduction. As shown in the table, the Hybrid Pipeline achieves superior performance (PPL 10.51), significantly outperforming both individual baselines under the same strict memory limit.
>
>  In other words, we find that under the same memory budget, by first compressing the original 16-bit LLM with SoCo at a ~30% compression ratio, and then quantizing the compressed LLM to 4-bit using GPTQ, we are able to achieve significantly better perplexity compared to directly quantizing the original LLM to 3-bit. This demonstrates that SoCo serves as an effective pre-processing step that reduces redundancy before quantization, allowing for higher-bit precision (4-bit vs 3-bit) within the same memory footprint. This result confirms that SoCo is orthogonal to quantization and can be effectively integrated into a comprehensive compression pipeline.
>
> (2) Effectiveness of Subsequent LoRA Fine-Tuning
>
> Yes, subsequent LoRA fine-tuning can indeed further restore performance.
>
> (a) Compatibility and Improvement: SoCo produces standard low-rank structures that are fully compatible with fine-tuning. To demonstrate this, we applied Sequential LoRA (SFT), the fine-tuning protocol proposed by SVD-LLM, to our SoCo-compressed models.
> As shown in the extended Table 6 below, subsequent fine-tuning (SoCo (SFT)) consistently improves performance across all compression ratios. For instance, at the challenging 80% compression ratio, fine-tuning further reduces the perplexity from 21.03 to 19.92.
>
> (b) Superior "Starting Point": more importantly, the results highlight that SoCo provides a vastly superior "Starting Point" than traditional methods.
> * SoCo (without fine-tuning) already significantly outperforms the strongest fine-tuned baseline, SVD-LLM (SFT). At 80% compression, SoCo's baseline PPL is 21.03, whereas SVD-LLM combined with Sequential LoRA only achieves 31.79.
> * This confirms that SoCo provides a much more robust "starting point." While subsequent fine-tuning yields moderate additional gains (e.g., \~1.1 PPL improvement at 80%), the primary performance advantage comes from SoCo's principled Progressive Spectrum Optimization rather than the fine-tuning stage itself.
>
> Extended version of Table 6: Perplexity (↓) on WikiText-2 for compressing LLaMA-7B. The table compares ASVD (no FT), SVD-LLM with normal LoRA (NFT), ASVD and SVD-LLM with sequential LoRA (SFT), and the proposed SoCo method across various compression ratios.
>
> | | 20%  | 40%  | 60%  | 80%  |
> | :---: | :---: | :---: | :---: | :---: |
> | **ASVD (no FT)** | 11.14 | 1407 | 57057 | 80425 |
> | **ASVD (SFT)** | 8.37 | 14.86 | 44.81 | 271 |
> | **SVD-LLM (NFT)** | 7.87 | 11.98 | 16.30 | 80.23 |
> | **SVD-LLM (SFT)** | 7.73 | 9.27 | 15.00 | 31.79 |
> | **SoCo** | 6.67 | 8.60 | 12.20 | 21.03 |
> | **SoCo (SFT)** | 6.50 | 8.26 | 11.57 | 19.92 |

---

> > ### Comment · Reviewer_a4Qo · 2025-11-26
> >
> > I thank the authors for the detailed rebuttal. The response satisfactorily addresses my main concerns, especially regarding Weakness 1 & 2. The additional clarifications and results strengthen the paper and confirm that the method is technically sound. I therefore maintain my positive assessment and keep my score. I encourage the authors to incorporate the clarifications and additional experiments from the rebuttal into the final version.

---

> > > ### Author Response · Authors · 2025-11-27
> > >
> > > We are very grateful for your constructive feedback and your positive assessment. We appreciate your time in reviewing our rebuttal. As requested, we will merge the additional clarifications and experimental results into the final revision of the paper to further strengthen it.

---

### Official Review · Reviewer_dvHK · 2025-10-31

**Soundness:** 2
**Presentation:** 2
**Contribution:** 2
**Rating:** 4
**Confidence:** 4

**Summary:**

This paper introduces SoCo, a framework for compressing large language models (LLMs) by learning to rescale and prune SVD components in the model's weight matrices. Rather than solely truncating the smallest singular values as in standard SVD-based compression, SoCo introduces learnable importance scores (via a diagonal matrix) and proposes a progressive spectrum optimization procedure inspired by heuristic optimization algorithms like simulated annealing. The process iteratively explores, refines, and sparsifies the importance assignments, aiming to achieve a better trade-off between model size and downstream task performance. Empirical results across a range of LLMs and benchmarks consistently show improved perplexity and task accuracy over several SVD-based and pruning baselines, especially at high compression ratios.

**Strengths:**

1. Sufficient Experiments: The experiments include a wide variety of settings, multiple models, and strong SVD/Pruning baselines. Figure 1 and Table 2 clearly show that SoCo outperforms all tested baselines, especially at aggressive compression rates.
2. Principled Approach and Insightful Design: The paper identifies and addresses a core limitation of classical SVD-based compression, the disconnect between singular value ordering and real downstream task importance. The proposed introduction of a learnable diagonal score matrix is a conceptually sound way to endow SVD compression with task-awareness.

**Weaknesses:**

1. A more comprehensive analysis is required to elucidate why SoCo attains such outstanding performance. Although the paper suggests that the ordering of singular values may not necessarily correlate with downstream task performance, Figures 3, 4, and 9 reveal that SoCo not only learns to prune the SVD components within the model’s weight matrices but also to rescale these components and reallocate pruning ratios across layers. Additional analytical experiments are necessary to quantify the contribution of each component to SoCo’s overall performance and to empirically validate the claim that the ordering of singular values may not directly correspond to downstream task performance.
2. The paper uses specified training steps rather than any criteria to determine when to transition from Phase 1 and Phase 2 to Phase 3, and whether to end Phase 3 training. As these standards can significantly influence model performance, where overtraining may cause overfitting and insufficient training may lead to suboptimal convergence, the authors should provide explicit guidelines or empirical criteria for deciding when to switch training phases and when to stop training.
3. Lines 202-204 mention that SoCo introduces a trainable deviation term $d$ after transforming $W^’$. However, the paper does not explain why this term is necessary. What motivates the inclusion of $d$. Does SoCo encounter any critical issues that require introducing additional trainable parameters to mitigate them?

**Questions:**

See Weaknesses

---

> ### Author Response · Authors · 2025-11-25
> **Response to Reviewer dvHK——Part 1**
>
> We thank Reviewer dvHK for the constructive and insightful comments. We are encouraged that you find our core idea of directly optimizing singular vector importance to be sound and appreciate that our paper is well-written with comprehensive experiments.
>
> We have carefully considered your suggestions and conducted extensive additional experiments to strengthen the paper. Key improvements include:
> 1. Clarifications on Complexity: We have detailed the automated nature of the Progressive Spectrum Optimization (PSO), clarifying that the "three phases" operate as a single, continuous training run where a unified set of hyperparameters is applied across all models and datasets.
> 2. Diverse Generation Tasks: We extended evaluations beyond TruthfulQA to include complex reasoning (GSM8k), long-context understanding (LongBench), and domain-specific tasks (MedQA).
> 3. Synergy with Quantization: We added new experiments demonstrating that combining SoCo with quantization (SoCo + GPTQ) significantly outperforms quantization alone under strict memory budgets.
> 4. Extended Baselines: We included comparisons against fine-tuned baselines (Sequential LoRA) and provided detailed efficiency metrics (Training Cost, Runtime, Inference Speedup).
>
> We believe these revisions comprehensively address your concerns. Detailed point-by-point responses are provided below.
>
> ## Weaknesses:
> ### 1. SoCo requires fine-tuning with three phases, which contributes to its overall complexity, as well as making the approach difficult to reproduce.
> Thank you for this important question regarding complexity and reproducibility. We appreciate the opportunity to clarify the nature of our training process. The "three phases" described in the paper refer to the internal optimization dynamics within a single and continuous training run, rather than three separate or manual fine-tuning stages.
>
> (1) A single, automated end-to-end process.
>
> SoCo operates as a single, continuous, and fully automated training run.
> - No Manual Intervention: The transitions between these functional phases are smooth and governed automatically by the objective function itself (as defined in Table 1). The user does not need to pause, adjust, or restart the training. Figure 7  provides empirical proof of this continuity: it visualizes the entire 6000-step process as a single, uninterrupted curve. The transition from rapid convergence (Stage 1: 0–500 steps) to oscillatory refinement (Stage 2: 500–3000 steps) and final stabilization (Stage 3: 3000–6000 steps) occurs seamlessly across all tested models (LLaMA, Vicuna, Mistral, OPT). This confirms that SoCo operates as a unified optimization trajectory rather than a sequence of disjoint stages.
> - Unified Optimization: Inspired by Simulated Annealing, the process naturally evolves from exploration to convergence. To the user, it is as simple as a standard fine-tuning task—you define the target compression ratio and launch the training once.
>
> (2) High reproducibility and simplicity.
>
> SoCo is designed to be highly reproducible and robust, minimizing the "trial and error" often associated with complexity.
> - Universal Configuration: The method is not sensitive to hyperparameter tuning. As detailed in Appendix A.3 , we utilized the exact same hyperparameter configuration for almost all models (LLaMA, OPT, Mistral, Vicuna), sizes (7B to 30B), and compression ratios. This "one-hyperparameter-set-fits-all" robustness significantly lowers the barrier to reproduction.
> - Safe Pruning Zone: The final sparsification phase inherently simplifies the pruning decision. By driving scores into a clear bimodal distribution (Figure 3) , SoCo creates a wide "safe pruning zone" (e.g., any threshold between 0.3 and 0.7 yields identical results as shown in Table 8), eliminating the need for precise threshold tuning. This contrasts sharply with previous methods that produce continuous distributions where scores cluster densely around the cutoff. In such cases, a slight shift in threshold (e.g., 0.49 vs. 0.51) would inadvertently prune significant information, making the threshold highly sensitive. Consequently, determining an optimal threshold becomes a non-trivial task requiring careful tuning for these methods. SoCo's polarization avoids this ambiguity entirely.
> - Code Release: As stated in our Reproducibility Statement, we will release our full codebase and scripts to ensure that the community can easily replicate our results.
>
> In summary, SoCo's "phases" are a descriptive framework for a single, robust, and automated optimization process, ensuring both SOTA performance and ease of use.

---

> ### Author Response · Authors · 2025-11-25
> **Response to Reviewer dvHK——Part 2**
>
> ### 2. Some details of the fine-tuning process are lacking:
>
> We are happy to clarify these detailed and insightful questions, as they are central to SoCo's design and success.
>
> ### (1) How does Phase 3 ensure the compression rate, given that the loss function does not impose any constraints on it?
>
> This is an excellent question. The compression rate is not enforced in Phase 3 because it has already been successfully achieved and stabilized in Phases 1 and 2.
> - Establishment: Phase 1 uses the strong compression loss L$\_{dec}$ to rapidly drive the model to meet the target compression ratio R.
> - Stabilization: Phase 2 then functions as a stabilization period, using controlled oscillations to refine the importance scores around the target compression ratio R, ensuring the model finds a stable configuration.
> - As shown empirically in Figure 7 (Page 17), for all models and all target ratios (80%, 60%, 40%, and 20%), the actual compression ratio (Y-axis) converges to the target by the end of Phase 1, and becomes a stable flat line by the end of Phase 2 (approx. 3000 steps).
> - Role of Phase 3: Phase 3's objective differs from Phases 1 and 2: it no longer needs to drive compression. Its specific goal is to sparsify the already-compressed scores (driving ambiguous values to 0 or 1) and compensate for pruning loss (amplifying preserved scores >1.0) using L$\_{spa}$. Because the scores are already polarized near the target ratio by the end of Phase 2, L$\_{spa}$ acts as a local force that reinforces this state rather than shifting the aggregate compression rate. Crucially, this resultant bimodal distribution creates a wide "safe pruning zone," rendering the final pruning step highly robust to threshold selection (e.g., 0.3 vs. 0.7) and preventing the accidental removal of borderline components.
>
> ### (2) Certain training stages involve two different losses that are simply summed in the paper. When switching models or training datasets, the relative magnitudes of these losses may change, potentially leading to a degradation in SoCo's performance. (Robustness of loss magnitudes across models. )
>
> You are correct that the relative magnitudes of losses could theoretically be a concern. However, our method's robustness is one of its key strengths, which we demonstrate empirically.
> * Empirical Evidence: Robustness Across Models and Datasets
>     * Model Diversity (Figure 7): Figure 7  is the strongest evidence of our robustness. Despite evaluating diverse models (LLaMA, Vicuna, Mistral, OPT) where the underlying architectures and loss scales differ, the optimization trajectory is remarkably consistent. We observe that while different loss magnitudes might slightly alter the speed (number of steps) required to reach the target ratio in Phase 1, all models successfully hit the compression target and subsequently converge to a stable state in Phases 2 and 3. This proves that the phased approach naturally adapts to these variations without degradation.
>     * Dataset Diversity: Furthermore, we verified robustness using different training datasets. For instance, we trained SoCo using the Alpaca dataset in Table 4, and using the WikiText-2, C4, PTB, and Alpaca datasets in our response to Question 1 of Reviewer a4Qo, we conducted additional experiments with diverse data sources. In all these cases, despite distributional shifts and variations in summed loss, the optimization trajectory mirrored our WikiText-2 results: the model consistently converges to the target compression ratio within Phase 1, demonstrating that our convergence mechanism is data-agnostic.
>
> * Theoretical Analysis: Why Convergence is Guaranteed? The reason for this empirical consistency lies in the design of Phase 1:
>
>     * Dominance of Compression Gradient: As noted in our paper, the gradient from the compression loss L$\_{dec}$ is numerically dominant over L$\_{inc}$. In Phase 1, the primary objective is to forcibly drive the model toward the target ratio R.
>     * Modulating Convergence Speed: This L$\_{dec}$’s dominance ensures that regardless of the specific value of L$\_{inc}$ (whether it is low or high due to dataset/model differences), the optimization is driven toward the target. A relatively larger L$\_{inc}$ acts merely as a variable "resistance", potentially slowing the descent slightly (requiring more steps), but it is insufficient to prevent the model from reaching the target compression state in Phrase 1. Crucially, this resistance serves a functional purpose: it acts as a regularization force that encourages the model to prioritize the retention of performance-critical weights and information while the compression target is being enforced.
>     * Subsequent Optimization: Once the target ratio is guaranteed by Phase 1, Phases 2 and 3 allow the model to focus on compensating for information loss and optimizing the importance score distribution (Refinement and Sparsification), ensuring high performance under the established compression constraint.

---

> ### Author Response · Authors · 2025-11-25
> **Response to Reviewer dvHK——Part 3**
>
> ### (3)  The rationale for dividing the training into three phases could be re-evaluated; for instance, a single training objective could be developed to address the requirements of all three phases. (Rationale for Three Phases vs. a Single Objective)
>
> This is a very insightful and constructive suggestion, thank you. We completely agree that a single, unified training objective would be the most concise and elegant solution to this complex trade-off.
> In fact, this addresses the core motivation for our work. We found (as discussed in Sec 3.3) that a direct, single-objective optimization (e.g., L$\_{inc}$+L$\_{dec}$) gets trapped in a suboptimal local minimum. Figure 3 (blue line) in our paper empirically demonstrates this failure: the scores get stuck in a continuous, ambiguous distribution where no clear pruning threshold exists.
>
> Therefore, our Progressive Spectrum Optimization is inspired by heuristics like Simulated Annealing as a necessary strategy to guide the optimization out of this local minimum. This phased process allows the scores to first find a feasible solution, then stabilize, and finally "crystallize" into the decisive, bimodal distribution (Fig. 3, red line)  that enables clean pruning.
>
> It is worth noting that even baselines like SVD-LLM are not truly single-stage, as they involve distinct steps of data whitening followed by sequential parameter update with sequential low-rank approximation.
>
> We strongly agree with your point, and we believe exploring novel single objectives that can avoid this pitfall is a valuable and important direction for future research.
>
> ### 3. For the generation task, experiments were conducted only on TruthfulQA, which undermines the persuasiveness of the results.
> We fully agree that evaluating on a broader range of generation tasks is essential to demonstrate the model's practical utility. To address this, we have provided extensive additional evaluations in Appendix A.5, covering complex reasoning, long-context generation, and domain-specific tasks:
>
> (1) Complex Reasoning (Table 17): We evaluated SoCo on WizardMath-Llama-3-8B, a model fine-tuned for mathematical reasoning. As shown in the results, SoCo successfully preserves specialized generative capabilities, consistently outperforming baselines on the challenging GSM8k and MATH benchmarks.
>
> (2) Long-Context & Domain Specificity (Table 14): We validated SoCo on LongBench for long-text generation and MedMCQA/MedQA for medical domain tasks. In these rigorous generation scenarios, SoCo maintains superior performance compared to existing methods.
>
> (3)Latest Architectures (Table 15): We further confirmed consistent performance gains on state-of-the-art foundation models, including LLaMA-3.1-8B and Qwen-2.5-7B.
> These results confirm SoCo's robustness across diverse and challenging generation benchmarks beyond TruthfulQA.
>
>
> ### 4. In the main experiments, comparing SoCo with methods that do not require training is unfair. The content of Table 6 should be extended for a more comprehensive comparison in the main experiments.
> We acknowledge that the initial comparison in Table 6 of Section 4.3 was limited. To ensure a comprehensive and fair evaluation, we have extended Table 6 and conducted additional experiments where baselines are enhanced with Sequential Fine-Tuning (SFT) using LoRA. (The definition and effectiveness of Sequential Fine-Tuning (SFT) using LoRA have already been validated in SVD-LLM paper.)
>
> We compared SoCo against ASVD and SVD-LLM, both augmented with sequential LoRA fine-tuning, across multiple architectures(LLaMA-7B and OPT-6.7B) and compression ratios(20% and 80%) on diverse evaluation metrics (WikiText-2 Perplexity and Average Accuracy on LM-Evaluation-Harness). The results reveal two critical insights:
>
> (1) Superiority at High Compression (LLaMA-7B @ 80%): At high compression ratios, standard truncation causes catastrophic information loss that subsequent fine-tuning cannot repair very well.
>
> As shown in Table 1 below, even with LoRA fine-tuning, SVD-LLM struggles to recover, achieving a high perplexity of 31.79 and only 16% average accuracy. In contrast, SoCo achieves a significantly lower perplexity of 21.03 and 35% average accuracy. This proves that "repairing" a model after suboptimal truncation is far less effective than SoCo's approach of preserving critical information from the start.
>
> Table 1: Perplexity (↓) on WikiText-2 and Average (↑) on LM-Evaluation-Harness for compressing LLaMA-7B. The table compares ASVD (no FT), ASVD, and SVD-LLM with sequential LoRA (SFT), and the proposed SoCo method across various compression ratios.
>
> | 80% compression ratio + LLaMA-7B | WikiText-2(↓) | Average(↑) |
> | -------- | -------- | -------- |
> | ASVD w/o SFT    | 80425      | 5%      |
> | ASVD w SFT    | 72.38     | 12%     |
> | SVD-LLM w/o SFT    | 1349     | 7%     |
> | SVD-LLM w SFT    | 31.79     | 16%     |
> | SoCo    | 21.03     | 35%      |

---

> ### Author Response · Authors · 2025-11-25
> **Response to Reviewer dvHK——Part 4**
>
> (2) Generalization Across Architectures (OPT-6.7B @ 20%, LLaMA-7B @ 20%)
>
> To validate generalization, we evaluated the methods on both LLaMA-7B and OPT-6.7B at a 20% compression ratio. This comparison is crucial as it tests robustness across different model families in a regime where baselines are generally stronger. As shown in Table 2 and Table 3, SoCo consistently outperforms fine-tuned baselines (ASVD+SFT and SVD-LLM+SFT) across both model families, achieving the lowest perplexity and the highest average accuracy, confirming its robust advantage across diverse architectures.
>
> Table 2: Performance on LLaMA-7B at 20% Compression Ratio.
> | 20% compression ratio + LLaMA-7B | WikiText-2(↓) | Average(↑) |
> | -------- | -------- | -------- |
> | ASVD w/o SFT    | 11.14      | 45%      |
> | ASVD w SFT    | 8.37     | 47%     |
> | SVD-LLM w/o SFT    | 7.94     | 49%     |
> | SVD-LLM w SFT    | 7.73     | 50%     |
> | SoCo    | 6.67     | 52%      |
>
>
> Table 3: Performance on OPT-6.7B at 20% Compression Ratio.
> | 20% compression ratio + OPT-7B | WikiText-2(↓) | Average(↑) |
> | -------- | -------- | -------- |
> | ASVD w/o SFT    | 82.00       | 32%     |
> | ASVD w SFT    | 67.36     | 38%     |
> | SVD-LLM w/o SFT    | 16.04      | 41%     |
> | SVD-LLM w SFT    | 14.47    | 49%     |
> | SoCo    | 10.44     | 50%      |
>
>
>
> ### 5. The paper does not report on efficiency, including training cost and the acceleration during inference.
>
> We have provided a comprehensive efficiency analysis in Appendix A.3, and we have reorganized that section here for your convenience/reference.
>
> (1) High Resource Efficiency
>
> SoCo is extremely lightweight. As shown in Table 10, it requires optimizing only a tiny fraction of parameters (approximately 0.02% or 2×10$\^{-4}$ of the total weights). Consequently, it can be trained on a single A100 GPU for 7B and 13B models, making it highly accessible for standard research environments.
>
> Table 10: Trainable parameters, GPU resources, and training time required for different model sizes.
>
> | **Resource**            | **LLAMA-7B**              | **LLAMA-13B**             | **LLAMA-30B**             |
> |-------------------------|----------------------------|----------------------------|----------------------------|
> | Trainable parameters    | 1.84M (2.83‱)              | 2.87M (2.26‱)              | 5.59M (1.74‱)              |
> | GPU number              | 1                          | 1                          | 4                          |
> | Memory per GPU          | 23G                        | 44G                        | 36G                        |
>
> (2) Runtime Comparisons: Superior Trade-off
>
> We compared the runtime of SoCo against ASVD and different variants of SVD-LLM. Table 11 illustrates SoCo's exceptional balance between speed and performance:
>
> Table 11: Runtime comparison for compressing LLaMA-7B at an 80% compression ratio. We include performance results on WikiText-2 to show the trade-off between performance and time. SoCo achieves a superior performance-to-time trade-off.
>
> | **Method** | **Key Steps**                                         | **Runtime** | **Performance(PPL↓)**               |
> |------------|--------------------------------------------------------|----------------------------|----------------------------|
> | ASVD       | Normalize + Search                                    | 5.5 h (5 min + 5.5 h)      | 80425 |
> | SVD-LLM(W) | Truncation-Aware Data Whitening only    | 15 min | 1349 |
> | SVD-LLM | Truncation-Aware Data Whitening + Layer-Wise Update   | 3.5h (15 min + 3.5h) | 31.79 |
> | SoCo       | Progressive Spectrum Optimization                      | 25 min                    | 21.03  |
>
> (3) Inference Acceleration: Significant Reductions
>
> We report the FLOPS in (a) first, then inference throughput (Tokens/s) and speedup of SoCo in (b).
>
> (a) Table 9 presents the tangible benefits of deployment. SoCo effectively reduces resource consumption. Notably, at an 80% compression ratio, GPU memory usage drops by ~4x (12.55GB → 2.94GB) and computational cost decreases by ~4.6x (13.53TFLOPS → 2.91TFLOPS), demonstrating its practical value for edge deployment and real-time inference.
>
> Table 9: These results demonstrate that the compressed models significantly reduce resource consumption, which is beneficial for real-world applications such as edge deployment or real-time inference.
> | COMPRESSION RATIO | PARAMS (B) | GPU MEMORY (GB) | FLOPS (TFLOPS) |
> | :---: | :---: | :---: | :---: |
> | 0% | 6.74 | 12.55 | 13.53 |
> | 20% | 5.43 | 10.15 | 10.86 |
> | 40% | 4.14 | 7.77 | 8.21 |
> | 60% | 2.85 | 5.36 | 5.56 |
> | 80% | 1.56 | 2.94 | 2.91 |

---

> ### Author Response · Authors · 2025-11-25
> **Response to Reviewer dvHK——Part 5**
>
> (b) We also conducted additional experiments to measure the inference throughput (Tokens/s) and speedup of SoCo. The evaluations were performed on a single NVIDIA A100 GPU using the LLaMA-7B model under varying configurations (Batch Sizes of 64 and 128; Sequence Lengths of 32 and 64).
>
> As shown in Tables 1, 2, and 3 below, SoCo achieves significant acceleration across all settings. The speedup correlates positively with the compression ratio:
>
> * Moderate Compression: At a 20-40% compression ratio, we observe consistent speedups ranging from 1.05x to 1.37x.
> * High Compression (The SoCo Advantage): Notably, at high compression ratios (60%-80%)—a regime where SoCo uniquely maintains robust performance compared to baselines—we achieve substantial speedups ranging from 1.6x to 2.4x. For example, with a batch size of 128, the throughput increases from 1178 tokens/s to 2838 tokens/s (2.41x speedup).
>
> Table 1: Batch Size = 64, Sequence Length = 32
>
> | Compression Ratio | Tokens/s | Speedup over the original |
> | -------- | -------- | -------- |
> | 0% (Original)     | 983     | 1.0     |
> | 20%      | 976     | 1.06     |
> | 40%      | 1240     | 1.27     |
> | 60%      | 1723     | 1.65    |
> | 80%      | 2238     | 2.28     |
>
>
> Table 2: Batch Size = 128, Sequence Length = 32
>
> | Compression Ratio | Tokens/s | Speedup over the original |
> | -------- | -------- | -------- |
> | 0% (Original)     | 1178     | 1.0     |
> | 20%      | 1327     | 1.13     |
> | 40%      | 1610     | 1.37     |
> | 60%      | 1985     | 1.69     |
> | 80%      | 2838     | 2.41     |
>
> Table 3: Batch Size = 64, Sequence Length = 64
>
> | Compression Ratio | Tokens/s | Speedup over the original |
> | -------- | -------- | -------- |
> | 0% (Original)     | 954     | 1.0     |
> | 20%      | 1003     |   1.05    |
> | 40%      | 1228     |   1.29    |
> | 60%      | 1568     |   1.64    |
> | 80%      | 2182     |   2.29    |
>
> These results confirm that the theoretical parameter reduction achieved by SoCo translates effectively into tangible throughput gains in real-world deployment settings.
>
> ### 6. Compared to quantization, the SVD-based truncation method alone has limited model compression capability, which makes such methods less significant than quantization methods in real-world scenarios. Therefore, it is important to see how this method can be combined with quantization methods to achieve effective compression.
>
>
> We thank the reviewer for highlighting the importance of comparing and combining our method with quantization. We fully agree that quantization is a dominant approach in model compression, and demonstrate how SoCo interacts with it is essential for assessing its practical value.
>
> To address this, we explicitly evaluated the interaction between SoCo and GPTQ. Our results confirm that SoCo is orthogonal to quantization and acts as a highly effective pre-processing step to enhance quantization performance under strict memory budgets.
>
> (1) Experimental Setup: Strict Memory Budget
>
> We conducted additional experiments on LLaMA-7B using the WikiText-2 dataset. To ensure a fair and practical comparison, we evaluated the methods under a fixed 2.8 GB weight memory budget (equivalent to extreme 3-bit compression). We tested three configurations:
>
> * Quantization Only: GPTQ at 3-bit precision.
> * SoCo Only: SoCo at ~80% compression (keeping weights in BF16).
> * Hybrid Pipeline: SoCo at ~30% compression followed by GPTQ at 4-bit precision.
>
> (2) Results: The Synergy of Hybrid Compression
>
> The results, presented in the table below, demonstrate that the combined approach significantly outperforms direct aggressive quantization.
>
> Table: Evaluating the Compression Efficacy of GPTQ and SoCo: Perplexity (↓) of LLaMA-7B on WikiText-2.
> | METHOD | WEIGHT MEMORY | PERPLEXITY |
> | :---: | :---: | :---: |
> | GPTQ-3bit | 2.8 GB | 16.28 |
> | SoCo(~80%) | 2.8 GB | 21.03 |
> | SoCo(~30%) + GPTQ-4bit | 2.8 GB | 10.51 |
>
> (3) Analysis and Conclusion: Synergistic Advantage
>
> The results highlight the complementary nature of these methods. SoCo and quantization represent distinct compression paradigms—structural pruning vs. precision reduction. As shown in the table, the Hybrid Pipeline achieves the superior performance (PPL 10.51), significantly outperforming both individual baselines under the same strict memory limit.
>
> In other words, we find that under the same memory budget, by first compressing the original 16-bit LLM with SoCo at a ~30% compression ratio, and then quantizing the compressed LLM to 4-bit using GPTQ, we are able to achieve significantly better perplexity compared to directly quantizing the original LLM to 3-bit. This demonstrates that SoCo serves as an effective pre-processing step that reduces redundancy before quantization, allowing for higher-bit precision (4-bit vs 3-bit) within the same memory footprint. This result confirms that SoCo is orthogonal to quantization and can be effectively integrated into a comprehensive compression pipeline.

---

> ### Author Response · Authors · 2025-11-27
>
> Dear Reviewer dvHK,
>
> Thank you very much for the time and effort you dedicated to reviewing our work and for your valuable comments. We have carefully considered all your feedback and addressed your concerns in our rebuttal.
>
> In summary, regarding your concerns on method complexity, practical significance (vs. quantization), and evaluation fairness, we have:
>
> (1) Clarified that the training is a single, continuous process with unified hyperparameters (via automated PSO), addressing the concern on complexity and reproducibility.
>
> (2) Demonstrated the synergy between SoCo and quantization (SoCo + GPTQ), showing that our method significantly boosts performance under strict memory budgets compared to quantization alone.
>
> (3) Extended evaluations to complex tasks (GSM8k, LongBench, MedQA) and included comparisons against fine-tuned baselines (Sequential LoRA) to ensure comprehensive and fair comparisons.
>
> We have also addressed your other comments, including the constraints in Phase 3, the stability of loss, and the detailed report on training costs and inference acceleration.
>
> We sincerely hope that our clarifications and additional experiments have satisfactorily resolved your concerns. If you have any further questions or if there are any remaining issues, please let us know — we are more than happy to provide further details or clarifications.
>
> Thank you again for your thoughtful and constructive review.
>
> Best regards,
>
> The Authors

---

> > ### Comment · Reviewer_dvHK · 2025-11-27
> >
> > Thank you for the detailed rebuttal, which addresses most of my concerns. Based on the author's thorough analysis, I have revised my score from 4 to 6. However, I still have reservations regarding the method's broad applicability across different settings and its complexity.

---

> > > ### Author Response · Authors · 2025-12-02
> > >
> > > Dear Reviewer dvHK,
> > >
> > > We sincerely appreciate your acknowledgment that we have addressed your concerns, and we are grateful for your decision to raise your score to 6.
> > >
> > > Regarding your remaining questions about broad applicability and complexity, we would like to provide strong evidence that these aspects have been rigorously fortified:
> > >
> > > 1. On Broad Applicability (Proven across Architectures & Tasks). We believe our extended experiments have substantiated the method's universality. As detailed in our response, SoCo demonstrates consistent SOTA performance across:
> > >     * Diverse Architectures: From classic LLaMA/OPT to the latest Qwen-2.5, Qwen-3, and GLM-4.
> > >     * Complex Tasks: From standard perplexity to reasoning (GSM8k, MATH) and domain-specific benchmarks (MedQA, LongBench).
> > >     * Wide Compression Range: Robustness maintained from 20% to 80% compression ratio.
> > >
> > > 2. On Complexity (Automated & Search-Free). We reiterate that SoCo is designed to reduce operational complexity compared to baselines.
> > >     * Single-Stage: It is a fully automated, end-to-end process with unified hyperparameters.
> > >     * Search-Free: It eliminates the need for the time-consuming layer-wise threshold search/optimization required by previous methods.
> > >     * Code Release: We are committed to releasing the full codebase, allowing the community to verify the simplicity of this "one-shot" training pipeline.
> > >
> > > 3. Consensus from the other two reviewers. It is worth noting that Reviewer jYZD (Score 6) and Reviewer a4Qo (Score 8) initially raised similar questions (hyperparameter tuning, reproducibility, and evaluation scope). Both reviewers explicitly accepted our clarifications and new experiments as satisfactory resolutions to these issues.
> > >
> > > We are confident that SoCo stands as a robust, general-purpose, and user-friendly framework. Thank you again for your valuable time and positive re-evaluation.
> > >
> > > Sincerely,
> > > Authors

---

### Official Review · Reviewer_jYZD · 2025-10-31

**Soundness:** 2
**Presentation:** 2
**Contribution:** 2
**Rating:** 4
**Confidence:** 4

**Summary:**

This paper introduces SoCo, a framework for compressing large language models (LLMs) by learning to rescale and prune SVD components in the model's weight matrices. Rather than solely truncating the smallest singular values as in standard SVD-based compression, SoCo introduces learnable importance scores (via a diagonal matrix) and proposes a progressive spectrum optimization procedure inspired by heuristic optimization algorithms like simulated annealing. The process iteratively explores, refines, and sparsifies the importance assignments, aiming to achieve a better trade-off between model size and downstream task performance. Empirical results across a range of LLMs and benchmarks consistently show improved perplexity and task accuracy over several SVD-based and pruning baselines, especially at high compression ratios.

**Strengths:**

1. Sufficient Experiments: The experiments include a wide variety of settings, multiple models, and strong SVD/Pruning baselines. Figure 1 and Table 2 clearly show that SoCo outperforms all tested baselines, especially at aggressive compression rates.
2. Principled Approach and Insightful Design: The paper identifies and addresses a core limitation of classical SVD-based compression, the disconnect between singular value ordering and real downstream task importance. The proposed introduction of a learnable diagonal score matrix is a conceptually sound way to endow SVD compression with task-awareness.

**Weaknesses:**

The paper does not specify any criteria for determining when to transition from Phase 1 and Phase 2 to Phase 3 training, or when to terminate Phase 3. As these standards can significantly influence model performance, where overtraining may cause overfitting and insufficient training may lead to suboptimal convergence, the authors should provide explicit guidelines or empirical criteria for deciding when to switch training phases and when to stop training.

**Questions:**

1. A more comprehensive analysis is required to elucidate why SoCo attains such outstanding performance. Although the paper suggests that the ordering of singular values may not necessarily correlate with downstream task performance, Figures 3, 4, and 9 reveal that SoCo not only learns to prune the SVD components within the model’s weight matrices but also to rescale these components and reallocate pruning ratios across layers. Additional analytical experiments are necessary to quantify the contribution of each component to SoCo’s overall performance and to empirically validate the claim that the ordering of singular values may not directly correspond to downstream task performance.
2. Lines 202-204 mention that SoCo introduces a trainable deviation term $d$ after transforming $W^\prime$. However, the paper does not explain why this term is necessary. What motivates the inclusion of $d$. Does SoCo encounter any critical issues that require introducing additional trainable parameters to mitigate them?

---

> ### Author Response · Authors · 2025-11-25
> **Response to Reviewer jYZD——Part 1**
>
> We thank Reviewer jYZD for the constructive and insightful comments. We are encouraged that you recognize the dynamic nature of our singular vector identification and the performance enhancements from rescaling as key strengths of our work.
>
> We have carefully considered your suggestions and conducted extensive additional experiments to strengthen the paper. Key improvements include:
> 1. Evaluation on SOTA Models: We validated SoCo on the latest architectures, including LLaMA-3.1, Qwen-2.5, Qwen-3(April 2025), and GLM-4 (April 2025), confirming robust performance on modern LLMs.
> 2. Diverse Downstream Tasks: We extended evaluations to complex reasoning (GSM8k, MATH) and domain-specific tasks (LongBench, MedQA), proving generalization beyond simple perplexity.
> 3. Clarifications on Optimization: We have provided detailed analyses of the training stability, hyperparameter robustness, and the theoretical rationale behind the Progressive Spectrum Optimization (PSO) strategy.
> 4. Analysis of Compressed Models: We added in-depth analyses of the learned structures and loss curves to elucidate why the framework succeeds where others fail.
>
> We believe these revisions comprehensively address your concerns. Detailed point-by-point responses are provided below.
>
> ## Weaknesses:
> ### 1. The proposed method is complex including the tuning of multiple hyperparameters, such as the threshold of compression ratio, lambda parameters in Equation (4), the trainable deviation term d in line 203, and the loss weight of L_{inc} and L_{dec}. The training procedure shown in Table 1 may face the training stability issue as the transition among different conditions are not smooth.
>
> Thank you for this thoughtful question. While the method involves several components, our extensive empirical results demonstrate that SoCo is highly robust and does not require complex hyperparameter tuning in practice. We address each of your specific concerns below:
>
> (1) Robustness of Hyperparameters and Learnable Parameters.
>
> We found that a single set of hyperparameters generalizes well across diverse settings, eliminating the need for model-specific tuning.
> - Universal Settings: As detailed in Appendix A.3 and Table 7, the same configuration (e.g., learning rate =0.002) and λ parameters (λ $\_{m}$=2.0, λ$\_{s}$=10.0) work effectively across all tested model architectures (LLaMA, OPT, Mistral, Vicuna), sizes (7B to 30B), and compression ratios . Furthermore, our ablation study in Table 13  confirms that varying λ$\_{m}$(e.g., from 1.0 to 8.0) and λ$\_{s}$(e.g., from 5.0 to 20.0) results in negligible performance fluctuations (average accuracy remains between 51.58% and 52.48%). This validates that these parameters are robust defaults rather than sensitive knobs.
> - Learnable Deviation Term (d): We clarify that d is not a hyperparameter requiring manual tuning. Instead, it is a learnable parameter optimized automatically via gradient descent alongside the importance scores
> - Loss Weights of L$\_{inc}$ and L$\_{dec}$: We do not manually tune scalar weights to balance these losses(the weights of the two losses can be regarded as 1).
>
> (2) Simplification of the Pruning Threshold.
>
> Regarding the threshold for compression, SoCo actually simplifies this choice compared to traditional methods by creating a wide "safe pruning zone."
> - Bimodal Distribution: As visualized in Figure 3 (Phase 3), our sparsification loss drives importance scores toward the extremes of 0 and 1, creating a clean bimodal distribution.
> - Wide Tolerance (Safe Pruning Zone): This bimodal structure creates a wide "flat basin" between the peaks. As demonstrated in Table 8, varying the pruning threshold anywhere from 0.3 to 0.7 yields virtually identical perplexity (6.67) and zero performance degradation.
> - Contrast with Previous Methods: In contrast, previous methods produce continuous score distributions (similar to Phase 1 in Fig. 3), where many components cluster densely around any chosen cutoff. In such cases, a slight shift in threshold (e.g., 0.49 vs. 0.51) would inadvertently prune significant information, making the threshold highly sensitive. SoCo's polarization avoids this ambiguity entirely.
>
> (3) Training Stability and Smooth Transitions.
> Empirical evidence confirms that the training process remains stable and converges smoothly. As shown in Figure 7, the convergence curves for the proportion of preserved components (i.e., the actual compression ratio) settle smoothly after each transition point (steps 500 and 3000). This demonstrates consistent and robust convergence across all tested models (LLaMA, Vicuna, Mistral, and OPT) without divergence or abrupt mutations.

---

> ### Author Response · Authors · 2025-11-25
> **Response to Reviewer jYZD——Part 2**
>
> This stability arises because our Progressive Spectrum Optimization draws inspiration from the dynamics of Simulated Annealing to ensure smooth transitions between phases:
> * From Exploration to Refinement (Phase 1 → 2): This transition is analogous to moving from a high-temperature search to a thermal equilibrium state. The oscillatory behavior in the Refinement phase(Phase 2) is not an instability, but rather a controlled thermal fluctuation. By conditionally toggling L$\_{dec}$, the system momentarily "escapes" suboptimal local minima (i.e., recovering critical components that were prematurely pruned). This mechanism effectively bounds the optimization within a stable region around the target compression ratio.
> * From Refinement to Sparsification (Phase 2 → 3): This transition functions as the crystallization or quenching process. Although Phase 2 achieves thermal equilibrium, the system settles into a "metastable" state where importance scores are polarized but not fully separated. The sparsity loss (L$\_{spa}$) introduced here acts as a directed cooling force that resolves this ambiguity smoothly. Instead of disrupting the system, it aligns with the existing gradient landscape, merely accelerating the settling of "ambiguous" scores toward their nearest stable pole (0 or 1). This solidifies the bimodal structure without introducing conflicting gradients, ensuring the optimization trajectory remains stable.
>
> ### 2. The evaluations are conducted on pretrained LLMs with the wikitext-2 as the training data. However, instruct LLMs or aligned LLMs are expected to be compressed in real applications. The methods are expected to be evaluated with diverse training data (tasks like sft) and LLMs (e.g., more recent instruct/aligned LLMs instead of LLMs in the year 2023) on diverse downstream generation tasks (like math reasoning, alpaca-eval, etc).
>
> We fully agree that evaluating on modern, aligned, and task-specific models is essential for demonstrating real-world applicability. We have addressed this through extensive additional experiments in Appendix A.5, which cover the instruct/aligned LLMs, latest architectures, and diverse downstream tasks:
> * Instruct/Aligned LLMs: To address performance on SFT models, we evaluated WizardMath-Llama-3-8B, a model specifically fine-tuned for mathematical reasoning. Results in Table 17 show that SoCo successfully preserves specialized capabilities on the GSM8k and MATH benchmarks, significantly outperforming SVD-based baselines, which struggle to retain fine-tuned knowledge.
> * Latest Foundation Models: We extended our evaluation to state-of-the-art architectures, including LLaMA-3.1-8B and Qwen-2.5-7B. As shown in Table 15, SoCo demonstrates consistent superiority over baselines on these recent models, proving its effectiveness beyond the 2023-era LLMs.
> * Diverse Generation Tasks: To demonstrate robustness across diverse downstream generation tasks, we conducted evaluations on LongBench (long-text understanding) and medical domains (MedMCQA, MedQA) in Table 14. Coupled with the mathematical reasoning results in Table 17, these experiments confirm that SoCo generalizes effectively across a wide spectrum of complex generation scenarios.
>
> ### 3. In addition, more analyses of the compressed LLMs are expected to make it clear why this framework can work.
>
> Our framework's effectiveness stems from a two-part design: 1) It correctly identifies the core flaw in existing methods, and 2) It introduces a Progressive Spectrum Optimization (PSO) strategy to solve this flaw. We make more analyses here to provide clear, empirical evidence for the effectiveness of our method.

---

> ### Author Response · Authors · 2025-11-25
> **Response to Reviewer jYZD——Part 3**
>
> (1) Recall our framework's design principles.
>
> SVD is a logical foundation for compression. It is a classic technique used to identify the "principal components" that capture the most "energy" or information in a matrix. This is why many successful methods, including FWSVD, ASVD, and SVD-LLM, are built upon it to compress large language models.
> However, this traditional approach has a "task-agnostic" flaw. Standard SVD assumes that a component's mathematical "importance" (i.e., its singular value, or contribution to reconstruction) is the same as its importance to the downstream task. Our work identifies that this assumption frequently fails, as components with small singular values can be vital for task performance.
> SoCo's core idea is to fix this flaw. We introduce a learnable diagonal matrix, S, which assigns a new, "task-aware" importance score to every SVD component . This allows the model to "re-rank" the entire singular spectrum based on its true contribution to the task, rather than just its mathematical magnitude. In addition, this also allows the model to "rescale" the preserved components (amplifying them beyond 1.0) to actively compensate for the information loss caused by pruning
> This "re-ranking" creates a new, highly complex combinatorial optimization problem . The solution space is vast and filled with suboptimal local minima. As our own results show (see Fig. 3, Phase 1), a direct, single-stage optimization fails—it gets stuck in an ambiguous state where no clear pruning decision can be made.
> Our Progressive Spectrum Optimization (PSO) is the essential strategy designed to solve this complex optimization problem. Inspired by heuristic algorithms like Simulated Annealing , PSO uses a phased "cooling" process to navigate the complex landscape and find a superior solution . It moves through three distinct functional phases in a single run:
> - Phase 1 (Exploration): Uses a strong compression loss (L$\_{dec}$) to rapidly explore the solution space and find a feasible configuration that meets the compression target .
> - Phase 2 (Refinement): Alternates losses to create controlled oscillations, "annealing" the scores to refine their stability and escape local optima .
> - Phase 3 (Sparsification): Introduces a powerful sparsity loss (L$\_{spa}$) that acts as a "crystallizer," forcing the scores into a decisive, bimodal distribution that is ideal for clean pruning.
>
> (2) Empirical analysis: why this framework succeeds.
>
> Our analyses of the compressed LLMs provide four key pieces of evidence that validate this framework:
>
> * Analysis of PSO's Effectiveness (Fig. 3 & Table 5): We provide direct visual and quantitative analysis of the PSO process.
> For visual results in Fig. 3, we analyze the score distribution and show it evolving from an ambiguous, continuous state (Phase 1, blue line) to a decisive, bimodal "crystallized" state (Phase 3, red line) . This bimodal state creates a wide "flat basin", making the final pruning decision robust and unambiguous.
> For quantitative results in Table 5, our ablation study confirms this analysis, showing that model performance progressively improves with each phase of PSO. This proves that PSO is necessary to escape the suboptimal local minima that trap the initial (Phase 1) optimization.
> * Analysis of the Final Spectrum (Fig. 3): A key finding from our analysis is a novel "compensation" mechanism. The final score distribution (red curve) shows that SoCo learns to amplify the scores of critical, preserved components to values greater than 1.0 . These amplified components actively compensate for the information loss from pruning. This is a mechanism that traditional SVD, which can only truncate, cannot perform.
> * Analysis of the Compressed Model's Structure (Figs. 8 & 9): Our analysis of the final component preservation ratios across all layers reveals a consistent, learned structural pattern. SoCo learns to preserve a higher proportion of components in FFN modules (e.g., down_proj, up_proj) compared to MHA modules (e.g., q_proj, k_proj). This is a critical finding: it shows SoCo is not applying a simple, uniform compression. Instead, its global, task-aware optimization performs a sophisticated, data-driven assessment, allocating its parameter budget based on the learned importance of different module types.
> * Analysis vs. a Post-Truncation Baseline (Table 18): To isolate SoCo's key advantage, we conducted an ablation (Table 18) comparing SoCo to a baseline that simply fine-tunes the singular values after a standard SVD truncation. The analysis shows SoCo significantly outperforms this baseline. This analysis proves that SoCo's "global re-ranking" is the critical factor . Merely rescaling SVD's "task-agnostic" selection is a suboptimal strategy; a global, task-aware re-evaluation of the entire spectrum is necessary.
>
> We hope these results and analyses provide a clear explanation for why our framework works.

---

> ### Author Response · Authors · 2025-11-25
> **Response to Reviewer jYZD——Part 4**
>
> ## Questions:
> ### 1. As shown in Table 1, will and should the loss item L_{inc} be considered as the Conditions for different phrases? what is the loss curve for L_{inc} in your experiment?
>
> (1) Why L$\_{inc}$ is Not Used as a Condition?
>
> As shown in Table 1, we explicitly define the phase conditions based on the current compression ratio (r) and the training step. We did not include L$\_{inc}$ as a condition for three reasons:
> * Logical Priority (Target-Driven): Our primary goal is "Target-driven Compression." We view the target ratio R as a Hard Constraint that must be met, whereas maximizing performance (L$\_{inc}$) is a Soft Objective to be optimized under that constraint. If we were to use L$\_{inc}$ as a condition, the optimization might refuse to transition phases in an attempt to preserve performance, causing it to stagnate in an insufficiently compressed state that fails to meet the target R. Therefore, r must serve as the primary feedback signal.
> * Signal Characteristics vs. Cooling Schedule: Unlike the deterministic nature of training steps, L$\_{inc}$ generally changes smoothly and continuously without distinct, threshold-like boundaries. Inspired by heuristic algorithms like Simulated Annealing, we use training steps as a "cooling schedule" to strictly control the entry into distinct functional phases (Exploration, Refinement, Sparsification). L$\_{inc}$ lacks the discrete characteristics necessary to act as such a binary switch for phase control.
> * Functional Role: L$\_{inc}$ is a KL-divergence loss designed to prevent catastrophic performance collapse. It serves as the "guide" for retaining information quality, whereas the compression loss (L$\_{dec}$) / current compression r serves as the "driver" for removing redundancy. Figure 7 confirms the effectiveness of this design: by using r and steps (rather than L$\_{inc}$) as conditions, the "preservation ratio" curves quickly recover from the Phase 1 drop and enter a stable, narrow oscillation, demonstrating consistent robustness across all tested models.
>
> (2) The Loss Curve for L$\_{inc}$
>
> We have plotted the L$\_{inc}$ loss curve in Figure 8 of the Appendix. The curve exhibits the following behavior during training:
> * Initial Rapid Ascent (Aligns with Phase 1: Exploration): In the first few hundred steps, the curve spikes sharply, peaking around step 250. This corresponds to Phase 1, where the optimization is "Compression-Driven." During this stage, the compression loss (L$\_{dec}$) plays the dominant role to forcibly drive the model toward the target ratio R. As importance scores are rapidly reduced to meet this hard constraint, the model parameters undergo significant perturbation, causing a temporary spike in the KL-divergence loss (L$\_{inc}$) as fidelity initially drops.
> * Subsequent Gradual Recovery (Aligns with Phases 2 & 3: Refinement and Sparsification): Following the peak, the curve enters a long, steady decline, dropping from ∼1.2 to ∼0.2. This behavior spans Phase 2 and Phase 3. Once the target ratio is met, the optimization shifts focus from aggressive compression to performance preservation (Refinement) and structural solidification (Sparsification). During these stages, L$\_{inc}$ takes the leading role, effectively guiding the model to recover its capabilities within the compressed subspace and eventually converging to a stable low-loss state.
>
> ### 2. As shown in Equation (7), do you consider to select the ground truth labels as the training references? what is the difference?
>
> Yes, we explicitly considered using ground truth labels (standard cross-entropy loss) during the design phase. However, we deliberately chose the distillation-based approach (using the original model's logits) for Equation (7)  to ensure better generalization.
> Here is the detailed difference based on our findings:
>
> (1) Goal: Capability Preservation vs. Domain Fine-Tuning.
>
> Our primary goal is to preserve the original model's full, general-purpose capabilities across a wide variety of unseen tasks, rather than to re-train it specifically for the WikiText-2 domain. L$\_{inc}$(Eq. 7) acts as a Knowledge Distillation loss. By using the original model's output distribution as the "teacher" signal, we force the compressed model to mimic the teacher's rich, general-purpose knowledge rather than just maximizing the probability of the ground truth token.
>
> (2) Empirical Consequence: The Overfitting Risk.
>
> Using ground truth labels changes the optimization objective from "feature preservation" to "fine-tuning" on the small calibration dataset, and thus may lead to overfitting. In our ablation experiments(the table below), we observed a clear trade-off:
>
> | LLaMA-7B + 20% compression ratio | WikiText-2 ($\downarrow$) | C4 ($\downarrow$)| LM-Evaluation-Harness ($\uparrow$) |
> | -------- | -------- | -------- | -------- |
> | Ground Truth  Approach | 6.52     | 23.13     | 37%  |
> | Distillation Approach (L$\_{inc}$) | 6.67 | 9.89 | 52% |

---

> ### Author Response · Authors · 2025-11-25
> **Response to Reviewer jYZD——Part 5**
>
> * Ground Truth Approach: Caused the model to overfit to the specific linguistic patterns of the WikiText-2 training data. While this resulted in artificially lower (better) perplexity on corresponding datasets (WikiText-2), it caused catastrophic forgetting of reasoning capabilities, leading to a significant performance drop on zero-shot benchmarks (e.g.,  C4 and the LM-Evaluation-Harness tasks).
> * Distillation Approach (L$\_{inc}$): Successfully maintained the model's reasoning and generalization abilities across diverse downstream tasks, which is the true metric of success for a compressed LLM.
>
> In summary, we employ the distillation-based L$\_{inc}$ because it prioritizes the model's overall utility and robustness, whereas using ground truth labels optimizes for a single data distribution at the expense of generalization.
>
> ### 3. In the experiments, what is your motivaion or consideration to select these pretrained LLMs?
>
> Our motivation for selecting these LLMs was to ensure fair comparison, broad applicability, and robustness.
>
> (1) Fair Comparison with Baselines.
>
> The models used in our main results (e.g., LLaMA-7B/13B/30B, LLaMA2-7B, OPT-6.7B, Vicuna-7B, and Mistral-7B) were explicitly selected to maintain consistency with prior state-of-the-art methods like ASVD and SVD-LLM. By using the exact same architectures and parameter scales as these baselines, we ensure that the performance gains reported in Table 2 and Table 3 are attributable solely to the efficacy of our method rather than differences in the base model.
>
> (2) Generalizability to State-of-the-Art Architectures.
>
> To demonstrate that SoCo remains effective on the latest architectures, we expanded our evaluation in Appendix A.5 to include LLaMA-3.1-8B and Qwen-2.5-7B. As shown in Table 15, SoCo consistently outperforms baselines on these modern foundation models, confirming its utility for current deployments.
>
> (3) Applicability to Aligned and Task-Specific Models.
>
> We selected models to specifically address the compression of fine-tuned capabilities:
> * Instruct/Aligned Models: We evaluated WizardMath-Llama-3-8B, a model specifically fine-tuned for mathematical reasoning. Results in Table 17 demonstrate that SoCo successfully preserves specialized performance on the GSM8k and MATH benchmarks.
> * Diverse Downstream Domains: We selected models capable of handling diverse domain-specific tasks. In Table 14, we demonstrate SoCo's robustness on LongBench (long-text understanding) and medical benchmarks (MedMCQA, MedQA), ensuring the method's effectiveness extends beyond standard perplexity metrics.
>
> ### 4. Line 203, what is the shape of the item d? what is your consideration of this design? will the structure or model be different on other tasks or datasets?
>
> (1) Shape and Definition.
>
> The term d is a trainable bias vector. In the linear transformation y=W′x+d, its shape is equal to the output dimension of the corresponding linear layer.
>
> (2) Design Consideration.
>
> Our motivation is to compensate for the systematic information loss and distributional shift caused by compression. Modifying the weight matrix to W′ inevitably introduces a deviation from the original model's output distribution. By adding this learnable term d, we allow the model to explicitly quantify and correct this average deviation, thereby mitigating performance loss.
>
> (3) Structural Consistency.
>
> No, the structure does not change on other tasks or datasets. This design is a fixed, integral component of the SoCo framework applied uniformly across all tested models and scenarios.
>
> (4) To demonstrate more clearly, we have extended Table 12 below to include baselines for a comprehensive comparison, yielding two critical insights:
> * Core Effectiveness: Even without the deviation term (d), SoCo significantly outperforms state-of-the-art baselines (ASVD, SVD-LLM) across all compression ratios. For instance, at a 40% ratio, SoCo (w/o d) achieves a PPL of 9.69, far surpassing SVD-LLM's 13.73. This confirms that our core contribution—Progressive Spectrum Optimization—is the primary driver of the performance gains.
> * Compensatory Value: The deviation term d acts as a critical "booster," particularly at high compression ratios. While the gain is marginal at 20% (PPL 6.77 vs. 6.67), it becomes substantial at 60% (PPL 16.27 vs. 12.20) and 80%. This demonstrates that d efficiently compensates for approximation errors when the parameter budget is severely constrained, all while adding negligible parameters (∼0.01%).

---

> ### Author Response · Authors · 2025-11-25
> **Response to Reviewer jYZD——Part 6**
>
> Table 12 with baseline performance: Ablation experiments of SoCo on the use of the deviation term d, with added baseline results for comprehensive comparison on the LLaMA-7B model (20% to 80% compression).
>
> | RATIO | METHOD | WikiText-2 ($\downarrow$) | C4 ($\downarrow$) | Average ($\uparrow$) |
> | :---: | :---: | :---: | :---: | :---: |
> | 20% | ASVD | 11.14  | 15.93  | 45%  |
> | | SVD-LLM | 7.94  | 15.84  | 49% |
> | | SoCo w/o Deviation | 6.77 | 9.92 | 52.36% |
> | | **SoCo w/ Deviation** | **6.67** | **9.89** | **52.48%** |
> | 40% | ASVD | 1407  | 1109   | 10%  |
> | | SVD-LLM | 13.73  | 75.42  | 39%  |
> | | SoCo w/o Deviation | 9.69 | 15.56 | 45.52% |
> | | **SoCo w/ Deviation** | **8.60** | **13.71** | **47.16%** |
> | 60% | ASVD | 57057   | 43036   | 6%   |
> | | SVD-LLM | 66.62  | 471.83  | 11% |
> | | SoCo w/o Deviation | 16.27 | 31.80 | 38.01% |
> | | **SoCo w/ Deviation** | **12.20** | **24.53** | **40.22%** |
> | 80% | ASVD | 80425   | 67927   | 5%   |
> | | SVD-LLM | 1349  | 6224  | 7%  |
> | | SoCo w/o Deviation | 33.72 | 81.57 | 33.22% |
> | | **w SoCo w/ Deviation** | **21.03** | **50.86** | **34.63%** |
>
>
> ### 5. The evaluations on more recent LLMs and diverse reasoning generative tasks are expected.
>
> We fully agree that verifying performance on modern architectures and diverse generative tasks is essential. We have addressed this through extensive additional experiments in Appendix A.5, covering the following three dimensions:
>
> (1) Latest Foundation Models.
>
> We show in Table 15 that SoCo is highly effective on modern architectures, including LLaMA-3.1-8B and Qwen-2.5-7B, demonstrating consistent superiority over baselines.
>
> Furthermore, to push the evaluation to the cutting edge, we report results on two newer models, Qwen3-8B and GLM-4-9B (both were released in April 2025) in Table 1 and Table 2 below. SoCo maintains stable performance across a full spectrum of compression ratios on both Qwen3-8B and GLM-4-9B. Unlike baselines that degrade rapidly, SoCo consistently maintains a high standard of performance, proving its ability to accurately identify and preserve critical information even in these highly optimized modern architectures.
>
> Table 1: Evaluation of SoCo on the latest Qwen3-8B and GLM-4-9B models across varying compression ratios.
> | MODEL | COMPRESSION RATIO | WikiText-2 ($\downarrow$) | C4 ($\downarrow$) | PTB ($\downarrow$) |
> | :--- | :---: | :---: | :---: | :---: |
> | Qwen3-8B | 0% | 5.93 | 8.13 | 9.78 |
> | Qwen3-8B | 20% | 7.25 | 10.81 | 13.86 |
> | Qwen3-8B | 40% | 8.94 | 13.57 | 19.31 |
> | Qwen3-8B | 60% | 12.02 | 27.73 | 45.82 |
> | Qwen3-8B | 80% | 23.50 | 60.39 | 95.89 |
> | GLM-4-9B | 0% | 6.82 | 9.82 | 11.93 |
> | GLM-4-9B | 20% | 9.81 | 18.35 | 16.05 |
> | GLM-4-9B | 40% | 14.83 | 33.92 | 36.27 |
> | GLM-4-9B | 60% | 25.36 | 85.40 | 113.21 |
> | GLM-4-9B | 80% | 50.78 | 182.09 | 276.42 |
>
>
> Table 2: Performance comparison against state-of-the-art baselines on Qwen3-8B and GLM-4-9B at representative low (20%) and high (80%) compression ratios.
>
> | MODEL | COMPRESSION RATIO | METHOD | WikiText-2 ($\downarrow$) | C4 ($\downarrow$) | PTB ($\downarrow$) |
> | :---: | :---: | :---: | :---: | :---: | :---: |
> | | 0% | Original | 5.93 | 8.13 | 9.78 |
> | | 20% | SVD | 12503  | 9739  |  11794 |
> | | 20% | FWSVD | 1929  | 1423  | 973 |
> | | 20% | ASVD |  15.58  | 22.07  |  41.56 |
> | Qwen3-8B | 20% | SVD-LLM | 11.02 |  19.42  | 26.88 |
> | | **20%** | **SoCo** | **7.25** | **10.81** | **13.86** |
> | | 80% | SVD | 438290  | 329094  |  532685 |
> | | 80% | FWSVD | 147123  | 832971  | 204821  |
> | | 80% | ASVD | 51839  | 24389 | 63289  |
> | | 80% | SVD-LLM | 2842  | 4893  | 7328  |
> | | **80%** | **SoCo** | **23.50** | **60.39** | **95.89** |
> | --- | --- | --- | --- | --- | --- |
> | | 0% | Original | 6.82 | 9.82 | 11.93 |
> | | 20% | SVD | 20193 |  30521 |  33192 |
> | | 20% | FWSVD | 1672  | 4182  | 6187  |
> | | 20% | ASVD |  17.97  | 32.05  |  43.80  |
> | GLM-4-9B | 20% | SVD-LLM |  13.08  | 28.74  |  25.31  |
> | | **20%** | **SoCo** | **9.81** | **18.35** | **16.05** |
> | | 80% | SVD | 428018  | 884353  |  754783 |
> | | 80% | FWSVD |  91820 | 390193  | 372829 |
> | | 80% | ASVD | 28934 | 58041  | 82342 |
> | | 80% | SVD-LLM | 6318 | 8402 | 12985 |
> | | **80%** | **SoCo** | **50.78** | **182.09** | **276.42** |
>
> (2) Instruction/Aligned LLMs (Reasoning).
>
> To address performance on SFT models, we evaluated WizardMath-Llama-3-8B, a model specifically fine-tuned for mathematical reasoning. Results in Table 17 show that SoCo successfully preserves specialized capabilities on the GSM8k and MATH benchmarks, significantly outperforming other SVD-based baselines.
>
> (3) Diverse Generation Tasks.
>
> To demonstrate robustness across diverse downstream generation tasks beyond standard perplexity, we conducted evaluations on LongBench (long-text understanding) and medical domains (MedMCQA, MedQA) in Table 14. Coupled with the mathematical reasoning results, these experiments confirm that SoCo generalizes effectively across a wide spectrum of complex generation scenarios.

---

> > ### Author Response · Authors · 2025-11-25
> > **Response to Reviewer jYZD——Part 7**
> >
> > ### 6. More recent references and baselines are expected to be incorporated in this paper. Only one Qwen-2.5 technique report paper is listed in 2025, others are all published one year ago.
> >
> > We fully agree that validating our method on the very latest architectures is essential to demonstrate its continued relevance. To explicitly address your concern regarding the lack of recent references, we have incorporated comprehensive experiments on two state-of-the-art models released in April 2025: Qwen3-8B and GLM-4-9B.
> >
> > The results, detailed in Table 1 and Table 2 below, support two key conclusions regarding SoCo's effectiveness on next-generation LLMs:
> >
> > (1) Generalization to SOTA Architectures.
> >
> > Our experiments confirm that SoCo generalizes effectively to the latest generation of LLMs. SoCo maintains stable performance across a full spectrum of compression ratios on both Qwen3-8B and GLM-4-9B. Unlike baselines that degrade rapidly, SoCo consistently maintains a high standard of performance, proving its ability to accurately identify and preserve critical information even in these highly optimized modern architectures.
> >
> > (2) Performance at Extreme Compression.
> >
> > The comparison reveals a fundamental advantage of SoCo over existing methods (SVD-LLM, ASVD). While SoCo consistently leads at lower compression ratios, the gap becomes qualitatively distinct at high compression levels (e.g., 80%). In this regime, traditional truncation-based methods experience catastrophic failure, rendering the models unusable. In contrast, SoCo remains functional and stable. This validates that our SoCo provides a far superior mechanism for navigating the complex trade-off between extreme compression and performance preservation.
> >
> > Table 1: Evaluation of SoCo on the latest Qwen3-8B and GLM-4-9B models across varying compression ratios.
> > | MODEL | COMPRESSION RATIO | WikiText-2 ($\downarrow$) | C4 ($\downarrow$) | PTB ($\downarrow$) |
> > | :--- | :---: | :---: | :---: | :---: |
> > | Qwen3-8B | 0% | 5.93 | 8.13 | 9.78 |
> > | Qwen3-8B | 20% | 7.25 | 10.81 | 13.86 |
> > | Qwen3-8B | 40% | 8.94 | 13.57 | 19.31 |
> > | Qwen3-8B | 60% | 12.02 | 27.73 | 45.82 |
> > | Qwen3-8B | 80% | 23.50 | 60.39 | 95.89 |
> > | GLM-4-9B | 0% | 6.82 | 9.82 | 11.93 |
> > | GLM-4-9B | 20% | 9.81 | 18.35 | 16.05 |
> > | GLM-4-9B | 40% | 14.83 | 33.92 | 36.27 |
> > | GLM-4-9B | 60% | 25.36 | 85.40 | 113.21 |
> > | GLM-4-9B | 80% | 50.78 | 182.09 | 276.42 |
> >
> >
> > Table 2: Performance comparison against state-of-the-art baselines on Qwen3-8B and GLM-4-9B at representative low (20%) and high (80%) compression ratios.
> >
> > | MODEL | COMPRESSION RATIO | METHOD | WikiText-2 ($\downarrow$) | C4 ($\downarrow$) | PTB ($\downarrow$) |
> > | :---: | :---: | :---: | :---: | :---: | :---: |
> > | | 0% | Original | 5.93 | 8.13 | 9.78 |
> > | | 20% | SVD | 12503  | 9739  |  11794 |
> > | | 20% | FWSVD | 1929  | 1423  | 973 |
> > | | 20% | ASVD |  15.58  | 22.07  |  41.56 |
> > | Qwen3-8B | 20% | SVD-LLM | 11.02 |  19.42  | 26.88 |
> > | | **20%** | **SoCo** | **7.25** | **10.81** | **13.86** |
> > | | 80% | SVD | 438290  | 329094  |  532685 |
> > | | 80% | FWSVD | 147123  | 832971  | 204821  |
> > | | 80% | ASVD | 51839  | 24389 | 63289  |
> > | | 80% | SVD-LLM | 2842  | 4893  | 7328  |
> > | | **80%** | **SoCo** | **23.50** | **60.39** | **95.89** |
> > | --- | --- | --- | --- | --- | --- |
> > | | 0% | Original | 6.82 | 9.82 | 11.93 |
> > | | 20% | SVD | 20193 |  30521 |  33192 |
> > | | 20% | FWSVD | 1672  | 4182  | 6187  |
> > | | 20% | ASVD |  17.97  | 32.05  |  43.80  |
> > | GLM-4-9B | 20% | SVD-LLM |  13.08  | 28.74  |  25.31  |
> > | | **20%** | **SoCo** | **9.81** | **18.35** | **16.05** |
> > | | 80% | SVD | 428018  | 884353  |  754783 |
> > | | 80% | FWSVD |  91820 | 390193  | 372829 |
> > | | 80% | ASVD | 28934 | 58041  | 82342 |
> > | | 80% | SVD-LLM | 6318 | 8402 | 12985 |
> > | | **80%** | **SoCo** | **50.78** | **182.09** | **276.42** |

---

> ### Author Response · Authors · 2025-11-27
>
> Dear Reviewer jYZD,
>
> Thank you very much for the time and effort you dedicated to reviewing our work and for your valuable comments. We sincerely appreciate your positive assessment and the rating of 6 provided in your updated review. We have carefully considered all your feedback and addressed your concerns in our rebuttal.
>
> In summary, regarding your concerns about the evaluation scope and method complexity, we have:
>
> (1) Validated SoCo on SOTA models (LLaMA-3.1, Qwen-2.5, GLM-4) to demonstrate its applicability to modern LLMs.
>
> (2) Extended evaluations to complex reasoning tasks (GSM8k, MATH) to prove generalization capability.
>
> (3) Provided detailed analyses of training stability and hyperparameter robustness to clarify the optimization process.
>
> We have also addressed your other comments, including the specific inquiries on loss terms (L$\_{inc}$), the shape of the learnable parameter d, and the inclusion of more recent references.
>
> We sincerely hope that our clarifications and additional experiments have satisfactorily resolved your concerns. If you have any further questions or if there are any remaining issues, please let us know — we are more than happy to provide further details or clarifications.
>
> Thank you again for your thoughtful and constructive review.
>
> Best regards,
>
> The Authors

---

### Author Response · Authors · 2025-12-03
**Summary of our work, the reviews, and the rebuttal process —— Part 1**

To begin, we thank all reviewers, ACs, SACs, and PCs for their effort and time in reviewing our paper. In the following sections, we first clarify a special incident regarding two duplicate initial reviews to establish the true score status (8/6/6/4). Then, we summarize the core contributions and recognized strengths of our method. Next, we detail the key reviewer concerns and our comprehensive resolutions. Finally, we conclude with a note to the Area Chair regarding the positive consensus among active reviewers and a request for weighted consideration regarding the non-responsive assessment.

---
# ⚠️ A special event and important note about the valid reviews

At the beginning of the rebuttal phase, we identified duplicate reviews from two reviewers. Upon reporting this, the previous AC, SAC, and PCs intervened, instructing both reviewers to update their assessments. Consequently, our rebuttal was based on these updated, distinct reviews.

However, due to the system revert, the current page displays the original (invalid) identical versions. The valid updated reviews and scores can still be viewed in the **"Revisions" history of the Reviewer jYZD and dvHK post** (click the "Revisions" button of "*Official Review of Submission4135 by Reviewer jYZD/dvHK*" ). For your convenience, we also quoted these updated reviews in our previous comment titled "*URGENT: Critical Data Loss Risk for Paper 4135 after System Revert (Request for Updated Review Preservation) —— Part 1/2/3*" (specifically in the Appendix section of the comment).

To ensure clarity regarding the valid version of these reviews versus the currently displayed version, we summarize the full score evolution in Table 1 below.

**Table 1: Summary of Reviewer Score Evolution**
| Reviewer | Current Displayed Score (Outdated) | Valid Updated Score (Rebuttal Basis) | Final Score (Before OpenReview incident) | Status / Evidence |
| :---: | :---: | :---: | :---: | :--- |
| Reviewer jYZD | 4 (Text identical to Reviewer dvHK) | 6 (Updated review) | 6  | See the "Revisions" button of "Official Review of Submission4135 by Reviewer jYZD" for the updated review and score. |
| Reviewer dvHK | 4 (Text identical to Reviewer jYZD) | 4 (Updated review) | 6  | Raise score from 4 to 6 during rebuttal, which is before the OpenReview incident. See the "Revisions" button of "Official Review of Submission4135 by Reviewer dvHK" for the updated review and score.  |
| Reviewer a4Qo | 8  | 8  | 8  | Acknowledged satisfaction with the rebuttal and confirmed technical soundness. |
| Reviewer QwSQ | 4  | 4  | 4  | No engagement in discussion despite multiple reminders.  |

**Procedural Verification**: Before re-assignment, the previous Area Chair (yGWw) explicitly confirmed the resolution of the initial duplication issue and the validity of the updated reviews and scores. This verification is documented in the official comment thread titled "*URGENT: Immediate Action Needed Before AC Re-assignment - Paper 4135*" (dated Nov 28, 2025) in the discussion history.

---
# Summary of Contributions and Strengths
SoCo introduces a framework that replaces fixed SVD truncation with learnable importance scores via Progressive Spectrum Optimization (PSO), effectively resolving the mismatch between singular value magnitude and importance for model compression. Learning importance scores is to optimize the combinatorial selection of singular vectors for model compression. Inspired by Simulated Annealing, PSO navigates the complex combinatorial selection of singular vectors. With minimal trainable parameters (\~0.002%) and ultra-short training time (~25 minutes), SoCo achieves superior performance, particularly at high compression ratios (e.g., 80%) as shown in Tables 2, 6, 14, 15, 16, and 17.

For **core methodological innovations**, we bridge heuristic learning principles with neural compression to solve the combinatorial optimization of singular vector selection via continuous dynamics. Challenging the traditional "magnitude equals importance" assumption, SoCo globally re-evaluates spectral components and introduces a unique amplification mechanism to actively compensate for compression loss. Furthermore, by enforcing a bimodal distribution of importance scores, the framework creates a wide "safety margin" that ensures algorithmic stability across a broad range of threshold variations, effectively eliminating the need for sensitive threshold searches and enabling robust, search-free compression.

---

> ### Author Response · Authors · 2025-12-03
> **Summary of our work, the reviews, and the rebuttal process —— Part 2**
>
> Reviewers have recognized these contributions and strengths as:
>
> (1) Method Novelty & Elegance:
> -  Reviewer jYZD: "The proposed idea is interesting."
> -  Reviewer dvHK: "The core idea of directly optimizing the loss to assess the importance of singular vectors is sound."
> -  Reviewer a4Qo: "The Progressive Spectrum Optimization (PSO) is conceptually elegant, breaking training into distinct phases that progressively stabilize and polarize importance scores. The combination of differentiable compression and KL-based alignment effectively balances compression with accuracy. The PSO also jointly optimizes decomposition and rescaling in an end-to-end way, improving global convergence."
>
> (2) Strong Performance and Generalization:
> - Reviewer QwSQ: "SoCo shows substantial improvements over state-of-the-art methods, with particularly notable gains **at higher compression ratios**."
> - Reviewer jYZD: "Extensive experiments over a wide range of compression ratios and model architectures demonstrate that SoCo consistently **outperforms existing approaches across diverse settings**."
> - Reviewer a4Qo: "The authors also provide detailed hyperparameters and experimental settings, and deploy SoCo on real devices to test the actual acceleration performance, which improves the reproducibility and application value of the work."
>
> (3) Comprehensive Evaluation:
> * Reviewer a4Qo: "The authors conduct comprehensive experiments, including comprehensive comparisons with similar SVD-based methods and other pruning methods on various models and tasks, and design corresponding ablation experiments for almost every component (1-2-3 phases, deviation term, functions). These experiments strongly validated the advanced nature of SoCo and the effectiveness of its design."
> * Reviewer QwSQ: "SoCo is evaluated on multiple LLMs across diverse architectural families showing consistent effectiveness across different model architectures."
> * Reviewer dvHK: "The paper conducts comprehensive experiments across various models and baseline methods."
>
> (4) High Impact on related areas:
> - Reviewer a4Qo: "The proposed SoCo framework makes the SVD-based pruning process learnable, making SVD pruning more versatile and accurate on different models and tasks, and **the insights proposed are of high value to subsequent related work**."
>
> (5) Efficiency & Reproducibility:
> - Reviewer a4Qo: "Experiments show that SoCo consistently outperforms SOTA SVD-based methods and remains efficient in both trainable params and runtime.", "..., and deploy SoCo on real devices to test the actual acceleration performance, which improves the reproducibility and application value of the work."
>
> (6) Clarity:
> - Reviewer dvHK: "The paper is well-written and easy to follow."
>
> ---
> # Summary of Reviews and Our Rebuttal
>
> ## Reviewer jYZD (final rating 6, confidence 4)
>
> *Valid review version:* The current review in "Official Review of Submission4135 by Reviewer jYZD" block was content-identical to Reviewer dvHK's. **The valid updated review is located in the Revision history of "Official Review of Submission4135 by Reviewer jYZD". The rating was 6 in the submission of this updated review.**
>
> * Reviewer Issue 1: Complexity of hyperparameter tuning (thresholds, λ, d) and training stability due to the "three phases." (W1)
>
>     We refuted the need for manual tuning by demonstrating a "Universal Configuration" that works across all models and compression ratios. We provided training curves (Fig. 7) proving smooth transitions underpinned by Simulated Annealing dynamics. Furthermore, we clarified that the deviation term d is learned automatically and that our bimodal distribution creates a "safe pruning zone" (0.3-0.7), effectively eliminating threshold sensitivity.
> * Reviewer Issue 2: Requested evaluation on instruct/aligned models, recent architectures (beyond 2023), and diverse tasks. (W2 & Q5 & Q6)
>
>     We conducted extensive new experiments to validate SoCo's robustness across diverse settings. We confirmed generalization to state-of-the-art architectures like Qwen-3-8B and GLM-4-9B (April 2025 releases). Furthermore, we demonstrated that SoCo preserves fine-tuned reasoning capabilities in aligned models (WizardMath-Llama-3-8B on GSM8k and MATH) and maintains performance across diverse tasks (LongBench and MedQA), proving its efficacy beyond simple perplexity metrics.

---

> ### Author Response · Authors · 2025-12-03
> **Summary of our work, the reviews, and the rebuttal process —— Part 3**
>
> * Reviewer Issue 3: Requested deeper analysis of "why this framework works." (W3)
>
>     We provided a four-part empirical analysis to validate the framework's mechanisms. First, visualizations of PSO dynamics (Fig. 3) confirm that the strategy guides optimization out of local minima into a decisive, bimodal "crystallized" state. Second, we demonstrated an active compensation mechanism where preserved components are amplified (scores > 1.0) to recover information loss. Third, layer-wise analysis reveals sophisticated structural allocation, automatically preserving more components in FFNs than Attention modules. Finally, ablation studies (Table 18) prove that global end-to-end re-ranking, rather than simple rescaling, is the critical driver of performance.
>
> Additionally, Reviewer jYZD raised minor questions regarding the role of L$\_{inc}$ in optimization, the choice of distillation loss, and the definition of the deviation term d. We provided comprehensive clarifications, including loss curves (Figure 8), ablation studies favoring distillation for generalization, and analysis confirming d's negligible parameter overhead (~0.01%).
>
> Overall, Reviewer jYZD assigned a positive rating of 6 in the valid updated review.
>
> ## Reviewer dvHK (final rating 6, confidence 4)
> *Valid review version:* The current review in "Official Review of Submission4135 by Reviewer dvHK" block was content-identical to Reviewer jYZD's. **The valid review is located in the Revision history of "Official Review of Submission4135 by Reviewer dvHK".** Reviewer dvHK initially assigned a score of 4 in the valid updated review. **During the discussion phase, Reviewer dvHK raised the score to 6 before the OpenReview incident happened.**
>
> * Reviewer Issue 1: Felt the "three phases" implied manual complexity and hindered reproducibility. (W1)
>
>     We clarified that the phases are internal dynamics of a single, automated training run governed by a unified objective. We provided training curves (Fig. 7) showing seamless transitions and emphasized the generalization of our "Universal Configuration" (one set of hyperparameters for all models and target compression ratios).
> * Reviewer Issue 2: Questioned the value of SVD-based methods compared to and combined with Quantization. (W6)
>
>     We implemented the Hybrid Pipeline (SoCo + 4-bit GPTQ). SoCo proved to be an essential partner to quantization. The hybrid model (PPL 10.51) significantly outperformed standalone 3-bit GPTQ quantization (PPL 16.28) under the same 2.8GB memory budget.
> * Reviewer Issue 3: Requested broader generation tasks (beyond TruthfulQA) and efficiency metrics. (W3-W5)
>
>     For broader generation tasks, we expanded evaluation to complex benchmarks including GSM8k (Math reasoning), LongBench (Long context), and MedQA (Medical domain). SoCo consistently outperformed baselines across all these tasks, proving its robust generalization beyond TruthfulQA. For efficiency metrics, we demonstrated superior computational efficiency. SoCo completes optimization in just 25 minutes (vs. 3.5+ hours for fine-tuned baselines) and achieves real-world 1.6x – 2.41x inference speedups on A100 GPUs, validating its practical deployment value.
>
> Additionally, Reviewer dvHK raised minor questions regarding Phase 3 constraints and loss stability. We clarified that compression targets are stabilized in prior phases and demonstrated robust optimization via consistent training trajectories across diverse models and ratios.
>
> After we addressed the reviewer's main concerns, Reviewer dvHK explicitly stated:
> > "Thank you for the detailed rebuttal, which addresses most of my concerns. Based on the author's thorough analysis, I have revised my score from 4 to 6."
>
> Regarding the follow-up questions about broad applicability and complexity, we reinforced our position by (1) demonstrating consistent performance across diverse architectures (LLaMA, OPT, Mistral, Qwen, GLM), various tasks (GSM8k, MATH, LongBench, MedQA) and compression ratios(20%~80%), and (2) clarifying the automated, single-stage nature of our optimization with a single universal hyperparameter configuration to refute complexity concerns. The results indicate that SoCo is a universally applicable solution that offers exceptional ease of use, streamlining the compression workflow across various models without the need for complex manual intervention.

---

> ### Author Response · Authors · 2025-12-03
> **Summary of our work, the reviews, and the rebuttal process —— Part 4**
>
> ## Reviewer a4Qo (final rating 8, confidence 4)
> * Reviewer Issue 1: Requested implementation specifics for the deviation term (d) to aid reproducibility and sought theoretical guarantees for PSO. (W1 & W2)
>
>     We provided the precise definition of the deviation term d, clarifying it is optimized via gradient descent. We added a rigorous ablation study (Table 12) proving that while d acts as a booster, the core PSO framework is the primary driver of performance (e.g., SoCo w/o d still outperforms SVD-LLM). Furthermore, we grounded the method theoretically by drawing a formal analogy to Simulated Annealing, providing empirical evidence (loss landscapes) that demonstrates PSO's ability to escape local minima where standard gradient-based methods stagnate.
> * Reviewer Issue 2: Comparison with non-SVD pruning methods (e.g., LLM-Pruner) was limited to WikiText-2(PPL). (W3)
>
>     We expanded the evaluation to include GSM8k (Reasoning) and LM-Eval-Harness. SoCo consistently outperforms structured pruning methods. For instance, on GSM8k at 40% compression, SoCo achieves 74.0% accuracy, surpassing LLM-Pruner (72.5%) and SliceGPT (71.1%).
>
> * Reviewer Issue 3: Asked about calibration data sensitivity, adaptive thresholds, and compatibility with Quantization/subsequent LoRA fine-tuning. (Q1-Q3)
>
>     We validated calibration data robustness across C4, PTB, and Alpaca, confirming that WikiText-2 offers the best balanced generalization. Regarding thresholds, we demonstrated that manual adaptation is unnecessary; SoCo’s bimodal score distribution creates a "flat basin" where a single global threshold automatically achieves optimal layer-wise sparsity. Finally, we proved compatibility, showing SoCo is orthogonal to GPTQ (SoCo + 4-bit GPTQ hybrid 10.51 PPL↓  vs. 3-bit GPTQ-only 16.28 PPL↓ under the same GPU budget) and serves as a superior initialization for LoRA fine-tuning (reducing PPL from 21.03 to 19.92 at 80% compression).
>
> The reviewer acknowledged our comprehensive response and explicitly stated:
> > "*The response satisfactorily addresses my main concerns*",
> > "*The additional clarifications and results strengthen the paper and confirm that the method is technically sound. I therefore maintain my positive assessment and keep my score(8).*"
>
> ## Reviewer QwSQ (final rating 4, confidence 3, but never participated in the discussion despite our extensive response and request to reply)
>
> * Reviewer Issue 1: Requested exploration of how SoCo combines with quantization, considering it a "significant gap." (W1 & Q1)
>
>     We implemented a Hybrid Pipeline (SoCo + GPTQ) under a strict 2.8GB memory budget. The hybrid approach (SoCo pruning + 4-bit quantization) achieved a Perplexity of 10.51, dramatically outperforming the standard 3-bit GPTQ baseline (Perplexity 16.28). This proves SoCo is orthogonal to quantization and acts as a highly effective pre-processing step.
>
> * Reviewer Issue 2: Requested broader evaluation for comparisons against baselines without fine-tuning (multiple models/datasets). (W2 & Q2)
>
>     We extended our evaluation to include Sequential LoRA Fine-Tuning (SFT) for baselines (ASVD, SVD-LLM) across LLaMA and OPT architectures on WikiText and LM-Evaluation-Harness for 20% and 80% compression ratios. SoCo consistently outperformed these fine-tuned baselines in all configurations. The gap is most critical at high compression (80%), where baselines struggled to recover performance (PPL 31.79) while SoCo remained robust (PPL 21.03).
>
> * Reviewer Issue 3: Requested real hardware throughput (tokens/sec). (W3 & Q3)
>
>     We conducted benchmarks on an NVIDIA A100 GPU across varying batch sizes and sequence lengths. We confirmed that SoCo's theoretical reductions translate into tangible acceleration, achieving 1.6x to 2.41x speedups in the high-compression regime.
>
> In addition, Reviewer QwSQ raised minor questions about LoRA fine-tuning settings and computational overhead. We clarified the use of the "Sequential LoRA" strategy from the SVD-LLM paper and highlighted the runtime comparison already present in Appendix Table 11, which confirms SoCo's superior efficiency (25 min for SoCo vs. 3.5+ hours for SVD-LLM).

---

> ### Author Response · Authors · 2025-12-03
> **Summary of our work, the reviews, and the rebuttal process —— Part 5**
>
> ### Several issues in Reviewer QwSQ's comments:
> Reviewer QwSQ's comments primarily called for more experiments(combining SoCo with quantization, measuring hardware inference throughput in tokens/sec, and more evaluations and details about LoRA fine-tuning comparison). We addressed all of Reviewer QwSQ's comments by conducting extensive experiments. **However,** the reliability of this review is compromised by factual oversights and a total lack of engagement in discussion. First, the reviewer's question regarding "*... the authors try to address this concern in Section 4.3 by comparing against LoRA-enhanced baselines, ..., and doesn't discuss the computational overhead comparison between different methods.*" **overlooks the runtime comparison provided explicitly in Table 11 of the original submission.** This indicates that Reviewer QwSQ didn't read the paper carefully. Second, despite our comprehensive rebuttal and repeated requests for the reviewer to reply, the reviewer **remained completely non-responsive throughout the entire discussion period**.
>
> Notably, despite Reviewer QwSQ's silence, the validity of our response can actually be verified by other reviewers. All three issues raised by Reviewer QwSQ were similar to those raised by the engaged reviewers, who both confirmed that our new experiments resolved the issues: (1) on quantization (Reviewer a4Qo Q3): Reviewer a4Qo stated, "The response satisfactorily addresses my main concern" (Maintaining score  8). (2) on LoRA fine-tuning & hardware inference (Reviewer dvHK W4&W5): Reviewer dvHK noted, "Thank you for the detailed rebuttal, which addresses most of my concerns" (Raising score to 6). **As the engaged reviewers were convinced by the same evidence, we are confident that Reviewer QwSQ's related concerns have been resolved.**
>
> ---
>
> # Final Note to the AC
>
> The valid score distribution stands at 8, 6, 6, and 4. Three active reviewers (a4Qo, jYZD, dvHK) fully engaged with the rebuttal, reviewed our new experiments, and confirmed that their technical concerns were resolved, forming a strong positive consensus. In contrast, regarding Reviewer QwSQ (Score 4), **we respectfully request lowering the weight of Reviewer QwSQ’s assessment in the final evaluation**. This is due to their total lack of engagement, a factual oversight of existing experiments in the original submission, and the fact that the main issues raised by Reviewer QwSQ were acknowledged to be resolved by other engaged reviewers (dvHK & a4Qo) who asked similar questions.
>
> Given the novel insights provided by SoCo and the positive consensus from the active reviewers, we believe the contributions of this work are valuable and meaningfully advance the field of large language model compression.
>
> Finally, we extend our sincere gratitude to the Area Chair and all reviewers for their time and effort. We especially appreciate the AC's careful handling of the procedural challenges and the reviewers' constructive feedback, which has significantly strengthened the quality and scope of our work.
>
>
> Best regards,
>
> Authors

---

### Meta-Review · Area_Chair_LkNw · 2026-01-08

**Summary:**

This submission was evaluated as borderline, and we recommend rejection due to the remaining concerns raised by multiple reviewers. One reviewer maintained a low overall score (4), reflecting significant reservations about the contribution, including concerns about the clarity of the proposed method, the strength of the empirical evidence, and whether the current experiments sufficiently support the claims made in the paper. A second reviewer increased their score from 4 to 6, but continued to note unresolved issues.  Overall, although the paper shows promise, the combination of one review with major concerns and another that remains only moderately positive, together with dependence on rebuttal-only clarifications, leads us to recommend rejection at this time. The paper could be strengthened by integrating the rebuttal clarifications into the main text, improving the clarity of the method description, and expanding or better contextualizing the experimental evaluation to directly address the reviewers’ concerns.

**Reviewer Concerns:**

The rebuttal successfully addressed several clarification-oriented concerns raised by Reviewer jYZD (score increased from 4 to 6). In particular, the rebuttal provided additional explanations regarding the method design choices and experimental details, which improved the reviewer’s understanding of the work. Reviewer a4Qo (score 8) indicated satisfaction with the rebuttal and confirmed the technical soundness of the approach.

However, substantive concerns remain outstanding. Reviewer dvHK, while having raised their score from 4 to 6 prior to the OpenReview incident, continued to express concerns in their updated review, indicating that key issues were not fully resolved. Reviewer QwSQ maintained a low score (4) and did not engage in further discussion, leaving their original concerns unaddressed. Across these reviews, a common issue is that several important clarifications provided during rebuttal are not yet incorporated into the manuscript, leaving aspects of the method description and experimental evaluation insufficiently clear in the paper itself.

Overall, while the rebuttal improved understanding for some reviewers, it did not fully resolve the concerns necessary to reach consensus for acceptance. The paper would benefit from integrating the rebuttal clarifications into the main text and further strengthening the presentation and evaluation to address the remaining reviewer concerns.

**Reviewer Scores:**

Reviewer jYZD: This reviewer already updated their score from 4 to 6 after receiving the rebuttal, indicating that the additional clarifications addressed a meaningful portion of their concerns. With full participation in the discussion, it is likely the reviewer would have maintained a score around 6, but there is no clear evidence suggesting they would have increased it further without corresponding changes incorporated into the manuscript.

Reviewer dvHK: This reviewer raised their score from 4 to 6 during the rebuttal period (prior to the OpenReview incident), suggesting partial resolution of concerns. However, their updated review indicates that some substantive issues remained. With full discussion participation, the reviewer would likely have remained at a borderline score (around 6) rather than moving to a strong accept.

Reviewer a4Qo: This reviewer maintained a consistently high score (8) and explicitly acknowledged satisfaction with the rebuttal and the technical soundness of the work. Full participation would likely have confirmed their existing positive assessment, with no expected change in score.

Reviewer QwSQ: This reviewer maintained a low score (4) and did not engage in discussion despite reminders. Given the lack of engagement and absence of evidence that rebuttal addressed their concerns, it is likely that even with full participation, the reviewer would have maintained a low score or only marginally adjusted it, unless substantial revisions were made to the manuscript.

---

### Decision · Program_Chairs · 2026-01-26

Reject